

# Internal Levin-Wen models

Vincentas Mulevičius[1,2*], Ingo Runkel[3] and Thomas Voß[3]

**1** Erwin Schrödinger Institute for Mathematics and Physics, Vienna, Austria
**2** Institute of Theoretical Physics and Astronomy, Vilnius University, Lithuania
**3** Fachbereich Mathematik, Universität Hamburg, Germany

* vincentas.mulevicius@ff.vu.lt

## Abstract

Levin–Wen models are a class of two-dimensional lattice spin models with a Hamiltonian that is a sum of commuting projectors, which describe topological phases of matter related to Drinfeld centres. We generalise this construction to lattice systems internal to a topological phase described by an arbitrary modular fusion category $\mathcal{C}$. The lattice system is defined in terms of an orbifold datum $\mathbb{A}$ in $\mathcal{C}$, from which we construct a state space and a commuting-projector Hamiltonian $H_\mathbb{A}$ acting on it. The topological phase of the degenerate ground states of $H_\mathbb{A}$ is characterised by a modular fusion category $\mathcal{C}_\mathbb{A}$ defined directly in terms of $\mathbb{A}$. By choosing different $\mathbb{A}$'s for a fixed $\mathcal{C}$, one obtains precisely all phases which are Witt-equivalent to $\mathcal{C}$. As special cases we recover the Kitaev and the Levin–Wen lattice models from instances of orbifold data in the trivial modular fusion category of vector spaces, as well as phases obtained by anyon condensation in a given phase $\mathcal{C}$.

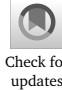

# 1 Introduction

A topological phase of matter is identified by a variety of exotic phenomena, most notably topological ground state degeneracy, that is, the dimension of the ground state space depends only on the topology of the system, not its size. Topological phases of matter can be modelled by physical systems whose effective theory at low energies is a topological quantum field theory (TFT). We refer to e.g. [51, Sec. 4] and [58, Ch. 6] for the physical background. In (2+1)-dimensions, there are two widely studied types of TFTs: state sum TFTs, also called Turaev–Viro–Barrett–Westbury models [11, 54], and surgery TFTs, aka. Reshetikhin–Turaev models [50, 53]. The surgery TFTs are more general, in that they reproduce the state sum TFTs as a special case.

The algebraic input datum defining a surgery TFT is a modular fusion category (MFC) $\mathcal{C}$. The same datum describes a distinguishing characteristic of a 2-dimensional topological phase of matter, namely the braiding statistics and the fusion rules of its point-like excitations, called anyons. We will often refer to such a topological phase of matter as "phase $\mathcal{C}$".

If the MFC $\mathcal{C}$ is given by the Drinfeld centre $\mathcal{Z}(\mathcal{S})$ of a spherical fusion category $\mathcal{S}$, two related special features occur. Firstly, the surgery TFT for $\mathcal{Z}(\mathcal{S})$ has an equivalent state sum description defined directly in terms of $\mathcal{S}$ (see e.g. [55]), and secondly, the corresponding topological phase of matter can be realised via a local lattice model. Such models include Kitaev's toric code and its variations – the quantum double models obtained from finite groups or, more generally, finite-dimensional semisimple Hopf algebras, as well as Levin–Wen models, see Figure 1. Levin–Wen models are defined in terms of spherical fusion categories $\mathcal{S}$ and are universal in the sense that they provide a local description of all phases given by Drinfeld centres $\mathcal{Z}(\mathcal{S})$. For a generic phase $\mathcal{C}$ no similar local lattice realisation is known.

The goal of this paper is to provide a weaker "non-local" version of a lattice realisation for phases that possibly are not Drinfeld centres. Non-locality here means that the overall state space of the lattice model will *not* in general be of the form $(\mathbb{C}^n)^{\otimes N}$, i.e. it will not consist of $N$ independent degrees of freedom $\mathbb{C}^n$. Instead, the state space is a lattice of anyonic particles

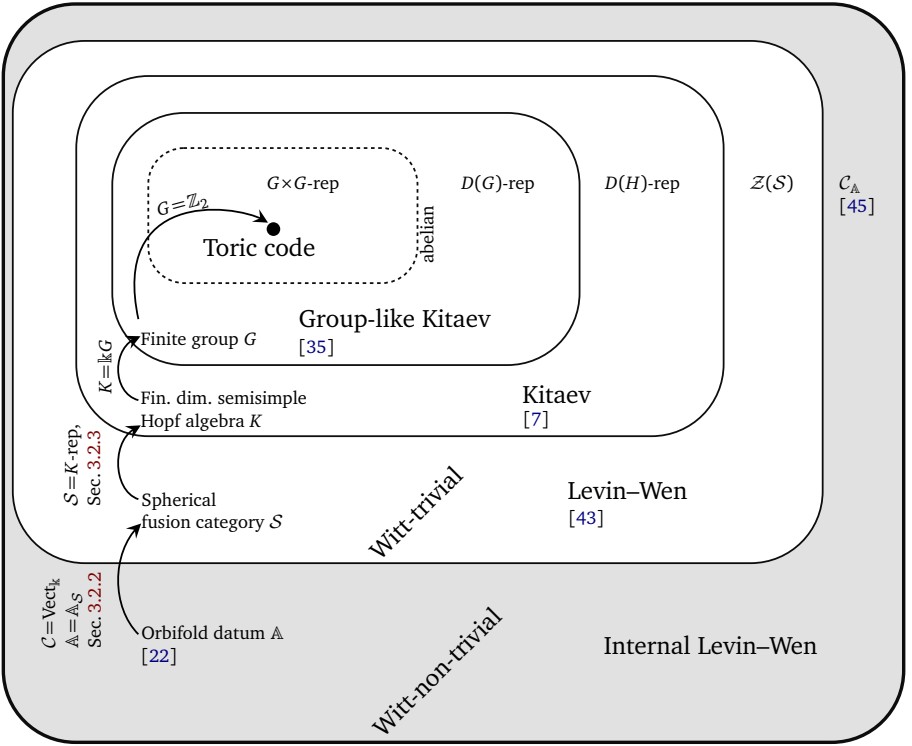

Figure 1: Hierarchy of lattice descriptions of topological phases of matter. The inner white boxes are local lattice models for phases corresponding to Drinfeld centres. The outer gray box amounts to the non-local lattice models introduced in this paper.

$X$ and their antiparticles $X^*$ in an "initial" – or "ambient" – phase $\mathcal{C}$:

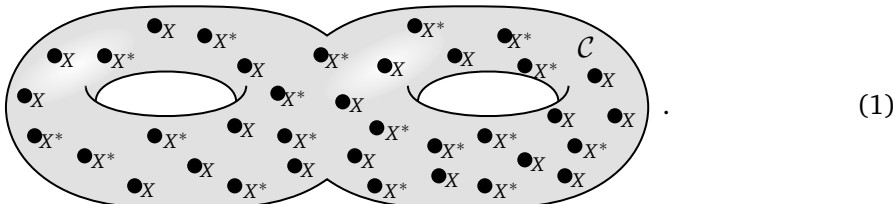

(1)

On the spaces of states for a surface in phase $\mathcal{C}$ with excitations $X$ and $X^*$ as shown above we introduce a commuting-projector Hamiltonian $H_{\mathbb{A}}$. The Hamiltonian is defined in terms of a so-called *orbifold datum* $\mathbb{A}$ in $\mathcal{C}$ which we discuss in more detail below. The ground state space of $H_{\mathbb{A}}$ will again be a topological phase of matter, but now for a different MFC $\mathcal{C}_{\mathbb{A}}$, i.e. our model "condenses phase $\mathcal{C}$ into phase $\mathcal{C}_{\mathbb{A}}$".

We refer to this construction as "internal Levin–Wen model" for two reasons. Firstly, as illustrated in (1), our model is formulated internally to a fixed ambient phase $\mathcal{C}$. Secondly, if we choose the phase $\mathcal{C}$ to be trivial, i.e. if we take $\mathcal{C}$ to be the MFC $\mathrm{Vect}_{\mathbb{C}}$ of finite dimensional vector spaces, the possible condensed phases $\mathcal{C}_{\mathbb{A}}$ turn out to be exactly the Drinfeld centres. In fact, in this particular case the lattice models defined here turn out to be local, and, for an appropriate choice of $\mathbb{A}$, are equivalent to the original Levin–Wen lattice systems.

**Universality of internal Levin–Wen models**

Our models unite and generalise two seemingly different constructions encountered in the study of topological phases: that of Levin–Wen string-net models and that of (bosonic) anyon

condensation. The latter in particular uses the aforementioned idea of bringing together (condensing) a number of anyons in an existing phase $\mathcal{C}$ to make up a new phase [9]. The anyons being condensed are taken to have bosonic properties – trivial twist and (almost) trivial self-braiding (see [9, Sec. III]). The mathematically precise statement is that this construction requires a *condensable algebra $A$* in the initial phase $\mathcal{C}$ [38]. The condensed phase is given by the MFC $\mathcal{C}_A^{\mathrm{loc}}$ of the so-called local $A$-modules and tends to be simpler than the initial phase since its total quantum dimension is smaller: $\mathrm{Dim}\,\mathcal{C}_A^{\mathrm{loc}} = \mathrm{Dim}\,\mathcal{C}/(\dim(A))^2$ (in particular $\mathcal{C}_A^{\mathrm{loc}}$ tends to have fewer simple objects (anyon species) than $\mathcal{C}$).

We will see later in this section (and in detail in Section 5) that such anyon condensations constitute examples of internal Levin–Wen models with the initial phase $\mathcal{C}$ being arbitrary, possibly even chiral, i.e. non-equivalent to a Drinfeld centre, and the phase of the ground states being $\mathcal{C}_A^{\mathrm{loc}}$. In particular, if the initial phase does happen to be non-chiral, i.e. $\mathcal{C} \cong \mathcal{Z}(\mathcal{S})$, the choice of the so-called *Lagrangian algebra $A \in \mathcal{Z}(\mathcal{S})$* condenses $\mathcal{Z}(\mathcal{S})$ to the vacuum phase $\mathrm{Vect}_{\mathbb{C}}$. Interestingly, the Levin–Wen model, which is also an example of our *internal* Levin–Wen models, does exactly the opposite – starting with excitations in the vacuum phase (i.e. local degrees of freedom) it produces a non trivial phase $\mathcal{Z}(\mathcal{S})$, i.e. in some sense they *invert the condensation* procedure. This is not a coincidence, as our models are in fact capable of implementing both the anyon condensation and its inversion procedures in an arbitrary (possibly chiral) phase $\mathcal{C}$. This follows from the analysis [47] of the properties of the oribifold data – the algebraic structures used in the construction of our models.

The ability to produce a more intricate phase out of a given one is where we see the biggest advantage of introducing the internal Levin–Wen models. At their core, they describe, assuming a full control over an anyon species $\mathcal{C}$, what states of several anyons in $\mathcal{C}$ to start with and how to braid/fuse them etc. so that another phase with new anyons would emerge. While in this paper we will only focus on defining the internal Levin–Wen models as well as showing that our definition reproduces some known constructions, let us outline one example, which we believe not be covered by previous constructions. Namely, consider two chiral topological phases, one of them being the Ising phase with 3 anyons, and the other being the $su(2)_{10}$ phase, which has 11 anyons. The $su(2)_{10}$ phase contains a condensible algebra, usually referred to as the $E_6$ algebra, whose underlying object is $\underline{0} \oplus \underline{6}$ (see [39, Sec. 6]). The condensed phase is known to be of Ising type, so in this example the number of anyon species was indeed reduced. Our model now allows one to explicitly invert the condensation, i.e. construct a state space and a commuting projector Hamiltonian inside the Ising phase, whose ground state space constitutes the $su(2)_{10}$ phase:

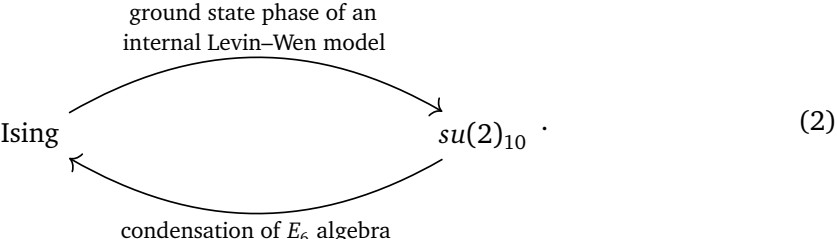

$$ \tag{2} $$

For example, the anyon $X = X^*$ to be inserted at the vertices of the lattice as in (1) in this instance can be chosen to be $X = \mathbb{1}^{\oplus 4} \oplus \sigma$ (where we denote by $\mathbb{1}, \varepsilon, \sigma$ with $\varepsilon \otimes \varepsilon \cong \mathbb{1}$ and $\sigma \otimes \sigma \cong \mathbb{1} \oplus \varepsilon$ the Ising anyons). The Hamiltonian is constructed as outlined below, with the input data explicitly provided in [46]. We leave the more detailed analysis of this example for future work.

Finally, let us comment on which other phases one can construct out of the initial phase $\mathcal{C}$ with the help of internal Levin–Wen models. Recall that two MFCs $\mathcal{C}$ and $\mathcal{D}$ are called *Witt-equivalent* [24] if there exists a spherical fusion category $\mathcal{S}$ and an equivalence $\mathcal{C} \boxtimes \widetilde{\mathcal{D}} \simeq \mathcal{Z}(\mathcal{S})$

(of ribbon categories), where $\boxtimes$ denotes the Deligne product and $\widetilde{\mathcal{D}}$ denotes the MFC $\mathcal{D}$ with inverse braiding and twist. In other words, two possibly chiral phases $\mathcal{C}$ and $\mathcal{D}$ are Witt equivalent if the product $\mathcal{C} \boxtimes \widetilde{\mathcal{D}}$ is non-chiral. For example, choosing $\mathcal{D} = \text{Vect}_{\mathbb{C}}$, we see that the Witt-class of the trivial phase $\text{Vect}_{\mathbb{C}}$ consists precisely of the Drinfeld centres (i.e. the non-chiral phases). A more physical description of Witt-equivalence is the requirement that there exists a gapped domain wall separating the phases $\mathcal{C}$ and $\mathcal{D}$ [29, 34, 36, 37]. It is shown in [47, Thm. 7.3] that two MFCs $\mathcal{C}$ and $\mathcal{D}$ are Witt equivalent if and only if there exists an orbifold datum $\mathbb{A}$ in $\mathcal{C}$, such that $\mathcal{C}_{\mathbb{A}} \cong \mathcal{D}$ as ribbon categories. An orbifold datum in $\mathcal{C}$ constitutes the input for an internal Levin–Wen model in $\mathcal{C}$, which implies that the possible phases of ground state-spaces are exactly those that are Witt-equivalent to $\mathcal{C}$.

In this language, one can state a universality property of the original Levin–Wen models: Collectively they provide lattice realisations of all phases in the trivial Witt-class. After these preparations, we can also state the similar universality property for our models:

> Internal Levin–Wen models are universal in the sense that starting from the initial phase $\mathcal{C}$, collectively they provide internal-to-$\mathcal{C}$ lattice realisations of all phases in the Witt-class of $\mathcal{C}$.

The relation between the various models discussed so far is summarised in Figure 1.

**Construction of internal Levin–Wen models**

The main tool used in the construction of our models is a three-dimensional *defect TFT*. Specifically, the defect TFT we need is of surgery type and is an extension of the Reshetikhin–Turaev TFT [50, 53], which includes surface defects in addition to the line defects already present in the original definition. Mathematically, the defect TFT is a symmetric monoidal functor

$$Z_{\mathcal{C}}^{\text{def}} \colon \widehat{\text{Bord}}_3^{\text{def}}(\mathbb{D}^{\mathcal{C}}) \to \text{Vect}_{\mathbb{C}}, \tag{3}$$

defined in terms of a MFC $\mathcal{C}$ [21]. Here $\mathbb{D}^{\mathcal{C}}$ is the set of labels for surface and for line defects, and $\widehat{\text{Bord}}_3^{\text{def}}(\mathbb{D}^{\mathcal{C}})$ is a category of three-dimensional bordisms as morphisms between two-dimensional surfaces as objects. The bordisms contain "foams" of line and surface defects labelled by data in $\mathbb{D}^{\mathcal{C}}$. These foams intersect the boundary surfaces transversally, so that these surfaces contain point and line defects (see Figures 3, 5a below for examples of defect bordisms). We will review the category of defect bordisms, as well as the definition of $Z_{\mathcal{C}}^{\text{def}}$ and its properties in Sections 2.1 and 2.2.

Let $X$ be an object of $\mathcal{C}$ (which need not be simple) and consider a surface $\Sigma_X$ with $m$ point defects labelled by $X$ and $n$ point defects labelled by $X^*$ as shown in (1). To $\Sigma_X$, the TFT $Z_{\mathcal{C}}^{\text{def}}$ assigns a complex vector space which will be the state space $V_X$ of the internal Levin–Wen model on $\Sigma_X$:

$$V_X = Z_{\mathcal{C}}^{\text{def}}(\Sigma_X). \tag{4}$$

One can express $V_X$ as a Hom-space of the category $\mathcal{C}$. For example, if $\Sigma$ is a sphere we have

$$V_X \cong \mathcal{C}(\mathbb{1}, X^{\otimes m} \otimes (X^*)^{\otimes n}). \tag{5}$$

Here $\mathcal{C}(U, W)$ denotes the vector space of morphisms from $U$ to $W$ in $\mathcal{C}$, and $\mathbb{1}$ denotes the tensor unit of $\mathcal{C}$. Note that if $\mathcal{C}$ is $\text{Vect}_{\mathbb{C}}$, then $X$ is a vector space, $X^* \cong X$ is its dual, and (5) can be written as $V_X \cong X^{\otimes(m+n)}$. That is, for $\mathcal{C} = \text{Vect}_{\mathbb{C}}$ – the trivial ambient topological phase – the state space is indeed local.

To construct the commuting projector Hamiltonian, we need the additional input of an *orbifold datum* $\mathbb{A}$ in $\mathcal{C}$. This consists of a label $A$ in $\mathbb{D}^{\mathcal{C}}$ for a surface defect, a label $T$ for a

line defect forming the junction of three surface defects, and two labels $\alpha$, $\overline{\alpha}$ for point defects where two such lines cross. More details including the conditions $\mathbb{A}$ has to satisfy can be found in Section 2.3.

We furthermore need to choose a lattice (or, in mathematical terms, a graph) $\Gamma$ on the surface $\Sigma_X$ whose vertices are trivalent and are given by the point defects on $\Sigma_X$. For example, a local patch of the lattice $\Gamma$ on $\Sigma_X$ could be:

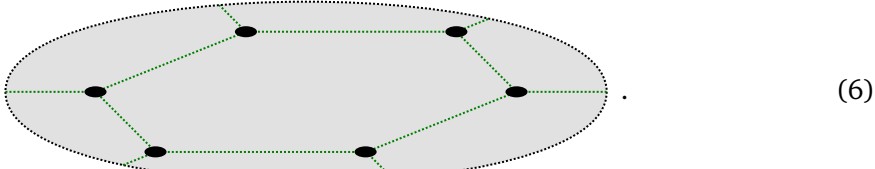

$$\tag{6}$$

We will assume that the lattice $\Gamma$ is fine enough so that the complement of $\Gamma$ in $\Sigma_X$ consists of a disjoint union of discs, which we refer to as *faces*.

For a face $f$ on $\Sigma_X$ (with respect to $\Gamma$), consider the defect bordism $C_f : \Sigma_X \to \Sigma_X$ which has the underlying topology of a cylinder $\Sigma_X \times [0,1]$ and contains vertical $X$ or $X^*$-labelled line defects everywhere except for around the face $f$. Near $f$, the bordism is given by the following arrangement of point, line and surface defects:

$$C_f = \quad \text{\raisebox{-2em}{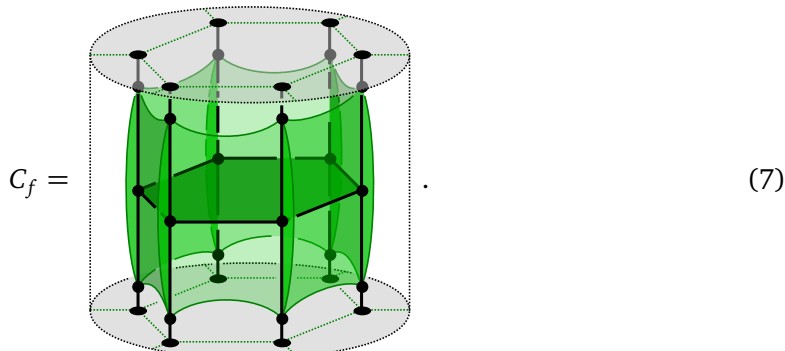}} \quad . \tag{7}$$

This bordism (7) is slightly simplified (for example all orientations are missing), and we explain the full version and all its ingredients in Section 3.1. Here we just note that the labels of the various defects are prescribed by the orbifold datum $\mathbb{A}$: the surface defects in $C_f$ are labelled by $A$, the line defects joining three surface defects by $T$, and the crossings of two $T$-lines by $\alpha$ or $\overline{\alpha}$ (depending on the orientations, see Section 2.3). $X$ is chosen such that $T$ is a subobject of $X$.

Applying the defect TFT $Z_{\mathcal{C}}^{\text{def}}$ to the cylinder $C_f$ gives a linear map

$$P_f := Z_{\mathcal{C}}^{\text{def}}(C_f) : V_X \to V_X . \tag{8}$$

We show in Section 3.1 that $P_f$ is a projector, $P_f^2 = P_f$, and that these projectors commute amongst themselves, $P_f P_{f'} = P_{f'} P_f$ for all faces $f, f'$ of $\Sigma_X$ with respect to $\Gamma$. The Hamiltonian $H_{\mathbb{A}}$ on $V_X$ is simply minus the sum of all face projectors,

$$H_{\mathbb{A}} = -\sum_f P_f . \tag{9}$$

Again, this expression is slightly simplified, and we refer to Section 3.1 for the full version. One important aspect of this paper is that we describe the face projectors $P_f$ not only abstractly as the value of a TFT on a bordism, but also as explicit linear maps on the state space expressed as a Hom-space as in (5). This is done in Section 4.

The ground state space $V_0 = V_0(\Sigma_X, \mathbb{A}, \Gamma) \subset V_X$ of $H_{\mathbb{A}}$ is simply the intersection of the images of all the face-projectors. The key observation is now that $H_{\mathbb{A}}$ and $V_0$ describe a new topological phase, namely that for the MFC $\mathcal{C}_{\mathbb{A}}$. To support this claim, let us denote by $\Sigma$ the surface without any defects underlying $\Sigma_X$. We show (Theorem 8 together with the equivalence in (33)):

**Theorem 1.** For any choice of a trivalent graph $\Gamma$ on $\Sigma$ such that $\Sigma \setminus \Gamma$ is a disjoint union of discs, we have an isomorphism of vector spaces

$$V_0(\Sigma_X, \mathbb{A}, \Gamma) \cong Z^{\mathrm{RT}}_{\mathcal{C}_{\mathbb{A}}}(\Sigma). \tag{10}$$

The theorem implies in particular that $V_0(\Sigma_X, \mathbb{A}, \Gamma)$ is independent of the choice of $\Gamma$, provided $\Sigma \setminus \Gamma$ is a disjoint union of discs. The ground states thus depend only on the topology of $\Sigma$ and not on the microscopic details of the lattice.

**Remark 2.** As the name suggests, the original use of an orbifold datum $\mathbb{A}$ was to define an orbifolding procedure for TFTs, or rather a generalisation thereof beyond group symmetries [20, 22]. Internal Levin–Wen models can in fact be defined for any orbifold datum in any defect TFT, not just the Reshetikhin–Turaev type theories we consider here. The generalised orbifold procedure was originally introduced for two-dimensional conformal field theories in [27]. Considering all such generalised orbifolds simultaneously can be understood as a completion procedure. This has been made precise for 2d TFTs in [19], for 3d TFTs in [15], and in terms of condensation monads and Karoubi-envelopes in [31].

We treat three examples of internal Levin–Wen models in detail, thereby illustrating how the general model reduces to previous constructions of topological phases of matter:

1. ***Condensable algebras (Section 5.1):*** For the orbifold datum $\mathbb{A}_A$ constructed from a condensable algebra $A \in \mathcal{C}$, i.e. a simple symmetric commutative $\Delta$-separable Frobenius algebra, one finds that the condensed phase $\mathcal{C}_{\mathbb{A}_A}$ agrees with the MFC of so-called local modules in $\mathcal{C}$ [45], which is precisely the description of the phase obtained by condensing $A$, see e.g. [9, 38].

2. ***Levin–Wen models (Section 5.2):*** Given a spherical fusion category $\mathcal{S}$, one can define an orbifold datum $\mathbb{A}_{\mathcal{S}}$ in $\mathrm{Vect}_{\mathbb{C}}$ such that the Hamiltonian $H_{\mathbb{A}_{\mathcal{S}}}$ is essentially the original Levin–Wen Hamiltonian from [43] (we include edge projectors and allow for slightly more general vertex projectors).

3. ***Kitaev models (Section 5.3):*** As a generalisation of the original toric code model [35], one can formulate a state space and Hamiltonian in terms of a finite-dimensional semisimple Hopf algebra $K$ [7]. We give an orbifold datum $\mathbb{A}_K$ in $\mathrm{Vect}_{\mathbb{C}}$, such that the Hamiltonian $H_{\mathbb{A}_K}$ agrees with the Hamiltonian of the Kitaev model.

There are many aspects of internal Levin–Wen models not covered in this paper, and to which we hope to return in the future. The three most notable omissions are, firstly, the description of anyon excitations in the condensed phase obtained from $\mathbb{A}$ in order to show that their fusion and braiding indeed agree with those of $\mathcal{C}_{\mathbb{A}}$. Secondly, we expect there to be an analogue of the string net description of Levin–Wen state spaces as given in [5, 43] also for the internal Levin–Wen models. Thirdly, the models listed in 1.–3. above are all already well-studied, and one should investigate some truly new examples. As mentioned before, an interesting candidate would be the Fibonacci-type orbifold data found in an Ising modular category [46], which results in the $su(2)_{10}$ WZW phase.

**Relation to other works**

Let us discuss some relations of our construction of internal Levin–Wen models to other works. **Condensation monads.** There is a strong connection to the work [31]. In that work, a general algebraic language to describe condensations of $n$-dimensional phases is formulated: The algebraic datum needed to describe such a process is a *condensation monad* in a (weak) $n$-category $\mathcal{X}$, in a sense encoding the information of the defects of an $n$-dimensional phase. A condensation monad $\mathcal{A}$ can be thought of as a higher analogue of an idempotent on an object $S \in \mathcal{X}$ and is then used to construct a commuting-projector Hamiltonian internal to the phase $S$ realising the phase comparable to the "image" of $\mathcal{A}$. Although it is the local systems (i.e. obtained when $S$ is the trivial phase and related to fully-extended TFTs) that are arguably of the greatest interest in this setting, the idea of internal models is mentioned as well (see [31, Sec. 1.3]).

Our model can be seen as a special case of this construction with $n = 3$, $\mathcal{X} = \mathrm{B}(\mathrm{Alg}_{\mathcal{C}})$ (delooping of the monoidal bicategory of algebras in a MFC $\mathcal{C}$, bimodules and bimodule morphisms) and $\mathcal{A}$ given by the orbifold datum $\mathbb{A}$. In our setting, the 3-dimensional TFT corresponding to the condensed phase is oriented so that the orbifold datum $\mathbb{A}$ has a built-in datum of a "homotopy fixed point of $SO(3)$", which for general condensation monads is not necessary, cf. [31, Rem. 1.4.7]. One advantage of the description of internal Levin–Wen models given here is that it is very explicit: we give a concrete vector space as state space and a concrete linear map as Hamiltonian (Section 4).

**Boundary theories of Walker–Wang models.** A closely related commuting-projector model is described in [34, §II B], which lives on the boundary of a Walker–Wang model [57]. The latter can be thought of as a three-dimensional generalisation of Levin–Wen models (and are more general than the three-dimensional models given already in [43]). The approach in [34] is via $\mathcal{A}$-enriched categories (for a unitary MFC $\mathcal{A}$). From the $\mathcal{A}$-enriched category one can obtain a unitary fusion category $\mathcal{X}$ such that $\mathcal{Z}(\mathcal{X}) \cong \mathcal{Z}^{\mathcal{A}}(\mathcal{X}) \boxtimes \mathcal{A}$ for a unitary MFC $\mathcal{Z}^{\mathcal{A}}(\mathcal{X})$ (the enriched centre). The phase $\mathcal{Z}^{\mathcal{A}}(\mathcal{X})$ can be realised via a commuting-projector model that lives on the boundary of the Walker–Wang model obtained from $\mathcal{A}$.

This is indeed closely related to how the condensed phase $\mathcal{C}_{\mathbb{A}}$ obtained from an orbifold datum can be described. Namely, by [47, Prop. 7.4] we have $\mathcal{C}_{\mathbb{A}} \boxtimes \widetilde{\mathcal{C}} \cong \mathcal{Z}(\mathcal{L})$, where as above, $\widetilde{\mathcal{C}}$ is $\mathcal{C}$ with inverse braiding and twist, and $\mathcal{L}$ can be thought of as a category of line defects living on the interface of the phases $\mathcal{C}$ and $\mathcal{C}_{\mathbb{A}}$.

The main difference between the boundary Walker–Wang models and our internal Levin–Wen models seems to be the following: the former provide one with a way to *realise* the initial (2+1)-dimensional phase $\mathcal{C}$ by the means of a (3+1)-dimensional *local* lattice model with boundary. This realisation is flexible enough to include domain walls and other defects on the (2+1)-dimensional boundary theory. Our models on the other hand provide one with a procedure to condense defects in the initial phase $\mathcal{C}$ to some other phase $\mathcal{C}_{\mathbb{A}}$ independently on how the phase $\mathcal{C}$ is realised. For example, our models can be "compiled to run" on top of the boundary Walker–Wang models, but can also be implemented in other systems realising the initial phase $\mathcal{C}$. In particular, our model is intrinsically (2+1)-dimensional, and there is no need to pass to 4d in order to define the Hamiltonian and to compute it as a spacific linear map.

The similarities of the mathematical tools used in both of these models does not seem to be coincidental and it would be interesting to give a more detailed relation between the commuting-projector model of [34] and the internal Levin–Wen model defined here.

**Anyon condensation and gauging finite groups.** As mentioned before, the idea of bringing together a number of anyons in an existing phase to make up a new phase was originally considered in terms of a condensable algebra $A$ in a given phase $\mathcal{C}$ [9,38]. The condensed phase is given by the MFC $\mathcal{C}_A^{\mathrm{loc}}$ of local $A$-modules.

If there is a finite group $G$ acting on $\mathcal{C}$, it may be possible to obtain a new phase by gauging $G$. This may be obstructed, and it may be possible in several ways. Technically, if $\mathcal{C}$ is the neutral part of a $G$-crossed ribbon fusion category $\mathcal{B}_G^\times$, one can pass to its equivariantisation $\mathcal{B}_G^{\times,G}$ which is again a MFC. For more details and the interpretation in the context of topological phases see [2, 14, 25].

Both of these constructions stay within a given Witt-class and are special cases of passing from $\mathcal{C}$ to $\mathcal{C}_\mathbb{A}$. But iterating condensations and gaugings does not span the entire Witt-class, so the internal Levin–Wen models are more general in that regard.

***Anyonic chains.*** There is a different class of models where one builds lattice models which are non-local in the sense that they start in an ambient theory of anyons, the so-called anyonic chains [3, 30, 48]. These are 1-dimensional spin chains, and the Hamiltonians typically studied are not of the commuting projector type, but instead are gapless and produce interesting conformal field theories as critical points.

The internal Levin–Wen model shares the idea of starting in a given anyonic system, but the model here is a 2-dimensional spin system and we use a commuting-projector Hamiltonian. This Hamiltonian is gapped by construction and its purpose is to produce a new topological phase of matter via its ground states.

***Further directions.*** The recent works [13, 41] give a parameter-dependent Hamiltonian which can implement the transition from one Witt-trivial phase $\mathcal{Z}(\mathcal{S})$ to another such phase obtained by anyon condensation. It would be interesting to study these transitions also in the present setting of internal Levin–Wen models. In the context of tensor network models one can use boundary conditions in Turaev–Viro–Barrett–Westbury to create interesting states in the ground state space of the Levin–Wen Hamiltonian [42]. The latter has a direct generalisation to our internal Levin–Wen models that would be interesting to investigate. A version of Levin–Wen models which are defined also for non-semisimple input categories was developed in [32, 33]. Here too it would be very interesting to see if an analogous generalisation is possible for internal Levin–Wen models.

## 2  Prerequisites

In this section we review the construction of 3-dimensional topological quantum field theories (TFTs) of Reshetikhin–Turaev type [53, Ch. IV], including its version incorporating line and surface defects [21], their so-called generalised orbifolds [20, 22], as well as the required algebraic notions of modular fusion categories (MFCs) [53, Sec. II.1], [4, Ch. 3], Frobenius algebras [26, 28] and orbifold data. The exposition skips most of the technicalities, a more detailed summary is available e.g. in [17, Sec. 3&4], which is mostly followed here.

### 2.1  TFT with line defects

Let $\Bbbk$ be an algebraically closed field of characteristic zero. For the application to topological phases of matter one can of course set $\Bbbk = \mathbb{C}$. A *modular fusion category (MFC)* is a ribbon fusion category $\mathcal{C}$ with non-degenerate braiding. We will often omit the tensor product symbol between objects of $\mathcal{C}$ to have less cluttered formulas, e.g. we often write $XY$ instead of $X \otimes Y$. Recall that $\mathcal{C}$ is called:

- *fusion* if it is $\Bbbk$-linear, monoidal, rigid, finitely semisimple and the monoidal unit $\mathbb{1} \in \mathcal{C}$ is a simple object; we denote by

$$\mathrm{Irr}_\mathcal{C}, \tag{11}$$

  a fixed set of representatives of isomorphism classes of simple objects of $\mathcal{C}$ such that $\mathbb{1} \in \mathrm{Irr}_\mathcal{C}$,

- *spherical* if each object $X \in \mathcal{C}$ has an associated dual object $X^* \in \mathcal{C}$ as well as compatible left/right (co-)evaluation morphisms $\mathrm{ev}_X : X^*X \rightleftarrows \mathbb{1} : \widetilde{\mathrm{coev}}_X$, $\widetilde{\mathrm{ev}}_X : XX^* \rightleftarrows \mathbb{1} : \mathrm{coev}_X$, such that the left and right categorical trace coincide. We write $\mathrm{tr}\, f \in \mathrm{End}_{\mathcal{C}} \mathbb{1} \cong \Bbbk$ for the trace of $f \in \mathrm{End}_{\mathcal{C}} X$ and $d_X = \mathrm{tr}\,\mathrm{id}_X$ for the categorical dimension of $X \in \mathcal{C}$. We denote by $\mathrm{Dim}\,\mathcal{C} := \sum_{i \in \mathrm{Irr}_{\mathcal{C}}} d_i^2$ the global dimension of $\mathcal{C}$,

- *ribbon* if $\mathcal{C}$ is spherical and equipped with braiding $\{c_{X,Y} : XY \xrightarrow{\sim} YX\}_{X,Y \in \mathcal{C}}$, such that for each object $X \in \mathcal{C}$ the left/right twist morphisms $\theta_X^l, \theta_X^r \in \mathrm{End}_{\mathcal{C}} X$, defined as left/right partial traces of $c_{X,X}$, are equal,

- *non-degenerate* if it is a ribbon fusion and the *s*-matrix $s_{ij} = \mathrm{tr}(c_{j,i} \circ c_{i,j})$, $i, j \in \mathrm{Irr}_{\mathcal{C}}$ is non-degenerate.

For $X, Y \in \mathcal{C}$ we denote the space of morphisms $X \to Y$ by $\mathrm{Hom}_{\mathcal{C}}(X, Y)$ or just $\mathcal{C}(X, Y)$ for short.

Given a MFC $\mathcal{C}$ and a square root $\mathcal{D} = (\mathrm{Dim}\,\mathcal{C})^{1/2}$ of the global dimension, one can define the Reshetikhin–Turaev TFT (RT TFT), which is a symmetric monoidal functor

$$Z_{\mathcal{C}}^{\mathrm{RT}} : \widehat{\mathrm{Bord}}_3^{\mathrm{rib}}(\mathcal{C}) \to \mathrm{Vect}_{\Bbbk}, \tag{12}$$

where the source category is a certain central extension of the category $\mathrm{Bord}_3^{\mathrm{rib}}(\mathcal{C})$. The latter has compact oriented 3-dimensional bordisms with embedded $\mathcal{C}$-coloured ribbon graphs as morphisms and oriented closed surfaces with $\mathcal{C}$-coloured framed points (or punctures) as objects. In this paper we will only use it to evaluate bordisms whose underlying topology is that of the cylinder $\Sigma \times [0, 1]$ for some oriented surface $\Sigma$. The central extension of $\widehat{\mathrm{Bord}}_3^{\mathrm{rib}}(\mathcal{C})$ is in general introduced to eliminate a gluing anomaly, which in the case of composing cylinders can be ignored.

As we will not require most of the technicalities needed to define the functor (12), we only present here some of its properties:

1. Define the object

$$L = \bigoplus_{i \in \mathrm{Irr}_{\mathcal{C}}} i \otimes i^* \in \mathcal{C}. \tag{13}$$

   Consider an object of $\mathrm{Bord}_3^{\mathrm{rib}}(\mathcal{C})$ of the form $\Sigma = (\Sigma, P)$, where $\Sigma$ is an oriented closed connected surface of genus $g$, and $P = \{p_1, \ldots, p_n\}$ is a finite set of framed points, each of which carries a label by an object $X_i \in \mathcal{C}$ (we will call such points punctures). Set $|p_i| = +$ if the framing of $p_i \in P$ coincides with the orientation of $\Sigma$ and $|p_i| = -$ otherwise and, for an object $X \in \mathcal{C}$, denote $X^+ = X$, $X^- = X^*$. One has:

$$Z_{\mathcal{C}}^{\mathrm{RT}}(\Sigma, P) \cong \mathcal{C}(\mathbb{1}, X_1^{|p_1|} \cdots X_n^{|p_n|} L^{\otimes g}). \tag{14}$$

2. Given objects $(\Sigma_a, P), (\Sigma_b, Q)$ as in part 1, a morphism $(\Sigma_a, P) \to (\Sigma_b, Q)$ in $\mathrm{Bord}_3^{\mathrm{rib}}(\mathcal{C})$ is a pair $M = (M, R)$, where $M : \Sigma_a \to \Sigma_b$ is a bordism and $R$ is an embedded ribbon graph, intersecting the boundary at points $P \cup Q$. The linear map $Z_{\mathcal{C}}^{\mathrm{RT}}(M)$ is given by postcomposition

$$\mathcal{C}(\mathbb{1}, X_1^{|p_1|} \cdots X_n^{|p_n|} L^{\otimes g_a}) \to \mathcal{C}(\mathbb{1}, Y_1^{|q_1|} \cdots Y_m^{|q_m|} L^{\otimes g_b}),$$
$$f \mapsto \Omega_M \circ f, \tag{15}$$

   with a morphism $\Omega_M \in \mathcal{C}(X_1^{|p_1|} \cdots X_n^{|p_n|} L^{\otimes g_a}, Y_1^{|q_1|} \cdots Y_m^{|q_m|} L^{\otimes g_b})$. In particular, when evaluating with $Z_{\mathcal{C}}^{\mathrm{RT}}$ one can compose the coupons of embedded graphs, see Figure 2.

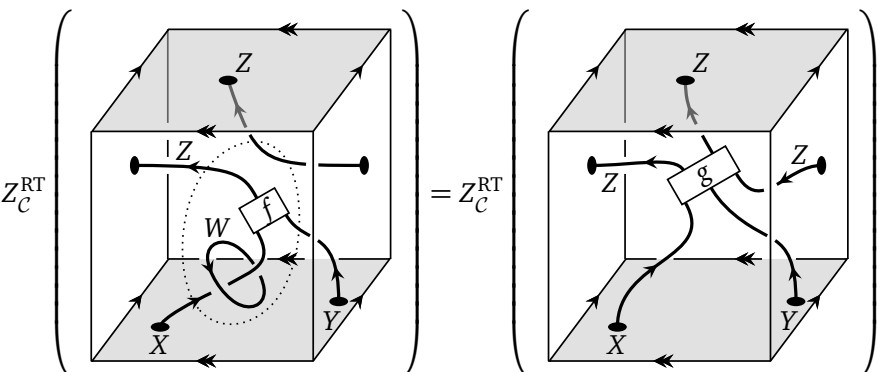

Figure 2: $Z_{\mathcal{C}}^{\mathrm{RT}}$ is invariant under skein relations: replacing the ribbon graph in the dotted ball on the left-hand side by the corresponding coupon on the right-hand side does not change the value of $Z_{\mathcal{C}}^{\mathrm{RT}}$.

3. In case one takes $\mathcal{C} = \mathrm{Vect}_{\mathbb{k}}$, the trivial MFC of finite dimensional vector spaces, the expression (14) simplifies to $X_1^{|p_1|} \cdots X_n^{|p_n|}$, the tensor product of the vector spaces labelling the punctures and their duals. The evaluation (15) of a bordism $(M, R)$ is then obtained by forgetting the topology of $M$ and reading off $R$ as a string diagram in $\mathrm{Vect}_{\mathbb{k}}$.

**Remark 3.** A modular fusion category $\mathcal{C}$ describes a $(2+1)$-dimensional topological order; the objects of $\mathcal{C}$ are interpreted as possible particle-like excitations (anyons) on a surface. The space of states of a surface having several such particles is exactly the space $Z_{\mathcal{C}}^{\mathrm{RT}}(\Sigma)$ assigned to a surface $\Sigma \in \mathrm{Bord}_3^{\mathrm{rib}}(\mathcal{C})$ with marked points. A ribbon graph embedded in the cylinder $\Sigma \times [0,1]$ describes a process in which the particles move around the surface and possibly interact at points, labelled by morphisms in $\mathcal{C}$.

## 2.2 TFT with line and surface defects

A defect TFT [12, 18, 20, 23] is a symmetric monoidal functor having the source category $\mathrm{Bord}_n^{\mathrm{def}}(\mathbb{D})$ of stratified[1] bordisms, with strata carrying labels from a so-called defect datum $\mathbb{D}$. The RT TFT (12) can be generalised to an instance of such defect TFTs [21], see also [15, 29, 37, 40],

$$Z_{\mathcal{C}}^{\mathrm{def}} \colon \widehat{\mathrm{Bord}_3^{\mathrm{def}}}(\mathbb{D}^{\mathcal{C}}) \to \mathrm{Vect}_{\mathbb{k}} \,. \tag{16}$$

The defect datum $\mathbb{D}^{\mathcal{C}}$ consists of: symmetric $\Delta$-separable Frobenius algebras in $\mathcal{C}$ (labels for 2-strata, i.e. surfaces), their (multi-)modules (labels for 1-strata, i.e. lines) and (multi-)module morphisms (labels for 0-strata, i.e. points). The overhat denotes a central extension analogous to that in (12), which as before will be ignored here.

Below we will rely on the graphical calculus of string diagrams to depict morphisms. Our convention is to read diagrams from bottom to top. String diagrams for spherical categories have oriented strands and can be deformed up to a plane isotopy, with downwards orientation denoting the dual object and the cups/caps denoting the (co)evaluation morphisms. In the case of a ribbon category, string diagrams can be read as isotopy classes of ribbon tangles with fixed base points.

---

[1]A *stratification* of a manifold $M$ is a filtration into subspaces $M = F_n \supseteq F_{n-1} \supseteq \cdots \supseteq F_0 \supseteq F_{-1} = \varnothing$, subject to certain non-degeneracy conditions. In particular, $F_j \setminus F_{j-1}$ is a (possibly empty) $j$-dimensional manifold whose connected components are called *$j$-strata* (or points, lines, surfaces depending on dimension). See e.g. [18, Sec. 2.1] for more details.

The precise definition of the defect TFT (16) is laid out in [21, Sec. 5] and summarised in [17, Sec. 3.2]. Here we will only need some properties of it. In particular, let us review the notions used in defining the set of labels $\mathbb{D}^{\mathcal{C}}$:

- A symmetric $\Delta$-separable Frobenius algebra $A \in \mathcal{C}$ is a simultaneous algebra/coalgebra object in $\mathcal{C}$, whose product $\mu = $ ⋀, coproduct $\Delta = $ ⋎, unit $\eta = $ ⌊, and counit $\varepsilon = $ ⌐ in addition satisfy the identities

$$
\underbrace{\qquad = \qquad = \qquad}_{\text{Frobenius}}, \quad \underbrace{\qquad = \qquad}_{\text{symmetric}}, \quad \underbrace{\qquad = \qquad}_{\Delta\text{-separable}}. \tag{17}
$$

- A module of a Frobenius algebra is just a module of the underlying algebra. For Frobenius algebras, each module is also a comodule via $M \xrightarrow{(\Delta \circ \eta) \otimes \mathrm{id}_M} AAM \xrightarrow{\mathrm{id}_A \otimes \rhd} AM$, where $\rhd : AM \to M$ is the action.

- For a $\Delta$-separable Frobenius algebra $A \in \mathcal{C}$, the relative tensor product of a right module $K \in \mathcal{C}$ and a left module $L \in \mathcal{C}$ can be expressed as the image of an idempotent:

$$
K \otimes_A L = \mathrm{im} \quad \begin{array}{c} K \quad L \\ \vdots \\ K \quad L \end{array} . \tag{18}
$$

- If $A, B \in \mathcal{C}$ are Frobenius algebras, so is their product $AB \in \mathcal{C}$, where we take the multiplication to be $ABAB \xrightarrow{c_{B,A}} AABB \to AB$ and the comultiplication $AB \to AABB \xrightarrow{c_{B,A}^{-1}} ABAB$. Given two sets of algebras $A_1, \ldots, A_n \in \mathcal{C}$, $B_1, \ldots, B_m \in \mathcal{C}$, a multimodule is a $(A_1 \cdots A_n)$-$(B_1 \cdots B_m)$-bimodule. Equivalently, it is an object $M \in \mathcal{C}$, which has a left action for each $A_i$ and a right action for each $B_j$, such that, for $i < j$ and $k < l$,

$$
\begin{array}{c} M \\ \vdots \\ A_i \ A_j \ M \end{array} = \begin{array}{c} M \\ \vdots \\ A_i \ A_j \ M \end{array}, \quad \begin{array}{c} M \\ \vdots \\ M \ B_k \ B_l \end{array} = \begin{array}{c} M \\ \vdots \\ M \ B_k \ B_l \end{array}, \quad \begin{array}{c} M \\ \vdots \\ A_i \ M \ B_k \end{array} = \begin{array}{c} M \\ \vdots \\ A_i \ M \ B_k \end{array} . \tag{19}
$$

- Given two multimodules $M, N$ as above, a multimodule morphism is a $(A_1 \cdots A_n)$-$(B_1 \cdots B_m)$-bimodule morphism $M \to N$. Equivalently, it is a morphism $f : M \to N$ in $\mathcal{C}$ commuting with the $A_i$- and $B_j$-actions.

It is convenient to generalise the graphical calculus of $\mathcal{C}$ to accommodate the morphisms between relative products of multimodules of symmetric $\Delta$-separable Frobenius algebras. This is done by using surface diagrams, where a surface depicts an algebra action, see Figures 3 and 4.

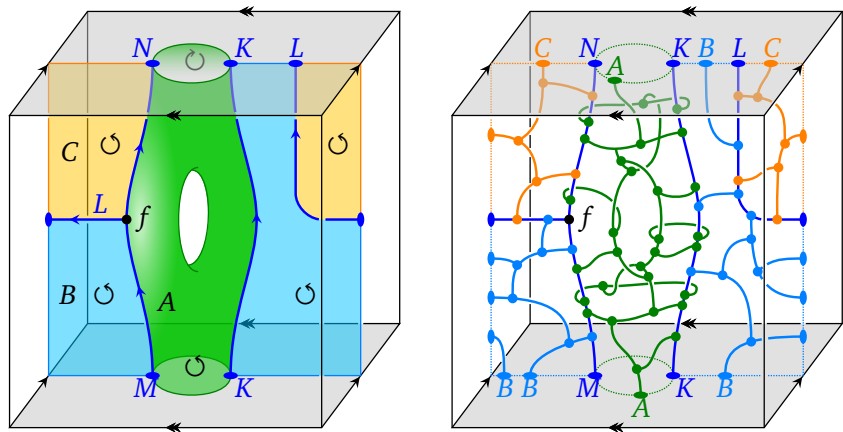

Figure 3: Replacing a stratification of a manifold (depicted for the cylinder of the torus $T^2 \times [0,1]$) by a ribbon graph. Cf. [17, Fig. 1].

**Remark 4.** It was explained in [18] that a 3-dimensional defect TFT has an associated tricategory with duals, whose $k$-morphisms are labels for codimension-$k$ strata. For $Z_{\mathcal{C}}^{\text{def}}$ it is the monoidal bicategory (equivalently, tricategory with one object) $\mathcal{F}\text{rob}^{\text{s}\Delta}$ of symmetric $\Delta$-separable Frobenius algebras, their bimodules and bimodule morphisms in $\mathcal{C}$. The surface diagrams are exactly the ones coming from the graphical calculus of tricategories [8] in this case.

Roughly, the defect TFT (16) works by replacing the 2-strata of a morphism in $\text{Bord}_3^{\text{def}}(\mathbb{D}^{\mathcal{C}})$ with networks of ribbon graphs and evaluating with $Z_{\mathcal{C}}^{\text{RT}}$, see Figure 3. This implies the following properties of it:

1. The category $\text{Bord}_3^{\text{rib}}(\mathcal{C})$ can be seen as a subcategory of $\text{Bord}_3^{\text{def}}(\mathbb{D}^{\mathcal{C}})$, where stratifications do not have 2-strata. The restriction of $Z_{\mathcal{C}}^{\text{def}}$ to this subcategory is precisely the TFT $Z_{\mathcal{C}}^{\text{RT}}$, see [21, Rem. 5.9].

2. For an object $\Sigma \in \text{Bord}_3^{\text{def}}(\mathbb{D}^{\mathcal{C}})$, let $C_\Sigma \colon \Sigma \to \Sigma$ in $\text{Bord}_3^{\text{def}}(\mathbb{D}^{\mathcal{C}})$ be the stratified cylinder bordism $\Sigma \times [0,1]$ and denote by $R(C_\Sigma, t)$ the corresponding bordism in $\text{Bord}_3^{\text{rib}}(\mathcal{C})$ obtained as in Figure 3 (here $t$ denotes the collection of ribbon graphs replacing the 2-strata of $C_\Sigma$). One has:
$$Z_{\mathcal{C}}^{\text{def}}(\Sigma) \cong \text{im} \, Z_{\mathcal{C}}^{\text{RT}}(R(C_\Sigma, t)). \tag{20}$$

3. Evaluation with $Z_{\mathcal{C}}^{\text{def}}$ is invariant upon performing graphical calculus of surface diagrams of symmetric $\Delta$-separable Frobenius algebras inside stratified balls [18, Sec. 3.3].

4. For the choice $\mathcal{C} = \text{Vect}_{\Bbbk}$, the evaluation with $Z_{\mathcal{C}}^{\text{def}}$ only depends on the topology of the union of 2-, 1- and 0-strata, and not on their embeddings into the 3-manifold. For an arbitrary MFC $\mathcal{C}$ the embedding does make a difference, for example the dimension of the vector space assigned to a torus with a single defect line depends on whether the line is contractible (cf. [21, Sec. 5]).

## 2.3 Internal state-sum construction

The *internal state-sum* or *generalised orbifold* takes as input a defect TFT and a so-called *orbifold datum* and produces a new TFT of the same dimension [15, 16, 19, 20, 27]. We will use the internal state sum construction for RT TFTs and will review it in some detail in this

section. In Section 3 we will employ this construction to define the Hamiltonian of the internal Levin-Wen model, and to compute its space of ground states.

The defect TFT we consider is $Z_{\mathcal{C}}^{\text{def}}$ for a MFC $\mathcal{C}$ as reviewed in the previous section. An *orbifold datum* for $\mathcal{C}$ is a tuple

$$\mathbb{A} = (A, T, \alpha, \overline{\alpha}, \psi, \phi), \tag{21}$$

where

- $A \in \mathcal{C}$ is a symmetric $\Delta$-separable Frobenius algebra and so has (co)product and (co)unit such that the relations (17) are satisfied. Note that $A$ is a label for a surface defect in $Z_{\mathcal{C}}^{\text{def}}$, see Figure 4a.

- $T \in \mathcal{C}$ is an $A$-$A \otimes A$-bimodule, i.e. a label for a line defect with three adjacent $A$-labelled surfaces as in Figure 4b. Equivalently a multimodule having one left $A$-action and two right $A$-actions, which will be denoted by:

$$\triangleright_0 = \quad , \quad \triangleleft_1 = \quad , \quad \triangleleft_2 = \quad , \tag{22}$$

and which satisfy the mutimodule conditions (19), which in this case read

$$\tag{23}$$

Note that the dual object $T^* \in \mathcal{C}$ can be seen as an $A \otimes A$-$A$-bimodule, whose actions are defined/denoted by:

$$\triangleleft_0 := \quad , \quad \triangleright_1 := \quad , \quad \triangleright_2 := \quad . \tag{24}$$

- $\alpha : T \otimes_2 T \rightleftarrows T \otimes_1 T : \overline{\alpha}$ are $A$-$A \otimes A \otimes A$ bimodule morphisms between two relative tensor products of multimodules $T$, turning $\alpha, \overline{\alpha}$ into labels for the two point defects as in Figures 4c–d. Here and below $\otimes_i$ denotes the relative tensor product over $A$ where $A$ acts to the left by $\triangleleft_i$ and to the right by $\triangleright_0$. Equivalently, $\alpha$ and $\overline{\alpha}$ can be given by balanced morphisms $\alpha : T \otimes T \to T \otimes T$, $\overline{\alpha} : T \otimes T \to T \otimes T$ in $\mathcal{C}$, meaning that they commute with various $A$-actions as follows:

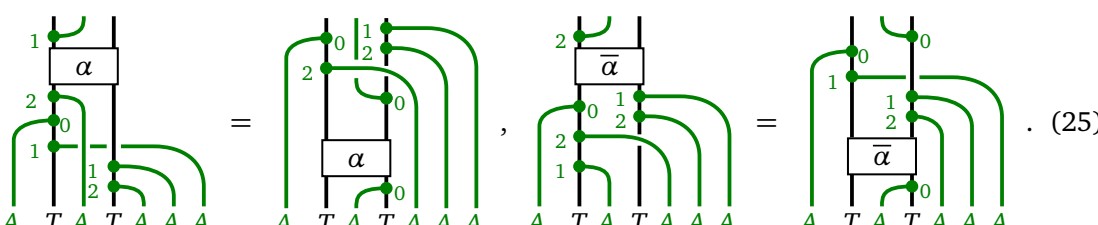

$$\tag{25}$$

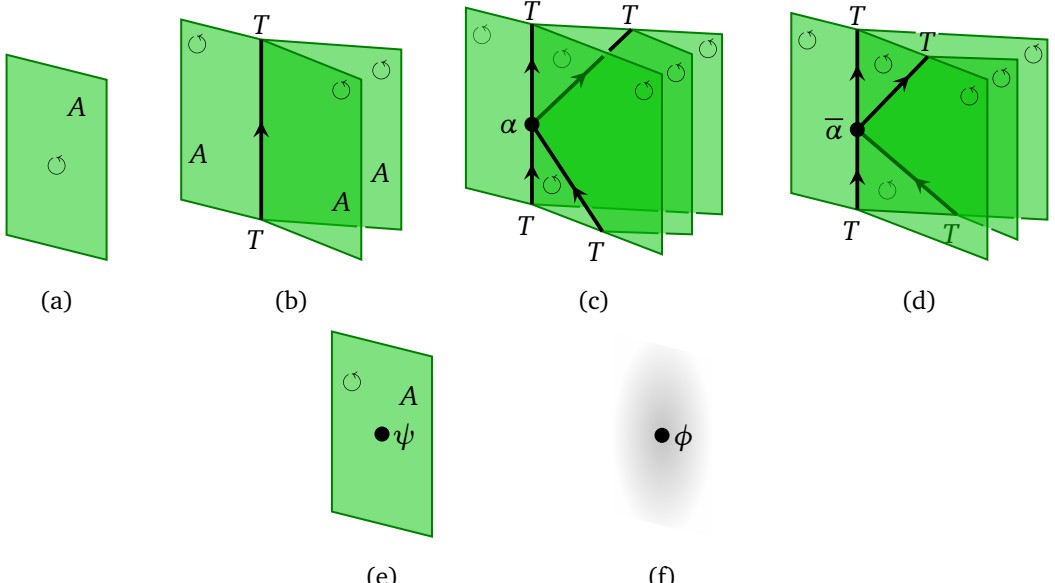

Figure 4: Defect configurations labelled by the entries of an orbifold datum.

- $\psi\colon A \to A$ is an *A-A*-bimodule isomorphism, i.e. a label for an invertible point defect on an *A*-labelled 2-stratum, see Figure 4e. Acting with $[\mathbb{1} \xrightarrow{\eta} A \xrightarrow{\psi} A]$ on a multimodule one obtains multimodule morphisms, which in the case of $T$ we denote (assuming $i, j \in \{0, 1, 2\}$ in the last equation):

$$\psi_0 := \;\psi\,, \quad \psi_1 := \;\psi\,, \quad \psi_2 := \;\psi\,, \quad \psi_{i,j} := \psi_j \circ \psi_i\,. \tag{26}$$

- $\phi \in \mathbb{k}^\times$ is a scalar, i.e. a label for an invertible point defect on a canonically labelled 3-stratum, see Figure 4f.

Moreover, an orbifold datum has to satisfy conditions (O1)–(O8) listed in Appendix A as string diagrams in Figure 22 and as surface diagrams in Figure 23.

Given an orbifold datum $\mathbb{A}$ in $\mathcal{C}$, the internal state sum construction applied to the defect TFT $Z_{\mathcal{C}}^{\text{def}}$ from Section 2.2 yields a symmetric monoidal functor

$$Z_{\mathcal{C}}^{\text{orb}\,\mathbb{A}}\colon \widehat{\text{Bord}}_3 \to \text{Vect}_{\mathbb{k}}\,, \tag{27}$$

whose construction we now review.

Given a bordism $M\colon \Sigma \to \Sigma'$ in $\text{Bord}_3$ one assigns to it an *admissible 2-skeleton*, i.e. a stratification $S$ such that its 3-strata are contractible and each point has a neighbourhood which locally looks like one of the stratifications depicted in Figure 4a–d (see Figure 5a for an example). One defines an *admissible 1-skeleton* of a surface in a similar way, for example $S$ restricts to admissible 1-skeleta $\Gamma$, $\Gamma'$ on the boundary components $\Sigma$, $\Sigma'$. For an admissible 2-skeleton $S$ of $M$, we denote by $S(\mathbb{A})$ the $\mathbb{A}$-*decoration* of $S$, which is obtained by assigning the labels $A$, $T$, $\alpha$, $\overline{\alpha}$ respectively to 2-, 1- and positively/negatively oriented 0-strata of $S$ (Figure 4a–d), as well as using $\psi$ and $\phi$ (Figure 4e,f) to perform the *Euler completion* on 2- and 3-strata of $S$. The latter means that each 2-stratum $F \subseteq S$ gets an additional point defect labelled by $\psi^{\chi_{\text{sym}}(F)}$ and each 3-stratum $U \subseteq S$ an additional point defect labelled by $\phi^{\chi_{\text{sym}}(U)}$, where $\chi_{\text{sym}}$ is the symmetric Euler characteristic of the corresponding stratum, defined as

$$\chi_{\text{sym}}(-) := 2\chi(-) - \chi(- \cap \partial M)\,. \tag{28}$$

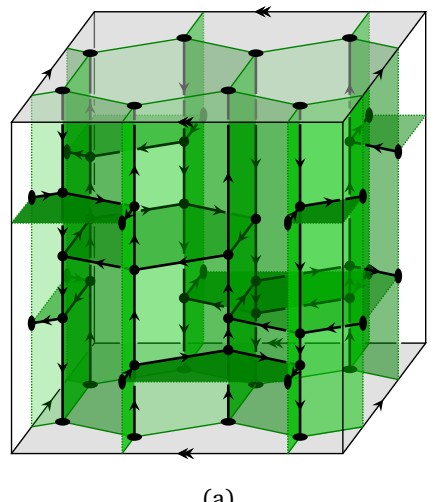
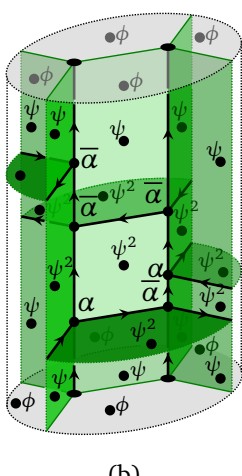

(a)                                              (b)

Figure 5: (a) An admissible skeleton of a manifold (depicted for the cylinder of a torus $T^2 \times [0,1]$). (b) Euler completion of 2- and 3-strata of an $\mathbb{A}$-decorated admissible skeleton by point insertions $\psi$ and $\phi$.

In particular, if e.g. $F$ is contractible, it receives an additional $\psi^2$-insertion if $\partial M \cap F = \varnothing$ and a $\psi$-insertion otherwise (2-strata of an admissible skeleton can only intersect $\partial M$ at a single connected component), see Figure 5b for an illustration.

The internal state-sum TFT $Z_{\mathcal{C}}^{\mathrm{orb}\,\mathbb{A}}$ is constructed as follows: Let $S(\mathbb{A})$ denote the $\mathbb{A}$-decoration of an admissible 2-skeleton of a bordism $M \colon \Sigma \to \Sigma'$ in $\mathrm{Bord}_3$. On the boundaries $\Sigma, \Sigma'$ it restricts to stratifications $\Gamma(\mathbb{A})$, $\Gamma'(\mathbb{A})$, which we appropriately call $\mathbb{A}$-decorations of admissible 1-skeleta. We denote by $(M, S(\mathbb{A})) \colon (\Sigma, \Gamma(\mathbb{A})) \to (\Sigma', \Gamma'(\mathbb{A}))$ the corresponding stratified bordism in $\mathrm{Bord}_3^{\mathrm{def}}(\mathbb{D}^{\mathcal{C}})$. Evaluating it with the defect TFT $Z_{\mathcal{C}}^{\mathrm{def}}$ one obtains the linear map

$$\Psi_\Gamma^{\Gamma'}(M) := Z_{\mathcal{C}}^{\mathrm{def}}(M, S(\mathbb{A})) \ : \ Z_{\mathcal{C}}^{\mathrm{def}}(\Sigma, \Gamma(\mathbb{A})) \longrightarrow Z_{\mathcal{C}}^{\mathrm{def}}(\Sigma', \Gamma'(\mathbb{A})). \tag{29}$$

The notation $\Psi_\Gamma^{\Gamma'}(M)$ intentionally does not mention the admissible 2-skeleton $S$, as one has (see [20, Thm. & Def. 3.10]):

**Proposition 5.** The linear maps $\Psi_\Gamma^{\Gamma'}(M)$ in (29) depend on the choice of $S$ only at the boundary of $M$, i.e. only on $\Gamma, \Gamma'$.

This is a direct corollary to an orbifold datum $\mathbb{A}$ being subject to the identities (O1)–(O8) upon evaluating with $Z_{\mathcal{C}}^{\mathrm{def}}$. Indeed they ensure that away from boundary the skeleton can be modified by what we call the admissible BLT (bubble–lune–triangle) moves (see Figure 6); any two skeletons restricting to $\Gamma(\mathbb{A})$, $\Gamma'(\mathbb{A})$ at the boundary can be related by a finite sequence of such moves [16, Thm. 2.12].

One can now define the functor $Z_{\mathcal{C}}^{\mathrm{orb}\,\mathbb{A}}$ in (27) as follows:

- For a fixed surface $\Sigma \in \mathrm{Bord}_3$, let us abbreviate $\Psi_\Gamma^{\Gamma'} := \Psi_\Gamma^{\Gamma'}(\Sigma \times [0,1])$. For three $\mathbb{A}$-decorated admissible 1-skeleta $\Gamma, \Gamma', \Gamma''$ one has by definition $\Psi_{\Gamma'}^{\Gamma''} \circ \Psi_\Gamma^{\Gamma'} = \Psi_\Gamma^{\Gamma''}$, i.e. the set $\{\Psi_\Gamma^{\Gamma'}\}_{\Gamma,\Gamma'}$ forms a directed system of linear maps. One takes

$$Z_{\mathcal{C}}^{\mathrm{orb}\,\mathbb{A}}(\Sigma) := \mathrm{colim}_{\Gamma,\Gamma'} \Psi_\Gamma^{\Gamma'}. \tag{30}$$

- For a morphism $M \colon \Sigma \to \Sigma'$ in $\mathrm{Bord}_3$, one takes

$$Z_{\mathcal{C}}^{\mathrm{orb}\,\mathbb{A}}(M) := \left[ Z_{\mathcal{C}}^{\mathrm{orb}\,\mathbb{A}}(\Sigma) \hookrightarrow Z_{\mathcal{C}}^{\mathrm{def}}(\Sigma, \Gamma(\mathbb{A})) \xrightarrow{\Psi_\Gamma^{\Gamma'}(M)} Z_{\mathcal{C}}^{\mathrm{def}}(\Sigma', \Gamma'(\mathbb{A})) \twoheadrightarrow Z_{\mathcal{C}}^{\mathrm{orb}\,\mathbb{A}}(\Sigma') \right], \tag{31}$$

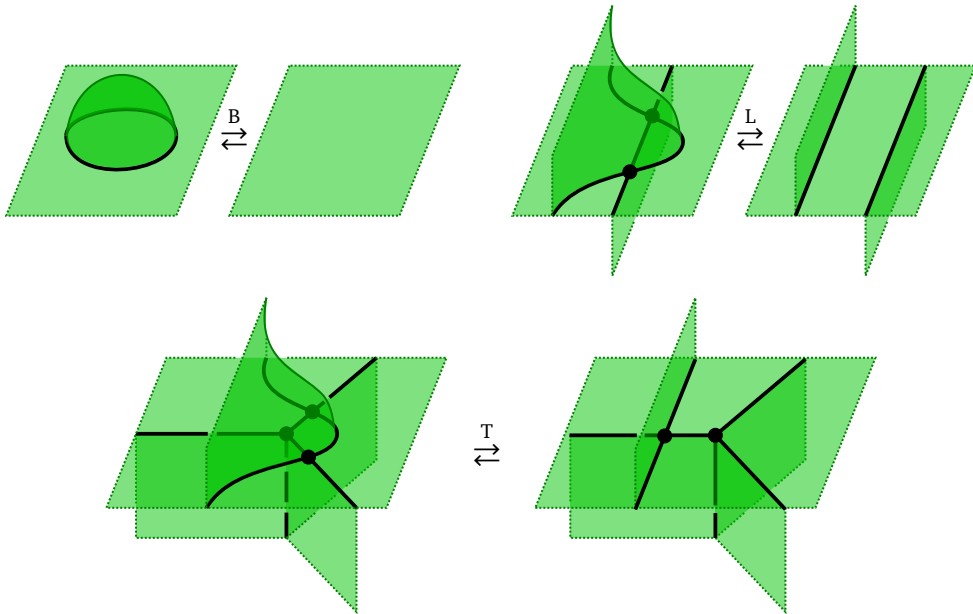

Figure 6: Bubble, lune and triangle (BLT) moves on admissible skeleta.

where the inclusion/projection morphisms in (31) come from the structure maps and the universal property of the colimit in (30).

Let us list some properties and results on the TFT $Z_{\mathcal{C}}^{\mathrm{orb}\,\mathbb{A}}$.

1. Given a surface $\Sigma \in \mathrm{Bord}_3$ and an arbitrary admissible 1-skeleton $\Gamma$ of $\Sigma$, the map $\Psi_{\Gamma}^{\Gamma}$ is an idempotent. An explicit choice for the colimit in (30) is given by its image so that one has

$$Z_{\mathcal{C}}^{\mathrm{orb}\,\mathbb{A}}(\Sigma) \cong \mathrm{im}\,\Psi_{\Gamma}^{\Gamma} \subseteq Z_{\mathcal{C}}^{\mathrm{def}}(\Sigma, \Gamma(\mathbb{A})). \tag{32}$$

2. It was shown in [45] that given an orbifold datum $\mathbb{A}$ in a MFC $\mathcal{C}$ one can construct a ribbon (multi)fusion category $\mathcal{C}_{\mathbb{A}}$. In [16,17] it was shown how $Z_{\mathcal{C}}^{\mathrm{orb}\,\mathbb{A}}$ can be generalised to a TFT $\mathrm{Bord}_3^{\mathrm{rib}}(\mathcal{C}_{\mathbb{A}}) \to \mathrm{Vect}_{\Bbbk}$, i.e. having the source category of bordisms with embedded $\mathcal{C}_{\mathbb{A}}$-coloured ribbon graphs.

3. We call $\mathbb{A}$ *simple* if the aforementioned category $\mathcal{C}_{\mathbb{A}}$ is fusion, i.e. its monoidal unit is a simple object. For a simple orbifold datum $\mathbb{A}$ in a MFC $\mathcal{C}$, the category $\mathcal{C}_{\mathbb{A}}$ was shown to be MFC as well (see [45, Thm. 3.17]). Moreover, fixing a square root of the global dimension $\sqrt{\mathrm{Dim}\,\mathcal{C}}$ yields a canonical choice for the square root $\sqrt{\mathrm{Dim}\,\mathcal{C}_{\mathbb{A}}}$. The orbifold TFT is then equivalent to the corresponding RT TFT (see [17, Thm. 4.1]):

$$Z_{\mathcal{C}}^{\mathrm{orb}\,\mathbb{A}} \cong Z_{\mathcal{C}_{\mathbb{A}}}^{\mathrm{RT}}. \tag{33}$$

In practice, $Z_{\mathcal{C}}^{\mathrm{orb}\,\mathbb{A}}(M)$ is obtained by (i) picking an $\mathbb{A}$-decorated admissible 2-skeleton $S(\mathbb{A})$ of $M$, (ii) converting it into a $\mathcal{C}$-coloured ribbon graph as shown in Figure 3 and (iii) then evaluating it with the RT TFT $Z_{\mathcal{C}}^{\mathrm{RT}}$.

# 3 Internal Levin–Wen models

In this section we lay out the details of the main construction of this paper, that of an anyonic lattice system in a topological phase described by a MFC $\mathcal{C}$. In a nutshell, this is done by

picking an orbifold datum $\mathbb{A}$ in $\mathcal{C}$ and taking the Hamiltonian of the system to be the sum of commuting projectors, whose composition is precisely the idempotent (32), projecting on the state-space, assigned by the internal state-sum TFT to the underlying surface containing the lattice, which serves as the degenerate space of ground states of the system.

## 3.1  State space, Hamiltonian, ground state

The usual way to define a quantum-mechanical system is to give a pair $(V, H)$ consisting of a Hilbert space of states $V$ and a Hamiltonian $H \colon V \to V$, i.e. a self-adjoint operator describing the evolution of states. In this paper we work with a simplified picture where $V$ simply a vector space and $H$ a linear map.

**Internal Levin–Wen data**

We construct the *internal Levin–Wen model* from an input $(\mathcal{C}, \mathbb{A}, \Lambda, \gamma, X, \pi, \iota, \Sigma, \Gamma)$, where (in the diagrams below, surfaces are drawn in paper plane orientation, i.e. as shown in Figure 4)

- $\mathcal{C}$ is a modular fusion category;

- $\mathbb{A} = (A, T, \alpha, \overline{\alpha}, \psi, \phi)$ is an orbifold datum in $\mathcal{C}$;

- $(\Lambda, \gamma)$: $\Lambda$ is a left $A$-module and $\gamma \colon \Lambda \to \Lambda$ is an $A$-module morphism such that the trace $\operatorname{tr} \gamma^2$, seen as a morphism $A \to A$ in the category of $A$-$A$-bimodules, is equal to $\psi^2 \colon A \to A$, or in terms of surface diagrams:

$$
\gamma^2 \bigcirc^{\Lambda}_{A} = \bullet\, \psi^2 \quad_A \tag{34}
$$

- $(X, \pi, \iota)$: $X \in \mathcal{C}$ is an object and

$$
\pi \colon X \rightleftarrows \Lambda^* \otimes_A T \otimes_{A \otimes A} (\Lambda \otimes \Lambda) \colon \iota \tag{35}
$$

are projection/inclusion morphisms, i.e. such that $\pi \circ \iota = \mathrm{id}$ and $\iota \circ \pi$ is an idempotent (i.e. $\pi$ and $\iota$ split an idempotent of $X$ projecting onto the relative tensor product $\Lambda^* \otimes_A T \otimes_{A \otimes A} (\Lambda \otimes \Lambda)$); the morphisms $\pi$, $\iota$ and their duals $\iota^*$, $\pi^*$ can be used to label point defects with five adjacent lines labelled by $X$, $T$ and $\Lambda$ as well as three $A$-labelled surfaces in the following configurations:

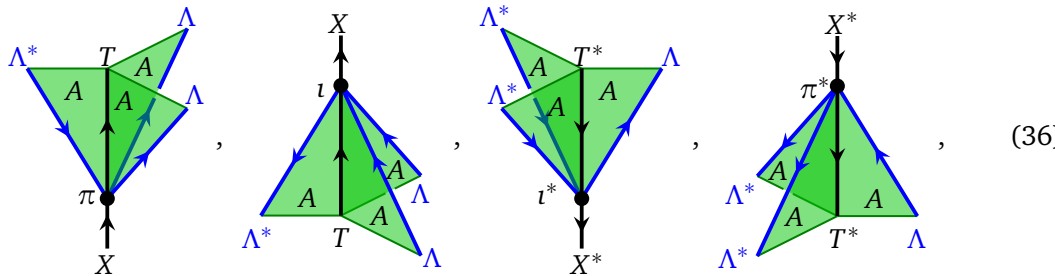

$$\tag{36}$$

so that the idempotent splitting identities become

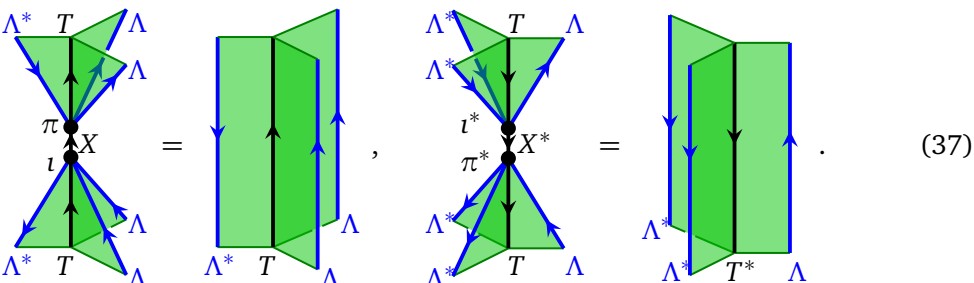

$$(37)$$

- $(\Sigma, \Lambda)$: $\Sigma$ is a compact oriented surface and $\Gamma \subseteq \Sigma$ is an admissible 1-skeleton. An example of an admissible 1-skeleton $\Gamma$ shown in a patch of $\Sigma$ is

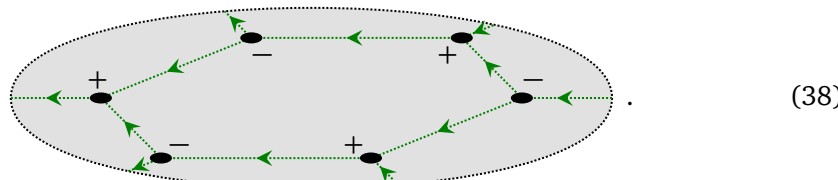

$$(38)$$

We review some examples of the above data in Section 3.2 and discuss the model for those data in more detail in Section 5. For now let us just mention that the entries can be interpreted as follows:

- $\mathcal{C}$ is the input for the RT TFT $Z_{\mathcal{C}}^{\mathrm{RT}}$, which will serve as the ambient theory.[2]

- $\mathbb{A}$ is the main constituent in defining the lattice theory inside $Z_{\mathcal{C}}^{\mathrm{RT}}$ and determines the degenerate ground state space of the model.

- $(\Sigma, \Gamma)$ is the topological datum for the lattice model. Roughly the vertices of $\Gamma$ can be thought of as the places for point excitations (anyons) on a surface $\Sigma$ on which the theory $Z_{\mathcal{C}}^{\mathrm{RT}}$ lives, and the edges of $\Gamma$ indicate their entanglement, favoured by the Hamiltonian.

The other entries are in principle optional as one always has canonical choices for them. Choosing to include them however provides one with a richer set of examples which moreover are easier to compare with existing lattice models in the literature:

- $(\Lambda, \gamma)$ will be used when defining the terms of the Hamiltonian constructed from the edges of $\Gamma$. The orbifold datum $\mathbb{A}$ always provides one with the canonical choice $\Lambda = A$, $\gamma = \psi$.

- $(X, \pi, \iota)$ will be used to define the state space, as well as the terms of the Hamiltonian constructed from the vertices of $\Gamma$. The orbifold datum $\mathbb{A}$ provides one with the canonical choice $X = T$, $\pi = \iota = \mathrm{id}_T$.

**State space**

The state space $V \equiv V_X$ of the model is defined as follows: Let $\Gamma_0$ be the set of vertices (0-strata) of $\Gamma$ and for a vertex $v \in \Gamma_0$ let $|v| = \pm$ denote its orientation. $\Gamma_0$ also provides a stratification of $\Sigma$ by a set of oriented punctures. We denote by $\Gamma_0(X)$ the labelled stratification, where each

---

[2]Technically the algebraic input for $Z_{\mathcal{C}}^{\mathrm{RT}}$ also includes a choice for the square root of the global dimension $\sqrt{\mathrm{Dim}\,\mathcal{C}}$, but we will only use this TFT to evaluate bordisms with the underlying topology of a cylinder $\Sigma \times [0, 1]$, in which case $\sqrt{\mathrm{Dim}\,\mathcal{C}}$ can be ignored.

point $v \in \Gamma_0 \subseteq \Sigma$ has the label $X^{|v|}$ with $X^+ := X$, $X^- := X^*$ as before. The state space of the internal Levin–Wen model is then defined to be

$$V := Z_C^{\mathrm{def}}(\Sigma, \Gamma_0(X)) = Z_C^{\mathrm{RT}}(\Sigma, \Gamma_0(X)), \tag{39}$$

i.e. it is the vector space assigned by the RT TFT to a surface with $X$- or $X^*$-labelled punctures at the vertices of $\Gamma$, cf. (1).

For explicit computations one can use the algebraic expression (14) for the RT TFT state spaces (cf. Section 4 below). For example, assuming $X$ to be self-dual, i.e. $X \cong X^*$, and $\Sigma$ to be the 2-sphere $S^2$ results in $V \cong C(\mathbb{1}, X^{\otimes n})$, where $n = |\Gamma_0|$ is the number of vertices. In other words, the dimension of $V$ is the multiplicity of the trivial anyon $\mathbb{1} \in C$ in the $n$-fold fusion of $X$-anyons. This illustrates the non-locality of the model as one cannot write $V$ as $U^{\otimes n}$ for some vector space of states $U$ assigned to each vertex (unless already $X = \mathbb{1}^{\oplus N}$).

**Hamiltonian**

We now turn to defining the Hamiltonian $H \equiv H_{\mathbb{A}}$ of the system. For each component $c \in \Gamma_0 \sqcup \Gamma_1 \sqcup \Gamma_2$ (i.e. vertex, edge, face) we will define a certain stratified cylinder

$$(\Sigma \times [0,1], S_c) : (\Sigma, \Gamma_0(X)) \to (\Sigma, \Gamma_0(X)). \tag{40}$$

To each such cylinder we can in turn assign a linear map

$$P_c := Z_C^{\mathrm{def}}(\Sigma \times [0,1], S_c) \in \mathrm{End}\, V, \tag{41}$$

to which we will refer as the projector of the component $c$. The total Hamiltonian of the system is then defined to be

$$H = \sum_{c \in \Gamma_0 \sqcup \Gamma_1 \sqcup \Gamma_2} (1 - P_c) = \sum_{v \in \Gamma_0} (1 - P_v) + \sum_{e \in \Gamma_1} (1 - P_e) + \sum_{f \in \Gamma_2} (1 - P_f)$$
$$= H_{\mathrm{V}} + H_{\mathrm{E}} + H_{\mathrm{F}}, \tag{42}$$

where we have grouped the summands into the vertex, edge and face Hamiltonians $H_{\mathrm{V}}$, $H_{\mathrm{E}}$ and $H_{\mathrm{F}}$ respectively. The summands of the identity in (42) are only to ensure that the ground state energy of the system is zero, omitting them does not change the model.

It remains to define the stratifications $S_c$ as in (40) for each component in $\Gamma_0 \sqcup \Gamma_1 \sqcup \Gamma_2$:

- for a vertex $v \in \Gamma_0$, $S_v$ consists of parallel $X$- or $X^*$-labelled lines $\Gamma_0(X) \times [0,1]$ with two additional point insertions on the line $v \times [0,1]$ labelled with $\pi$ and $\iota$ if $|v| = +$ and $\iota^*$ and $\pi^*$ if $|v| = -$, whose neighbourhoods are as in (36), see Figure 7a;

- for an edge $e \in \Gamma_1$ connecting source and target vertices $v_s, v_t \in \Gamma_0$, one constructs $S_e$ by similarly adding $\pi$ and $\iota$ insertions on $v_s \times [0,1]$ and $v_t \times [0,1]$ and then cross-joining the $\Lambda$-lines as well as the $A$-surfaces which lie on the strip $e_i \times [0,1]$, on which one also adds a $\psi^{-2}$-insertion, see Figure 7b;

- for a face $f \in \Gamma_2$ bounding a vertex-edge chain $v_1 \xrightarrow{e_1} v_2 \ldots \xrightarrow{e_{l-1}} v_l = v_1$ the stratification $S_f$ is obtained by first cross-joining the lines $v_i \times [0,1]$ and $v_{i+1} \times [0,1]$ along the edges $e_i$ as it was done for $S_{e_i}$, and then inserting a horizontal $A$-labelled surface with a $\psi^2$-insertion, which joins the previous network of defects at $T$-labelled lines and $\alpha$- or $\overline{\alpha}$-labelled points as shown in Figure 7c.

**Proposition 6.** The maps $P_c$, $P_{c'}$ commute for all components $c, c' \in \Gamma_0 \sqcup \Gamma_1 \sqcup \Gamma_2$. Moreover, if $c' \subseteq c$ then $P_c \circ P_{c'} = P_{c'} \circ P_c = P_c$, in particular all $P_c$ are idempotents.

*Proof.* It is clear that the vertex projectors $P_v$, $v \in \Gamma_0$ commute, as they are defined by the networks of defects as in Figure 7a, which are localised around the strips $v \times [0, 1]$ of $\Sigma \times [0, 1]$ and upon composing $P_{v'} \circ P_v$ can be moved pass each other. That they are idempotents follows from the identities (37).

The commutation of the edge projectors $P_e$, $e \in \Gamma_1$ can again be shown using (37). For example, in the case of two edges sharing a vertex, one first joins the $\Lambda$-lines and the $A$-surfaces and then deforms the resulting network of defects, see Figure 8a, at which point the insertion of the identity (37) allows one to separate them again. If $v \in \Gamma_0$ is either the source or the target vertex of $e$, a similar argument can be used to show that $P_e$ "absorbs" $P_v$, i.e. one has the identity $P_e \circ P_v = P_v \circ P_e = P_e$. The idempotent property of $P_e$ follows from (37) and (34) as illustrated in Figure 8b.

That the face projectors $P_f$, $f \in \Gamma_2$ commute with each other can be shown using the identities (O1)–(O8) defining an orbifold datum, which, as mentioned in Section 2.3, can equivalently be seen as implying the oriented BLT moves on the skeleta, shown in Figure 6. For example, in the case of two faces sharing an edge, part of the computation showing the commutativity is sketched in Figure 8c, where the labels for strata and the $\psi$- and $\phi$-insertions are omitted for brevity. Identities (37) and a similar computation as in Figure 8b show that $P_f$ "absorbs" the projectors of the components $c' \subsetneq f$. Finally, one can show that $P_f$ is an

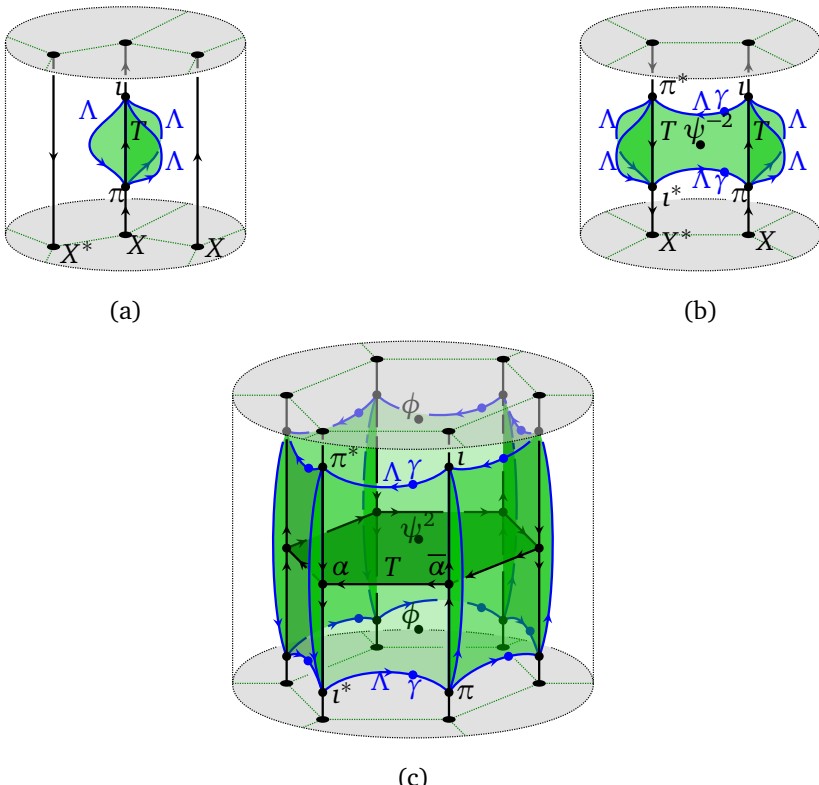

(a)

(b)

(c)

Figure 7: Examples of stratified cylinders $(\Sigma \times [0, 1], S_c)$ for the component $c$ of $\Gamma$ being (a) a vertex (b) an edge (c) a face. Only the relevant patch of $\Sigma \times [0, 1]$ is shown. Evaluating these cylinders with the defect TFT $Z_C^{\text{def}}$ one obtains the projectors $P_c$ used in the Hamiltonian (42). The orientations of surfaces are as in Figure 4, in particular here they coincide with the paper plane orientation.

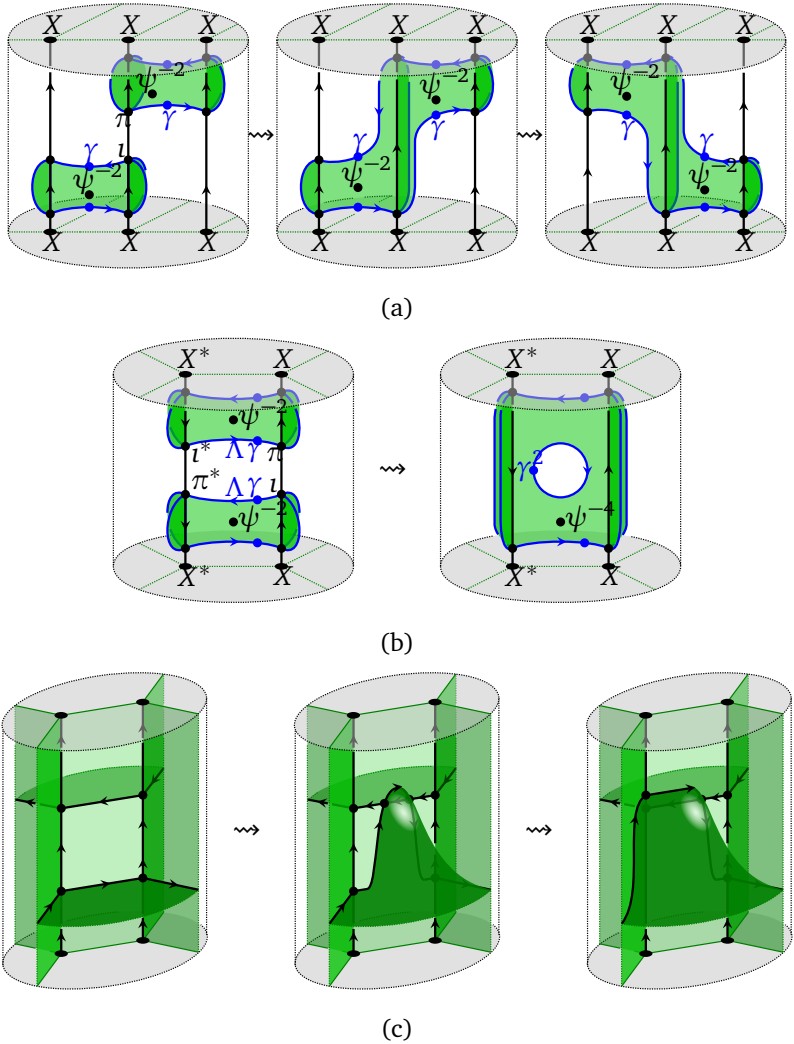

Figure 8: Sketches for computations showing that (a) edge projectors commute; (b) edge projectors are indeed idempotents; (c) face projectors commute.

idempotent as follows: one makes the top horizontal defect "crawl" onto the bottom one using the L (or (O2)–(O7)) and T (or (O1)) moves and in the end removes the newly created bubble defect using the B (or (O8)) move. An instance of this computation is shown in Figure 9. □

**Remark 7.** As explained in Section 2.2, the defect TFT $Z_\mathcal{C}^{\mathrm{def}}$ works by replacing the stratification with a ribbon graph and evaluating with $Z_\mathcal{C}^{\mathrm{RT}}$ (see Figure 3). Let us make the choice $\Lambda = A$, in which case $\gamma = \psi$ and $\Lambda \otimes_A T \otimes_{A \otimes A} (\Lambda \otimes \Lambda) \cong T$. The stratified cylinders in Figures 7b and 7c defining the maps $P_e$ and $P_f$ become the cylinders with ribbon graphs sketched in Figure 10. In Figure 10b, the $\phi$-labelled points become scalar factors upon evaluation with $Z_\mathcal{C}^{\mathrm{RT}}$, and the $\psi_1^2$-insertion uses the notation (26).

**Ground state**

Let $V_0 \subseteq V$ be the subspace of ground states of an internal Levin–Wen model $(\mathcal{C}, \mathbb{A}, \Lambda, \gamma, X, \pi, \iota, \Sigma, \Gamma)$, i.e. the 0-eigenspace of the Hamiltonian (42). At the heart of the construction of the internal Levin–Wen models is the following relation to the internal state-sum TFT $Z_\mathcal{C}^{\mathrm{orb}\,\mathbb{A}}$ from Section 2.3.

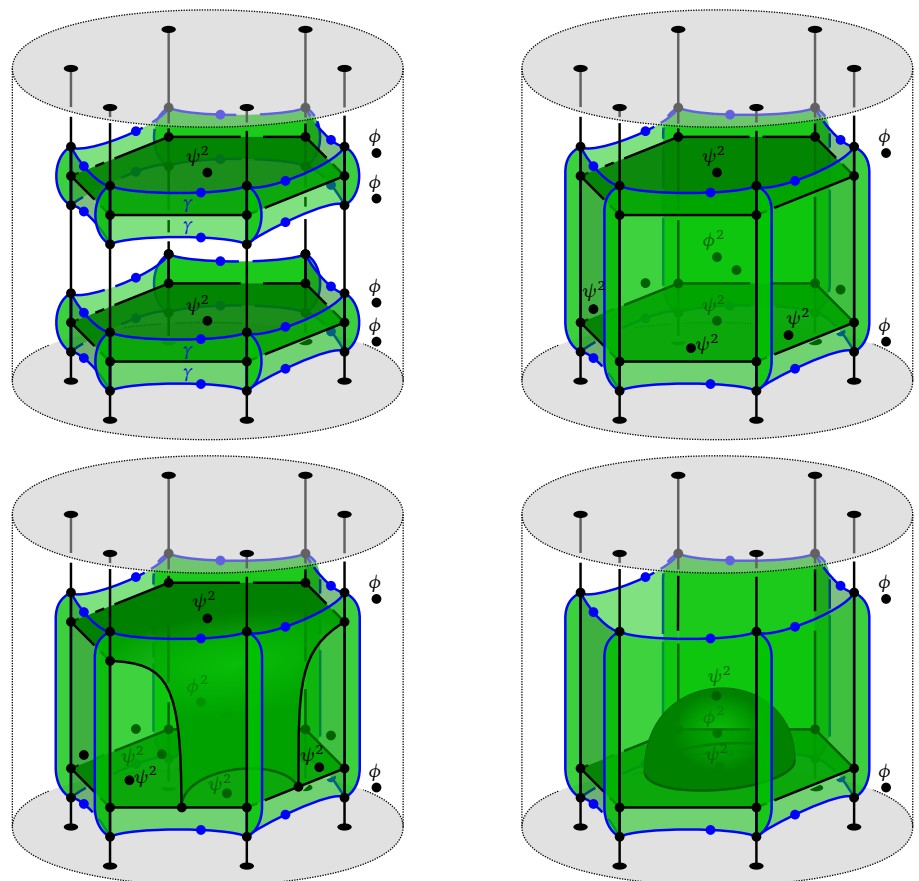

Figure 9: Computation showing that $P_f$ is an idempotent.

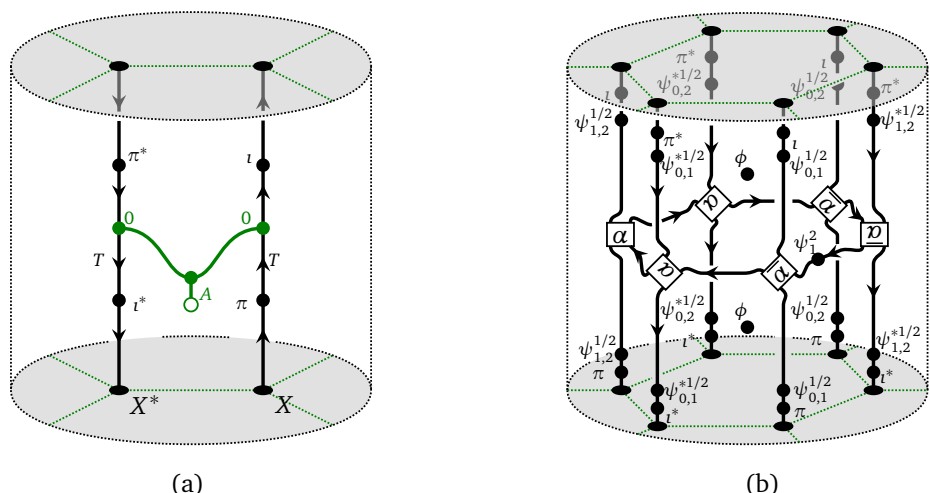

Figure 10: Examples of ribbon graphs replacing the stratifications for edge and face projectors in Figures 7b and 7c under assumptions $\Lambda = A$, $\gamma = \psi$.

**Theorem 8.** One has an isomorphism of vector spaces

$$V_0 \cong Z_{\mathcal{C}}^{\mathrm{orb}\,\mathbb{A}}(\Sigma). \tag{43}$$

*Proof.* It follows from Proposition 6 that the maps $P_c$ for each component $c \in \Gamma_0 \sqcup \Gamma_1 \sqcup \Gamma_2$ are simultaneously diagonalisable and can only have 0 and 1 as eigenvalues. The form (42) of the Hamiltonian then in turn implies that the subspace $V_0 \subseteq V$ is the common eigenspace of all $P_c$'s corresponding to the eigenvalue 1, or equivalently

$$V_0 = \bigcap_{c \in \Gamma_0 \sqcup \Gamma_1 \sqcup \Gamma_2} \mathrm{im}\, P_c = \mathrm{im} \prod_{c \in \Gamma_0 \sqcup \Gamma_1 \sqcup \Gamma_2} P_c = \mathrm{im} \prod_{f \in \Gamma_2} P_f, \tag{44}$$

where in the second equality we used that $P_c$'s are commuting idempotents and in the third that the vertex and the edge projectors are absorbed by the face projectors. In terms of the defect TFT $Z_{\mathcal{C}}^{\mathrm{def}}$, the map on the right-hand side of (44) is obtained by evaluating the cylinder $\Sigma \times [0, 1]$ filled with the networks of defects as in Figure 7c for each face $f \in \Gamma_2$, which upon evaluating can be joined into a connected network of defects by a similar computation as in Figure 8b, which then looks exactly like an admissible $\mathbb{A}$-decorated skeleton of $\Sigma \times [0, 1]$ (cf. Figure 5), except near the boundary components $\{0\} \times \Sigma$, $\{1\} \times \Sigma$, where it has $X$-labelled lines, $\Lambda$-labelled arcs with $\gamma$-insertions and $\pi^{\pm}$-, $\iota^{\pm}$-labelled points. Let us define the maps

$$m : Z_{\mathcal{C}}^{\mathrm{def}}(\Sigma, \Gamma_0(X)) \rightleftarrows Z_{\mathcal{C}}^{\mathrm{def}}(\Sigma, \Gamma(\mathbb{A})) : c, \tag{45}$$

by collecting the latter near-boundary strata and adding a $\psi^{-1}$-insertion to every $A$-labelled surface adjacent to a $\Lambda$-labelled line. Again using a similar calculation as in Figure 8b one then gets the identity $m \circ c = \mathrm{id}$. Furthermore, by functoriality of $Z_{\mathcal{C}}^{\mathrm{def}}$, one obtains the decomposition

$$\prod_{f \in \Gamma_2} P_f = c \circ \Psi_{\Gamma}^{\Gamma} \circ m, \tag{46}$$

where $\Psi_{\Gamma}^{\Gamma}$ is as in (29). Altogether, one has

$$V_0 = \mathrm{im} \prod_{f \in \Gamma_2} P_f \cong \mathrm{im}\, \Psi_{\Gamma}^{\Gamma}, \tag{47}$$

where in the second step we used that $c$ is injective and that $m$ is surjective. The statement of the theorem now follows from (32). □

Recall from Section 2.3 that from the MFC $\mathcal{C}$ and the orbifold datum $\mathbb{A}$ one obtains a new MFC $\mathcal{C}_{\mathbb{A}}$. If we combine the above theorem with the isomorphism of TFTs in (33), we get the isomorphism of vector spaces

$$V_0 \cong Z_{\mathcal{C}_{\mathbb{A}}}^{\mathrm{RT}}(\Sigma). \tag{48}$$

This lends strong support to the claim that the Hamiltonian $H$ and its ground states $V_0$ describe the topological phase $\mathcal{C}_{\mathbb{A}}$. Indeed, we expect $\mathcal{C}_{\mathbb{A}}$ to describe the anyonic excitations of the model. This is true for the examples discussed in Section 5.

By the results of [47], any other MFC which is Witt-equivalent to $\mathcal{C}$ can be obtained as $\mathcal{C}_{\mathbb{A}}$ for an orbifold datum $\mathbb{A}$ in $\mathcal{C}$. In this sense, the internal Levin-Wen models are universal for a given Witt class.

## 3.2 Examples of internal Levin–Wen data

Let us now look at some of the examples of the input datum $(\mathcal{C}, \mathbb{A}, \Lambda, \gamma, X, \pi, \iota, \Sigma, \Gamma)$ defining an internal Levin–Wen model. We keep the surface $\Sigma$ and the graph $\Gamma \subseteq \Sigma$ arbitrary and focus on the algebraic entries needed to define the Hamiltonian (42), most importantly the orbifold datum $\mathbb{A}$ in the MFC $\mathcal{C}$. The Hamiltonian itself will be computed in the more detailed treatment of these examples in Section 5.

### 3.2.1 Condensable algebras

As usual, a *commutative algebra* $A \in \mathcal{C}$ is defined using the braiding of $\mathcal{C}$ via the condition

$$
\begin{array}{ccc}
\text{(diagram)} & = & \text{(diagram)} = \text{(diagram)}
\end{array}
\tag{49}
$$

where either equality implies the other. A commutative Frobenius algebra $A$ is automatically cocommutative (i.e. analogous identities as in (49) hold for the comultiplication). A commutative symmetric $\Delta$-separable Frobenius algebra $A$ in a MFC $\mathcal{C}$ is called *condensable* if it is in addition haploid, i.e. one has $\dim \mathcal{C}(\mathbb{1}, A) = 1$. A condensable algebra $A \in \mathcal{C}$ yields the following simple orbifold datum [22, Prop. 3.4]:

$$
\mathbb{A} = \left( A, \ T = A, \ \alpha = \overline{\alpha} = \ \text{(diagram)} \ , \ \psi = \mathrm{id}_A, \ \phi = 1 \right),
\tag{50}
$$

where the actions (22) are by multiplication with the multimodule identity (23) implied by the commutativity of $A$. We expressed $\alpha, \overline{\alpha}$ as balanced maps and the property (25) again holds by commutativity.

To obtain from (50) an internal Levin–Wen datum, one chooses the remaining entries canonically. The resulting model is summarised in Table 1.

It was shown in [45] that in this case the MFC $\mathcal{C}_{\mathbb{A}}$ is equivalent to the category $\mathcal{C}_A^{\mathrm{loc}}$ of *local modules* of the commutative algebra $A \in \mathcal{C}$. See e.g. [26] for more on local modules and [38] for how $\mathcal{C}_A^{\mathrm{loc}}$ arises from condensing $A$ in the context of topological phases of matter.

### 3.2.2 Internal Levin–Wen data from a spherical fusion category

In this section we will construct internal Levin–Wen data in $\mathcal{C} = \mathrm{Vect}_{\Bbbk}$ from a spherical fusion category $\mathcal{S}$. The orbifold datum $\mathbb{A}_{\mathcal{S}}$ associated to $\mathcal{S}$ is the one from [22, Prop. 4.2], [45, Sec. 4.2] where it was also shown that the generalised orbifold TFT (27) is isomorphic to the Turaev–Viro–Barrett–Westbury-TFT from $\mathcal{S}$ - or alternatively the MFC $\mathcal{C}_{\mathbb{A}_{\mathcal{S}}}$ associated to $\mathbb{A}_{\mathcal{S}}$ is equivalent to the Drinfeld centre $\mathcal{Z}(\mathcal{S})$. We will review this orbifold datum here and extend it to an internal Levin–Wen datum.

Let $I = \mathrm{Irr}_{\mathcal{S}}$ denote a set of representatives of the irreducibles of $\mathcal{S}$ as in (11). We define $T$ and $T^*$ as direct sums of Hom-spaces

$$
T := \oplus_{i,j,k \in I} \mathcal{S}(k, ij), \qquad T^* := \oplus_{i,j,k \in I} \mathcal{S}(ij, k).
\tag{51}
$$

Table 1: Internal Levin-Wen model data from a condensable algebra $A$.

| MFC $\mathcal{C}$ | arbitrary |
|---|---|
| Frob. alg. $A$ | a condensable algebra in $\mathcal{C}$ |
| $T, \alpha, \overline{\alpha}, \psi, \phi$ | as in (50) |
| $\Lambda, \gamma$ | $A$ as left module over itself, $\mathrm{id}_A$ |
| $X, \pi, \iota$ | $A, \mathrm{id}_A, \mathrm{id}_A$ |

The trace pairing establishes $T$ and $T^*$ as dual to each other:

$$\text{ev}\colon T^*\otimes T\to\Bbbk,\quad \widehat{\gamma}\otimes\mu\mapsto\text{tr}_{\mathcal{S}}\left(\widehat{\gamma}\circ\mu\right)=$$

$$\text{coev}\colon\Bbbk\to T\otimes T^*,\quad 1_{\Bbbk}\mapsto\sum_{\lambda}\lambda\otimes\widehat{\lambda}.\tag{52}$$

Here the first expression is only defined for compatible $\mu$ and $\widehat{\gamma}$, i.e. $\mu\in\mathcal{S}(k,i\otimes j)$ and $\widehat{\gamma}\in\mathcal{S}(i\otimes j,k)$, and the second expression is understood as a sum over dual bases. If $\mu$ and $\widehat{\gamma}$ are non-compatible, i.e. if the codomain of $\mu$ does not coincide with the domain of $\widehat{\gamma}$ or vice versa, we define $\text{ev}(\widehat{\gamma}\otimes\mu)=0$.

For elements $\lambda\colon c\to a\otimes b$, $\mu\colon b\to d\otimes e$ in $T$ and $\widehat{\gamma}\colon f\otimes e\to c$, $\widehat{\delta}\colon a\otimes d\to f$ in $T^*$, we define the $F$-symbol and $\overline{F}$ for similarly compatible morphisms $\rho,\tau\in T$ and $\widehat{\theta},\widehat{\eta}\in T^*$

$$F^{\lambda\widehat{\gamma}}_{\mu\widehat{\delta}}=\qquad,\qquad \overline{F}^{\rho\widehat{\theta}}_{\tau\widehat{\eta}}=\qquad.\tag{53}$$

Here $f$ is the codomain of $\widehat{\delta}$ and $c$ is the domain of $\widehat{\eta}$. We extend $F$ and $\overline{F}$ to non-compatible morphisms, $\lambda,\mu,\dots$ by setting it $0$. For compatible basis elements $\lambda,\mu\in T$ and dual basis elements $\widehat{\lambda''},\widehat{\mu''}\in T^*$ we have the following invertibility condition:

$$\sum_{\lambda',\mu'}d_f d_c\overline{F}^{\mu'\widehat{\lambda''}}_{\lambda'\widehat{\mu''}}F^{\lambda\widehat{\mu'}}_{\mu\widehat{\lambda'}}=\begin{cases}1,&\text{if }\lambda''=\lambda,\ \mu''=\mu,\\0,&\text{otherwise.}\end{cases}\tag{54}$$

We can now give the orbifold datum $\mathbb{A}_{\mathcal{S}}$ in $\mathcal{C}=\text{Vect}_{\Bbbk}$ (cf. [22, Prop. 4.3], [45, Sec. 4.2]):

$$A:=\bigoplus_{i\in I}\Bbbk\quad\text{(direct sum of trivial Frobenius algebras)},\tag{55a}$$

$$T:=\bigoplus_{i,j,k\in I}\mathcal{S}(k,ij),\tag{55b}$$

$$\alpha\colon\lambda\otimes\mu\mapsto\sum_{\lambda',\mu'}F^{\lambda\widehat{\mu'}}_{\mu\widehat{\lambda'}}\cdot\mu'\otimes\lambda',\tag{55c}$$

$$\overline{\alpha}\colon\lambda'\otimes\mu'\mapsto\sum_{\lambda'',\mu''},\overline{F}^{\lambda'\widehat{\mu''}}_{\mu'\widehat{\lambda''}}\cdot\mu''\otimes\lambda'',\tag{55d}$$

$$\psi^2:=\text{diag}(d_i)_{i\in I}:=\oplus_{i\in I}d_i\cdot\text{id}_i\qquad(\psi\text{ is a choice of square root}),\tag{55e}$$

$$\phi^2:=\frac{1}{\text{Dim}\,\mathcal{S}}=\left(\sum_{i\in I}d_i^2\right)^{-1}\qquad(\phi\text{ is a choice of square root}).\tag{55f}$$

The $A$-$A\otimes A$ bimodule structure on $T$ is taken to be such that, denoting $1_i\in A$, $i\in I$ the unit of the $i$-th copy of $\Bbbk$ in $A$, the action by $1_k$-$1_i\otimes 1_j$ projects onto $\mathcal{S}(k,ij)\subseteq A$.

We extend this to an internal Levin–Wen datum by taking

$$\Lambda=A\quad\text{as a left-regular module over itself, with}\quad\gamma=\psi,\tag{55g}$$

$$X=\bigoplus_{i,j,k\in I}x,\tag{55h}$$

Table 2: Internal Levin-Wen model data from a spherical fusion category $\mathcal{S}$.

| MFC $\mathcal{C}$ | $\mathrm{Vect}_{\Bbbk}$ |
|---|---|
| Frob. alg. $A$ | $\bigoplus_{i \in I} \Bbbk$ |
| $T$ | $\bigoplus_{i,j,k \in I} \mathcal{S}(k, ij)$ |
| $\alpha, \overline{\alpha}$ | given by $F$-symbols (55c), (55d) |
| $\psi^2$ | $\mathrm{diag}(d_i)_{i \in I}$ |
| $\phi^2$ | $(\mathrm{Dim}\,\mathcal{S})^{-1} = \left(\sum_{i \in I} d_i^2\right)^{-1}$ |
| $\Lambda, \gamma$ | $\Lambda = A$ with left regular action, $\gamma = \psi$ |
| $X$ | $\bigoplus_{i,j,k \in I} x$ |
| $\pi, \iota$ | sums of splittings $\mathrm{id} : \mathcal{S}(k, ij) \overset{\iota_{ijk}}{\to} x \overset{\pi_{ijk}}{\to} \mathcal{S}(k, ij)$ |

where $x = \Bbbk^N$ and $N$ is the maximum multiplicity of simples in all the $i \otimes j$, i.e.

$$N = \max_{i,j,k \in I} \dim, \mathcal{S}(k, ij). \tag{56}$$

We then choose mono- and epimorphisms

$$\mathcal{S}(k, ij) \overset{\iota_{ijk}}{\to} x, \qquad x \overset{\pi_{ijk}}{\to} \mathcal{S}(k, ij), \tag{57}$$

for $i, j, k \in I = \mathrm{Irr}_{\mathcal{S}}$ satisfying $\pi_{ijk} \circ \iota_{ijk} = \mathrm{id}$. Having $\Lambda = A$ we can identify $T = \Lambda \otimes_A T \otimes_{A \otimes A} (\Lambda \otimes \Lambda)$. In turn, this allows us to take the direct sums of the $\iota_{ijk}$ and $\pi_{ijk}$ for $\iota : T \to X, \pi : X \to T$.

An overview of the orbifold data from a spherical fusion category and the additional data for the internal Levin–Wen model is given in Table 2. In Section 5.2 we will show how to obtain the original Levin–Wen model using this data.

### 3.2.3 Internal Levin–Wen data from Hopf algebras

In this section we present an internal Levin–Wen datum constructed from a semisimple, finite-dimensional Hopf algebra $K$ in $\mathrm{Vect}_{\Bbbk}$, where $\Bbbk$ is an algebraically closed field of characteristic 0. In Section 5.3 we illustrate how the internal Levin–Wen model for this datum recovers the Kitaev model based on $K$.

**Prerequisites on Hopf algebras**

We start by reviewing some prerequisites on Hopf algebras. For more details and proofs, consider e.g. [44,49]. For the remainder of this section we denote $K = (K, \mu, \eta, \Delta_{\mathrm{Ho}}, \varepsilon_{\mathrm{Ho}}, S)$ a semisimple, finite-dimensional Hopf algebra over $\mathbb{C}$ with multiplication $\mu$, unit $\eta$, comultiplication $\Delta_{\mathrm{Ho}}$, counit $\varepsilon_{\mathrm{Ho}}$ and antipode $S$. Since below we will also make use of the canonical Frobenius algebra structure on $K$, in this section we write $\Delta_{\mathrm{Ho}}, \varepsilon_{\mathrm{Ho}}$ for the coalgebra structure leading to a Hopf algebra, and $\Delta_{\mathrm{Fr}}, \varepsilon_{\mathrm{Fr}}$ for that of the Frobenius algebra (see below for explicit expressions).

We use Sweedler notation $\Delta_{\mathrm{Ho}}(h) = h_{(1)} \otimes h_{(2)}$ for the comultiplication of $K$ and denote $1_K$ the unit element of $K$. Since the Hopf algebra $K$ is semisimple, it has a unique normalised Haar integral $\lambda \in K$ and cointegral $\int : K \to \Bbbk$. They are defined by the properties

$$h\lambda = \varepsilon_{\mathrm{Ho}}(h)\lambda = \lambda h, \quad \varepsilon_{\mathrm{Ho}}(\lambda) = 1 = \textstyle\int(1_K),$$
$$\textstyle\int(h_{(1)}) \cdot h_{(2)} = \int(h) \cdot 1_K = \int(h_{(2)}) \cdot h_{(1)}, \tag{58}$$

for any $h \in K$. The Haar (co-)integrals fulfil some useful identities, in particular we have for $h, k \in K$:

$$\lambda_{(1)} \otimes \lambda_{(2)} = \lambda_{(2)} \otimes \lambda_{(1)}, \qquad \int(hk) = \int(kh), \tag{59a}$$

$$h\lambda_{(1)} \otimes \lambda_{(2)} = \lambda_{(1)} \otimes S(h)\lambda_{(2)}, \qquad \lambda_{(1)}h \otimes \lambda_{(2)} = \lambda_{(1)} \otimes \lambda_{(2)}S(h), \tag{59b}$$

$$\int \circ S = \int, \quad S(\lambda) = \lambda, \qquad \int(\lambda) = \frac{1}{|K|}, \tag{59c}$$

where we write $|K| := \dim_{\Bbbk} K$. Associated to the Hopf algebra $K$ there is a $\Delta$-separable, symmetric Frobenius algebra structure $A = K_{\mathrm{Fr}} = (K, \mu, \eta, \Delta_{\mathrm{Fr}}, \varepsilon_{\mathrm{Fr}})$ on the vector space $K$ with the same multiplication $\mu$ and unit $\eta$ as the Hopf algebra $K$. The comultiplication and the counit are given by

$$\Delta_{\mathrm{Fr}} \colon K \to K \otimes K, \qquad \varepsilon_{\mathrm{Fr}} \colon K \to \Bbbk,$$
$$h \mapsto h\lambda_{(1)} \otimes S(\lambda_{(2)}), \qquad h \mapsto |K| \cdot \int(h). \tag{60}$$

The fact that $\lambda$ is normalised (i.e. $\varepsilon_{\mathrm{Ho}}(\lambda) = 1$) ensures that $K_{\mathrm{Fr}}$ is $\Delta$-separable and symmetry follows from the cyclic invariance of $\int$ in (59a). This is the Frobenius algebra $A$ used for the orbifold datum (21) defined in this section.

**Orbifold data from Hopf algebras**

For the $(A, A \otimes A)$-bimodule $T$ of the orbifold datum we take the vector space $T = K \otimes K$ with the $K$-actions:

$$\rhd_0 \colon A \otimes T \to T, \quad h \rhd_0 [m \otimes n] = h_{(2)}m \otimes h_{(1)}n,$$
$$\lhd_1 \colon T \otimes A \to T, \quad [m \otimes n] \lhd_1 h = mh \otimes n,$$
$$\lhd_2 \colon T \otimes A \to T, \quad [m \otimes n] \lhd_2 h = m \otimes nh. \tag{61}$$

Here and in the following we distinguish the two copies of $K$ in $T = K \otimes K$ in a larger tensor product by putting them in square brackets.

We consider the relative tensor products $T \otimes_1 T, T \otimes_2 T$ as subspaces of $T \otimes T = K^{\otimes 4}$ by identifying them with the images of the corresponding idempotents (18). Explicitly, these images are given by

$$T \otimes_1 T = \left\{ \left[ aS(\lambda_{(1)}) \otimes b \right] \otimes \left[ \lambda_{(3)}c \otimes \lambda_{(2)}d \right] \mid a, b, c, d \in K \right\},$$
$$T \otimes_2 T = \left\{ \left[ a \otimes bS(\lambda_{(1)}) \right] \otimes \left[ \lambda_{(3)}c \otimes \lambda_{(2)}d \right] \mid a, b, c, d \in K \right\}. \tag{62}$$

We now define the maps $\alpha, \overline{\alpha}$ as balanced maps:

$$\alpha \colon T \otimes T \to T \otimes_1 T \subseteq T \otimes T,$$
$$[a \otimes b] \otimes [c \otimes d] \mapsto \left[ S(\lambda_{(1)}) \otimes b_{(1)}d \right] \otimes \left[ \lambda_{(3)}a \otimes \lambda_{(2)}b_{(2)}c \right],$$
$$\overline{\alpha} \colon T \otimes T \to T \otimes_2 T \subseteq T \otimes T,$$
$$[a \otimes b] \otimes [c \otimes d] \mapsto \left[ a_{(2)}c \otimes S(\lambda_{(1)}) \right] \otimes \left[ \lambda_{(3)}a_{(1)}d \otimes \lambda_{(2)}b \right]. \tag{63}$$

The remaining parts of the orbifold data are defined by $\psi = \mathrm{id}_K$ and $\phi = \frac{1}{\sqrt{\dim K}}$. The proof of the following proposition is given in Appendix B.

Table 3: Internal Levin-Wen model data from a semisimple Hopf algebra $K$.

| MFC $\mathcal{C}$ | $\mathrm{Vect}_{\Bbbk}$ |
|---|---|
| Frob. alg. $A$ | $K_{\mathrm{Fr}}$ |
| $T$ | $K^{\otimes 2}$ with the $K$-actions (61) |
| $\alpha, \overline{\alpha}$ | (63) |
| $\psi$ | $\mathrm{id}_K$ |
| $\phi$ | $\frac{1}{\sqrt{\dim K}}$ |
| $\Lambda, \gamma$ | $\Lambda = K$ with left regular $K$-action, $\gamma = \mathrm{id}_K$ |
| $X, X^*, \pi, \iota$ | $K^{\otimes 3}$ (64) |

**Proposition 9.** The tuple $(A = K_{\mathrm{Fr}}, T, \alpha, \overline{\alpha}, \psi, \phi)$ is an orbifold datum.

We now extend this orbifold datum to an internal Levin–Wen datum by providing the remaining entries $(\Lambda, \gamma, X, \pi, \iota)$. For the left $A$-module $\Lambda$ we simply take $\Lambda = K$ with the left regular $K$-action on itself. This allows us to identify $T = \Lambda^* \otimes_A T \otimes_{A \otimes A} (\Lambda \otimes \Lambda)$. We also take $\gamma = \mathrm{id}_K$. Finally, we choose $X = K \otimes K \otimes K$ together with the maps:

$$\pi: X \to T, \qquad\qquad \iota: T \to X,$$
$$a \otimes b \otimes c \mapsto a_{(2)} b \otimes a_{(1)} c, \qquad\qquad b \otimes c \mapsto S(\lambda_{(1)}) \otimes \lambda_{(3)} b \otimes \lambda_{(2)} c. \qquad (64)$$

A direct computation using the normalisation of the Haar integral shows that $\pi \circ \iota = \mathrm{id}_T$.

We define the dual objects $K^*, T^*, X^*$ using the (co-)integrals of $K$, instead of just taking the dual vector space - this allows us to relate them to the Kitaev model more easily. In particular, we set $K^* = K$ as vector space, together with the (co-)evaluation maps given by the Frobenius (co)pairing on $K$:

$$\mathrm{coev}: \Bbbk \to K \otimes K^*, \qquad\qquad \mathrm{ev}: K^* \otimes K \to \Bbbk,$$
$$1 \mapsto \Delta_{\mathrm{Fr}}(1_K), \qquad\qquad h \otimes k \mapsto \varepsilon_{\mathrm{Fr}}(hk). \qquad (65)$$

Similarly we set $T^* = K \otimes K$ and $X^* = K \otimes K \otimes K$, with (co-)evaluation maps induced by those of $K$. The dual $K$-actions on $T^*$ from (24) can be shown to be

$$\lhd_0: T^* \otimes K \to T^*, \qquad [m \otimes n] \lhd_0 h = m h_{(2)} \otimes n h_{(1)},$$
$$\rhd_1: T^* \otimes K \to T^*, \qquad h \rhd_1 [m \otimes n] = hm \otimes n,$$
$$\rhd_2: T^* \otimes K \to T^*, \qquad h \rhd_2 [m \otimes n] = m \otimes hn. \qquad (66)$$

The maps dual to $\pi$ and $\iota$ are

$$\iota^*: X^* \to T^*, \qquad\qquad \pi^*: T^* \to X^*,$$
$$a \otimes b \otimes c \mapsto b a_{(2)} \otimes c a_{(1)}, \qquad\qquad b \otimes c \mapsto \lambda_{(1)} \otimes b S(\lambda_{(2)}) \otimes c S(\lambda_{(3)}). \qquad (67)$$

An overview of the orbifold datum from a finite-dimensional semisimple Hopf algebra and the additional datum for the internal Levin–Wen model is given in Table 3.

**Remark 10.** The MFC $\mathcal{C}_{\mathbb{A}}$ for the orbifold datum $\mathbb{A}$ defined in this section is equivalent to the category of $K$-Yetter-Drinfeld modules or the category of $D(K)$-modules for the Drinfeld double $D(K)$. We do not show this here, but only give a quick sketch: (i) The category $\mathcal{C}_{\mathbb{A}}$ as defined in [45] can be shown to be the category of so called $K$-Hopf bimodules - the two $K$-coactions can be derived from the maps $\tau_1$ and $\tau_2$ from [45]; (ii) The category of $K$-Hopf bimodules was shown to be monoidally equivalent to the category of $K$-Yetter–Drinfeld modules in [52].

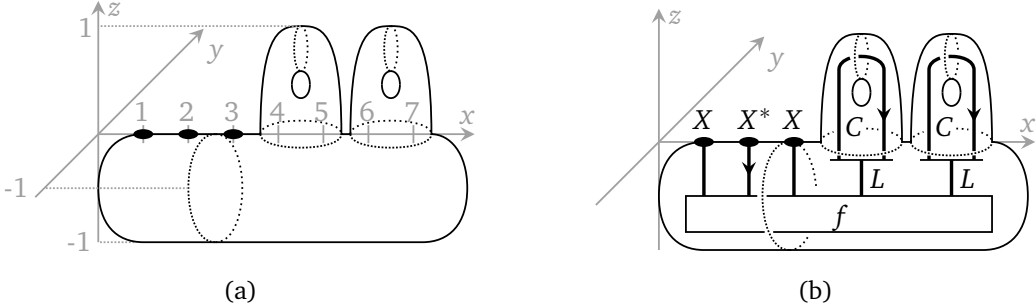

(a)        (b)

Figure 11: (a) The standard surface $\Sigma_{g,n} \subseteq \mathbb{R}^3$ with the genus $g = 2$ and the number of punctures $n = 3$, defined as the boundary of the standard handlebody in $\mathbb{R}^3$. (b) The standard handlebody $M_{g,n}(f)$ with embedded ribbon graph and coupon labelled by $f \in \mathcal{C}(\mathbb{1}, XX^*XL^{\otimes 2})$.

# 4 Explicit computations

In the previous section we defined the internal Levin–Wen model in terms of the defect TFT $Z_\mathcal{C}^{\mathrm{def}}$. This definition allowed us to use the graphical calculus of surface diagrams, which made it easy to demonstrate some of the properties of the model (e.g. that the face projectors $P_f$ are idempotents, see Figure 9). Any explicit computation however requires unpacking both the definition of $Z_\mathcal{C}^{\mathrm{def}}$, and that of the TFT with embedded ribbon graphs $Z_\mathcal{C}^{\mathrm{RT}}$. Even though this is a standard procedure as summarised in Section 2.2, it is illustrative to explain it in the context of our lattice model. In this section we describe how to compute the projectors in general (Section 4.1) as well as work out the concrete linear maps for three examples of edges and an example of a hexagonal face on a 2-torus (Section 4.2).

## 4.1 Projector maps explicitly

Let $(\mathcal{C}, \mathbb{A}, \Lambda, \gamma, X, \pi, \iota, \Sigma, \Gamma)$ be an internal Levin–Wen datum. Throughout this section we use the simplifying assumptions $\Lambda = A$, so that one can take $\gamma = \psi$ and $\Lambda \otimes_A T \otimes_{A \otimes A} (\Lambda \otimes \Lambda) \cong T$ as in Remark 7.

**The state space as a Hom-space**

Here we construct an isomorphism $\Phi$ between the state space $V = Z_\mathcal{C}^{\mathrm{RT}}(\Sigma, \Gamma_0(X))$ from (39) and a Hom-space of $\mathcal{C}$ of the form (14). Let $\Sigma_{g,n}$ be the standard genus $g$ surface with $n$-punctures, defined as a subspace of $\mathbb{R}^3$ as in Figure 11a. The isomorphism $\Phi$ is determined by a choice of a homeomorphism

$$\varphi: \Sigma_{g,n} \longrightarrow (\Sigma, \Gamma_0(X)). \tag{68}$$

Note that this automatically fixes an order on the set of vertices: $\Gamma_0 = \{v_1, \ldots, v_n\}$.

Recall from (13) that $L = \bigoplus_{i \in \mathrm{Irr}_\mathcal{C}} i \otimes i^* \in \mathcal{C}$. In what follows it will be useful to introduce the object $C = \bigoplus_{i \in \mathrm{Irr}_\mathcal{C}} i$, so that $L$ is a subobject of $C \otimes C^*$ with the projection/inclusion maps $L \rightleftarrows C \otimes C^*$ denoted by

$$\tag{69}$$

$L$          $C \quad C^*$

,          .

$C \quad C^*$          $L$

Consider the handlebody $M_{g,n}(f)$ in Figure 11b. The coupon is labelled by a morphism $f \in \mathcal{C}(\mathbb{1}, X^{|v_1|} \cdots X^{|v_n|} L^{\otimes g})$. As a bordism in $\mathrm{Bord}_3^{\mathrm{rib}}(\mathcal{C})$, we have $M_{g,n}(f) \colon \varnothing \to \Sigma_{g,n}$. Denote by $C_\varphi \colon \Sigma_{g,n} \to (\Sigma, \Gamma_0(X))$ the mapping cylinder for $\varphi$. The isomorphism

$$\Phi \colon \mathcal{C}(\mathbb{1}, X^{|v_1|} \cdots X^{|v_n|} L^{\otimes g}) \longrightarrow Z_{\mathcal{C}}^{\mathrm{RT}}(\Sigma, \Gamma_0(X)) = V\,, \tag{70}$$

is given by

$$\Phi(f) = \left[ \mathcal{C}(\mathbb{1}, X^{|v_1|} \cdots X^{|v_n|} L^{\otimes g}) \xrightarrow{Z_{\mathcal{C}}^{\mathrm{RT}}(M_{g,n}(f))} Z_{\mathcal{C}}^{\mathrm{RT}}(\Sigma_{g,n}) \xrightarrow{Z_{\mathcal{C}}^{\mathrm{RT}}(C_\varphi)} Z_{\mathcal{C}}^{\mathrm{RT}}(\Sigma, \Gamma_0(X)) \right]. \tag{71}$$

That this is indeed an isomorphism is one of the properties of RT TFT as discussed in Section 2.1. By abuse of notation we will use $\Phi$ to identify $V$ with the Hom-space which is the domain of $\Phi$. We will comment on the irrelevance of the choice of $\varphi$ in Remark 12 below.

The remainder of this section is devoted to describing the projectors $P_c \colon V \to V$ in (41) by expressing them as linear maps of the form (15), i.e. as post-composition with an appropriate endomorphism,

$$P_c(f) = \Omega_c \circ f\,, \quad \text{for some} \quad \Omega_c \in \mathrm{End}_{\mathcal{C}}(X^{|v_1|} \cdots X^{|v_n|} L^{\otimes g})\,. \tag{72}$$

For a morphism $h \colon Q \to R$ let us write $h^+ := h$ and $h^- := h^* \colon R^* \to Q^*$, the dual morphism. For a vertex $v_i \in \Gamma_0$ the endomorphism $\Omega_{v_i}$ giving rise to the vertex projector is particularly simple:

$$\Omega_{v_i} = \mathrm{id}_{X^{|v_1|}} \otimes \cdots \otimes (\imath \circ \pi)^{|v_i|} \otimes \cdots \otimes \mathrm{id}_{X^{|v_n|}} \otimes \mathrm{id}_{L^{\otimes g}}\,. \tag{73}$$

Edge and face projectors will turn out to be more involved and will need extra preparation.

**The half-braiding of $L$**

The object $L \in \mathcal{C}$ has natural half-braiding morphisms $\gamma = \{\gamma_X \colon X \otimes L \xrightarrow{\sim} L \otimes X\}_{X \in \mathcal{C}}$, defined by

$$\bigoplus_{i,j \in \mathrm{Irr}_{\mathcal{C}}} \sum_q \quad \boxed{\begin{array}{c} j \quad j^* \qquad X \\ b_q^{Xij} \quad \overline{b_q^{Xij}}^* \\ X \qquad i \quad i^* \end{array}} \quad = \quad \sum_p \quad \boxed{\begin{array}{c} L \qquad X \\ C \quad C \\ b_p^X \quad \overline{b_p^X}^* \\ X \qquad L \end{array}} \quad , \tag{74}$$

where $\{b_q^{Xij}\}$ forms a basis of $\mathcal{C}(Xi, j)$ and $\{\overline{b_q^{Xij}}\}$ is its dual basis of $\mathcal{C}(j, Xi)$ with respect to the composition pairing, i.e. one has

$$b_{q'}^{Xij} \circ \overline{b_q^{Xij}} = \delta_{qq'} \cdot \mathrm{id}_j\,, \quad \sum_{j,q} \overline{b_q^{Xij}} \circ b_{q'}^{Xij} = \mathrm{id}_{Xi}\,, \tag{75}$$

and on the right-hand side of (74) we have collected the index triple $(i, j, q)$ into a single index $p$ and introduced the morphisms

$$b_p^X = [XC \twoheadrightarrow Xi \xrightarrow{b_q^{Xij}} j \hookrightarrow C]\,, \quad \overline{b_p^X} = [C \twoheadrightarrow j \xrightarrow{\overline{b_q^{Xij}}} Xi \hookrightarrow XC]\,. \tag{76}$$

This makes the pair $(L, \gamma)$ an object of the Drinfeld centre $\mathcal{Z}(\mathcal{C})$. See e.g. [10, Sec. 9] and [5, Thm. 2.3] for more details on this half-braiding.

**Transporting a puncture along a path**

To proceed further we will find it convenient to have an explicit description of the action of some mapping class groupoid elements on the vector space (14). In particular, we look at how to permute the punctures by moving one of them along a path.

To this end, let $\gamma \colon [0,1] \to \Sigma_{g,n}$ be a simple path between two punctures, i.e. such that $\gamma(0) = v_j$, $\gamma(1) = v_k$ and $\gamma((0,1))$ does not contain any punctures and has no self-intersections. Furthermore, we assume the tangent vectors $\dot{\gamma}(0)$, $\dot{\gamma}(1)$ of $\gamma$ at points $v_j$ and $v_k$ not to lie on the $y$-axis in the conventions of Figure 11a, i.e. one can unambiguously say whether $\gamma$ leaves $v_j$ to the left or right, and whether it approaches $v_k$ from the left or right. This is to account for the framing dependence of the punctures when moving them along a path (cf. Figure 12a). Define a permutation $\tau_\gamma$ on $[n] = \{1, 2, \ldots, n\}$ as follows: set $\epsilon = +1$ if $\dot{\gamma}_x(1) < 0$ and $-1$ if $\dot{\gamma}_x(1) > 0$; take the map $\widetilde{\tau} \colon [n] \to \frac{1}{2}\mathbb{Z}$ to be $\widetilde{\tau}(j) = k + \epsilon\frac{1}{2}$ and id on $[n] \setminus \{j\}$; then $\tau$ is the composition of $\widetilde{\tau}$ with the unique order preserving map $\operatorname{im} \widetilde{\tau} \to [n]$ (for example, the permutation for the path in Figure 12a sends $(1,2,3)$ to $(2,1,3)$). The aforementioned element of the mapping class groupoid then has the action whose form is

$$
\mathcal{C}(\mathbb{1}, X_1^{|v_1|} \cdots X_n^{|v_n|} L^{\otimes g}) \quad \to \quad \mathcal{C}(\mathbb{1}, X_{\tau_\gamma^{-1}1}^{|v_{\tau_\gamma^{-1}1}|} \cdots X_{\tau_\gamma^{-1}n}^{|v_{\tau_\gamma^{-1}n}|} L^{\otimes g}),
$$
$$
f \quad \mapsto \quad \sigma_\gamma(X_1^{|v_1|}, \ldots, X_n^{|v_n|}) \circ f,
\tag{77}
$$

where $\sigma_\gamma$ is a certain natural transformation between two functors $\mathcal{C}^{\boxtimes n} \to \mathcal{C}$

$$
\sigma_\gamma \colon ((-)^{\otimes n} \otimes L^{\otimes g}) \Rightarrow ((-)^{\otimes n} \circ \tau_\gamma) \otimes L^{\otimes g}),
\tag{78}
$$

with $(-)^{\otimes n} \colon \mathcal{C}^{\boxtimes n} \to \mathcal{C}$ denoting the $n$-fold tensor product functor and $\tau_\gamma$ being interpreted as the permutation endofunctor on $\mathcal{C}^{\boxtimes n}$.

The maps $\sigma_\gamma$ can be obtained as follows: Take the puncture $v_j \in \Gamma_0$ on the standard handlebody as in Figure 11b and move it along the path $\gamma$ while shifting the other punctures so that at the end of the move they are in the standard positions. The strand connected to $v_j$ will then braid with the other strands connected to the coupon $f$, as well as with the $C$-labelled strands at the cores of the handles. This braiding pattern is what constitutes the map $\sigma_\gamma$ and, depending on $\gamma$, can come as a concatenation of simple paths of three types:

- $\gamma$ does not wrap around a handle. Assuming that $\gamma$ passes neither through nor over the arc of a handle, all $\sigma_\gamma$ does is braid the strand at the end of the puncture $v_j$ with the other punctures and possibly the $L$-labelled strands assigned to the handles as in Figure 11b. Note that because of the framing the strand of $v_j$ might acquire twists. An example is shown in Figure 12a.

- $\gamma$ wraps around the meridian of a handle. If $\gamma$ passes through the arc of a handle, the $L$-labelled strand is split into a pair of $C$- and $C^*$-labelled strands using the inclusion morphism $L \to CC^*$ (see (69)), which then braid with the strand of $v_j$ and get projected back onto $L$ as in Figure 12b.

- $\gamma$ wraps around the parallel of a handle. If $\gamma$ passes over the arc of a handle, $\sigma_\gamma$ incorporates the halfbraiding (74) of the object $L \in \mathcal{C}$. This is because after moving $v_j$ along $\gamma$, the strand at the end of it will have a segment which is parallel to one of the $C$-labelled arcs as in Figure 11b, at which point one can utilise (76) and (75) as illustrated in Figure 12c.

For later convenience, we also introduce what could be called the "$\gamma$-twisted associator" natural transformation between two functors $\mathcal{C}^{\boxtimes(n+1)} \to \mathcal{C}$

$$
a_\gamma \colon ((-)^{\otimes(n+1)} \otimes L^{\otimes g}) \Rightarrow ((-)^{\otimes n} \circ \tau_\gamma) \otimes L^{\otimes g}) \circ \otimes_{j,j+1},
\tag{79}
$$

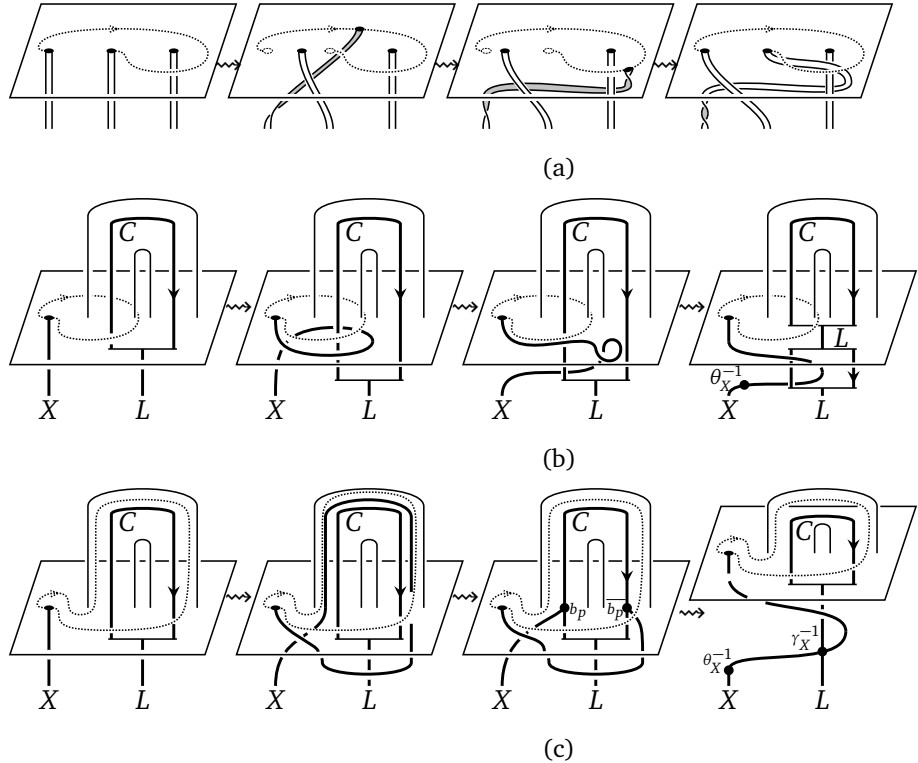

(a)

(b)

(c)

Figure 12: Three types of paths to transport punctures on the standard surface. (a) first type braids the punctures away from handles; (b) second type transports a puncture through a handle; (c) third type transports a puncture over a handle (here the sum over the (multi-)index $p$ is implied, see (74)–(76)). The ribbon graph embedded into the standard handlebody is modified as shown. A generic path can be composed from the paths of these three types.

where $\otimes_{j,j+1} \colon \mathcal{C}^{\boxtimes(n+1)} \to \mathcal{C}^{\boxtimes n}$ is the functor taking the tensor product of $j$ and $(j+1)$ $\boxtimes$-factors, as follows:

$$
\begin{aligned}
a_\gamma(X_1, &\ldots, Y, X_j, \ldots, X_n) \\
&:= \sigma_\gamma^{-1}(X_1, \ldots, X_j, \ldots, (X_k Y), \ldots X_n) \circ \sigma_\gamma(X_1, \ldots, (Y X_j), \ldots, X_k, \ldots X_n).
\end{aligned}
\tag{80}
$$

Precomposing with $a_\gamma$ has the effect of i) unfusing the $(YX_j)$-labelled puncture into separate $Y$- and $X_j$-labelled punctures next to each other; ii) transporting $Y$ away from $X_j$ and next to $X_k$ along the path $\gamma$, while leaving $X_j$ and $X_k$ in their positions; iii) fusing $Y$ and $X_k$ into a single $(X_k Y)$-labelled puncture:

$$
\tag{81}
$$

When using $a_\gamma$ we will usually omit all the arguments except for that of the object, labelling the puncture which is being transported, assuming that the others are clear from the context, for example:

$$
a_\gamma(Y) = a_\gamma(X_1, \ldots, Y, X_j, \ldots, X_n), \quad a_\gamma^{-1}(Y) = a_\gamma^{-1}(X_1, \ldots, Y, X_j, \ldots, X_n).
\tag{82}
$$

**Remark 11.** As we will see below, the natural transformations $a_\gamma$ do appear in the explicit expression of the Hamiltonian of the internal Levin–Wen model introduced in Section 3. The use of $a_\gamma$ reflects the fact that the ambient topological phase $\mathcal{C}$ is already a non-local anyonic model, as it involves the braiding of the MFC $\mathcal{C}$. In the special case of the trivial MFC $\mathcal{C} = \text{Vect}_\Bbbk$, the maps $\sigma_\gamma$ and $a_\gamma$ depend only on the permutation $\tau_\gamma$ and not explicitly on the path $\gamma$, i.e. all they do is permute the tensor factors.

**Edge projectors**

Let $e$ be an edge of $\Gamma$ with the source and the target vertices $v_j$ and $v_k$, respectively. Recall from Remark 7 that in terms of the TFT $Z_\mathcal{C}^{\text{RT}}$, the edge projector $P_e$ is obtained by evaluating a cylinder with a ribbon graph similar to the one in Figure 10a. As explained in the beginning of the section, one obtains the map $P_e$ in the form (72) by taking the surface $\Sigma$ to be the standard surface as in Figure 11a. This puts the endpoints of the $X^\pm$-labelled strands in the standard positions and the horizontal $A$-labelled strand within the strip $e \times [0,1]$ of the cylinder $\Sigma \times [0,1]$. We assume that on the standard surface each vertex of $\Gamma$ has one of the two possible neighbourhoods

$$
\begin{array}{cc}
\includegraphics{placeholder} & 
\end{array}
$$


,                (83)

in particular so that each edge starts 'to the left' of a vertex and ends 'from the right', with the 0-labelled edge starting/ending on the $x$-axis. The indexing of the half-edges in (83) is introduced as a convention to be the same as that of $A$-actions $\triangleright_0$, $\triangleleft_1$, $\triangleleft_2$ on $T$ and $\triangleleft_0$, $\triangleright_1$, $\triangleright_2$ on $T^*$ (see (22), (24)). We set $s(e), t(e) \in \{0,1,2\}$ to be the indices of the edge $e$ at the source and the target respectively (note that $s(e) = 0$ is only possible if $|v_j| = +$ and $t(e) = 0$ is only possible if $|v_k| = -$).

One defines the morphism $\Omega_e$ in (72) as follows:

$$\Omega_e = (\text{id}_{X^{|v_1|}} \otimes \cdots \otimes \iota^{|v_k|} \otimes \cdots \otimes \iota^{|v_j|} \otimes \cdots \otimes \text{id}_{X^{|v_n|}}) \tag{84a}$$

$$\circ (\text{id}_{X^{|v_1|}} \otimes \cdots \otimes \triangleleft_{t(e)} \circ \otimes \cdots \otimes \text{id}_{T^{|v_j|}} \otimes \cdots \otimes \text{id}_{X^{|v_n|}}) \tag{84b}$$

$$\circ\, a_e(X^{|v_1|}, \ldots, T^{|v_k|}, \ldots, A, T^{|v_j|}, \ldots, X^{|v_n|}) \tag{84c}$$

$$\circ (\text{id}_{X^{|v_1|}} \otimes \cdots \otimes \text{id}_{T^{|v_k|}} \otimes \cdots \otimes \triangleright_{s(e)} \circ (\Delta \circ \eta \otimes \text{id}_{T^{|v_j|}}) \otimes \cdots \otimes \text{id}_{X^{|v_n|}}) \tag{84d}$$

$$\circ (\text{id}_{X^{|v_1|}} \otimes \cdots \otimes \pi^{|v_k|} \otimes \cdots \otimes \pi^{|v_j|} \otimes \cdots \otimes \text{id}_{X^{|v_n|}}). \tag{84e}$$

Figure 14 below contains three concrete examples of this expression: one for each type of path in Figure 12 for $a_e(A)$ in step (84c).

**Face projectors**

Let $f$ be a face of $\Gamma$. Analogously to the edge projectors, to obtain $P_f$ in the form (72) one replaces the stratification of the cylinder as in Figure 7c by a ribbon graph as in Figure 10b and takes $\Sigma$ to be the standard surface with the endpoints of $X^\pm$-labelled strands at the standard positions. The vertex-edge chain bounding $f$ can then be taken to be

$$v_{f(1)} \xrightarrow{e_1} v_{f(2)} \xrightarrow{e_2} \cdots \xrightarrow{e_l} v_{f(l+1)} = v_{f(1)}. \tag{85}$$

Here $f(1), \ldots, f(l) \in \{1, \ldots, n\}$ are the vertex indices and $e_1, \ldots, e_l$ denote the edges of $\Gamma$ between them with the convention that:

i) $f(1)$ is maximal out of $f(1), \ldots, f(l)$,

ii) the orientation $v_{f(i)} \xrightarrow{e_i} v_{f(i+1)}$ is the opposite to the orientation of this edge as induced by the orientation of the face $f$.

Note that the orientation of the edge $e_i$ in $\Gamma$ may or may not coincide with the orientation $v_{f(i)} \xrightarrow{e_i} v_{f(i+1)}$ depending on $i$. We set $|e_i| = +1$ if these orientations coincide and $|e_i| = -1$ otherwise and take $\bar{s}(e_i), \bar{t}(e_i) \in \{0, 1, 2\}$ to mean respectively $s(e_i), t(e_i)$ if $|e_i| = +1$ and $t(e_i), s(e_i)$ if $|e_i| = -1$. Note also that in case $f$ is degenerate, not all indices $f(1), \ldots, f(l)$ and the edges $e_1, \ldots, e_l$ need to be distinct.

The steps to obtain the morphism $\Omega_f$ defining $P_f$ can be sketched as

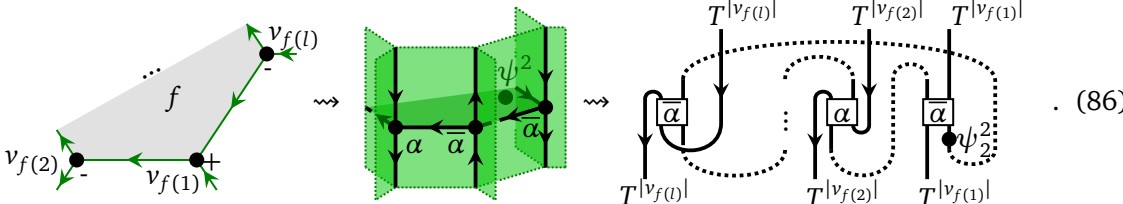

$$\tag{86}$$

The last step in particular involves taking a planar projection of the ribbon graph as in Figure 10b and straightening its coupons so that they can be unambiguously read as morphisms in $\mathcal{C}$. This step is the most tedious and requires some case analysis, which is listed in Table 4. In its first column we list the 6 different configurations of the pair of edges $v_{f(i-1)} \xrightarrow{e_{i-1}} v_{f(i)} \xrightarrow{e_i} v_{f(i+1)}$ at vertex $v_{f(i)}$ and the second column depicts the corresponding $\alpha/\bar{\alpha}$-labelled junction in the stratification defining $P_f$ in terms of the defect TFT $Z_{\mathcal{C}}^{\text{def}}$ as in Figure 7c. The third column then shows the corresponding morphism

$$\omega_i \colon T^{\pm} T^{\pm} \to T^{\pm} T^{\pm}, \tag{87}$$

replacing the junction in the projection of the ribbon graph. The fourth column introduces the morphisms $h_i^{\psi}$ which are used to handle the $\psi^2$-insertion on the horizontal surface, for example in the configuration of the first row, $\omega_i \circ h_i^{\psi}$ replaces

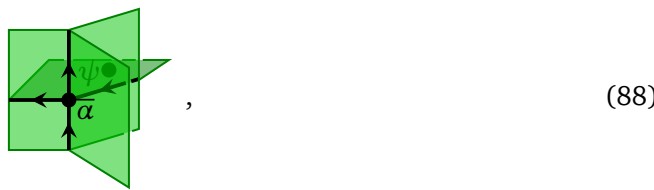

$$\tag{88}$$

as is incidentally already evident in (86).

Finally, let us denote:

$$\text{coev}_1 := \begin{cases} \text{coev}_T \otimes \text{id}_{T^{|v_{f(1)}|}}, & \text{if } |e_1| = +1, \\ \text{id}_{T^{|v_{f(1)}|}} \otimes \widetilde{\text{coev}}_T, & \text{if } |e_1| = -1, \end{cases} \qquad \text{ev}_1 := \begin{cases} \text{ev}_T \otimes \text{id}_{T^{|v_{f(1)}|}}, & \text{if } |e_1| = +1, \\ \text{id}_{T^{|v_{f(1)}|}} \otimes \widetilde{\text{ev}}_T, & \text{if } |e_1| = -1. \end{cases} \tag{89}$$

Table 4: Possible configurations of a face $f$ at a vertex $v_{f(i)}$ and the corresponding $\alpha/\overline{\alpha}$-labelled junctions in the stratification $S_f$ as in Figure 7c. The morphisms $\omega_i$ are conventional choices for the planar projection of the $\alpha/\overline{\alpha}$- labelled coupons when replacing $S_f$ by a ribbon graph as in Figure 10b. Precomposition with the morphisms $h_i^{\psi}$ adds a $\psi$-insertion on the horizontal surface, cf. (88).

| Configuration at the vertex $v_{f(i)}$ | Neighbourhood of the $\alpha/\overline{\alpha}$-labelled junction | $\omega_i$ | $h_i^{\psi}$ |
|---|---|---|---|
| | | | |

One then defines the morphism $\Omega_f$ as follows:

$$\Omega_f = \phi^2$$

$$\cdot\,(\mathrm{id}_{X^{|v_1|}}\otimes\cdots\otimes\left(\psi^{|v_{f(l)}|1/2}_{\bar{s}(e_l),\bar{t}(e_{l-1})}\circ\iota^{|v_{f(l)}|}\right)\otimes\cdots\otimes\left(\psi^{|v_{f(1)}|1/2}_{\bar{s}(e_1),\bar{t}(e_l)}\circ\iota^{|v_{f(1)}|}\right)\otimes\cdots\otimes\mathrm{id}_{X^{|v_n|}}) \tag{90a}$$

$$\circ\,(\mathrm{id}_{X^{|v_1|}}\otimes\cdots\otimes\mathrm{ev}_1\otimes\cdots\otimes\mathrm{id}_{X^{|v_n|}}) \tag{90b}$$

$$\circ\,a^{-|e_1|}_{e_1}(T^*) \tag{90c}$$

$$\circ\left(\mathrm{id}_{X^{|v_1|}}\otimes\cdots\otimes(\omega_1\circ(h^{\psi}_1)^2)\otimes\cdots\otimes\mathrm{id}_{X^{|v_n|}}\right)\circ a^{|e_l|}_{e_l}(T) \tag{90d}$$

$$\circ\,a^{|e_1|}_{e_1}(T^*) \tag{90e}$$

$$\circ\left(\prod_{i=2}^{l}(\mathrm{id}_{X^{|v_1|}}\otimes\cdots\otimes\omega_i\otimes\cdots\otimes\mathrm{id}_{X^{|v_n|}})\circ a^{|e_{i-1}|}_{e_{i-1}}(T)\right) \tag{90f}$$

$$\circ\,(\mathrm{id}_{X^{|v_1|}}\otimes\cdots\otimes\mathrm{coev}_1\otimes\cdots\otimes\mathrm{id}_{X^{|v_n|}}) \tag{90g}$$

$$\circ\,(\mathrm{id}_{X^{|v_1|}}\otimes\cdots\otimes\left(\pi^{|v_{f(l)}|}\circ\psi^{|v_{f(l)}|1/2}_{\bar{s}(e_l),\bar{t}(e_{l-1})}\right)\otimes\cdots\otimes\left(\pi^{|v_{f(1)}|}\circ\psi^{|v_{f(1)}|1/2}_{\bar{s}(e_1),\bar{t}(e_l)}\right)\otimes\cdots\otimes\mathrm{id}_{X^{|v_n|}}). \tag{90h}$$

A concrete example of this expression is written out in Figure 15 below. Note that steps (90c) and (90e) are somewhat redundant: all they do is move a $T^*$ puncture away and back along $e_1$. This is to avoid even more case analysis which would otherwise be needed in step (90d).

**Remark 12.** The expressions (84) and (90), as well as (70) of the state space $V$, evidently depend on the choice of the diffeomorphism $\varphi$ into the standard surface, as it dictates the order of the vertices of $\Gamma$ and the expression of the map $a_e$ for a given edge $e\subseteq\Gamma$. A different choice $\varphi'$ of this diffeomorphism yields a different form of the state space $V'$ and an isomorphism $\rho(\varphi'\circ\varphi^{-1})\colon V\xrightarrow{\sim}V'$, obtained from the action of the mapping class group element of $\varphi'\circ\varphi^{-1}$ on the standard surface $\Sigma_{g,n}$. For each component of the graph $c\subseteq\Gamma$, the corresponding projector $P'_c$ built from $\varphi'$ is related to $P_c$ by conjugation: $P'_c=\rho(\varphi'\circ\varphi^{-1})\circ P_c\circ\rho^{-1}(\varphi'\circ\varphi^{-1})$. The model $(V',H')$, where the Hamiltonian $H'$ is obtained by similarly conjugating (42), is therefore equivalent to $(V,H)$.

## 4.2 Computation on the torus

In this section we look at a concrete example, illustrating the expression (72) for the edge and face projectors, and in particular the formulas (84) and (90) for the morphisms $\Omega_c$ in terms of which they are defined.

Let $(\Sigma,\Gamma)$ be the 2-torus with the admissible 1-skeleton $\Gamma$ as shown in Figure 13a, which has 8 vertices, 12 edges and 4 faces; we will consider the projectors of three edges, marked as $e$, $e'$, $e''$ in the figure, and of the face $f$, which in the figure is shaded. As explained below (70), one starts by picking a homeomorphism into the standard surface $\varphi\colon\Sigma\to\Sigma_{g=1,n=8}$. The image $\varphi(\Gamma)$ for our choice of $\varphi$ is as shown in Figure 13c. The order of the vertices in Figure 13 is the one imposed by $\varphi$, which means that we can denote them by $v_1,\dots,v_8$ with $|v_i|=+$ for $i=1,6,7,8$ and $|v_i|=-$ for $i=2,3,4,5$. Consequently, the state space in the form (70) is:

$$V=\mathcal{C}(\mathbb{1},XX^*X^*X^*X^*XXXL). \tag{91}$$

We now turn to describing the morphisms $P_e$, $P_{e'}$, $P_{e''}$, $P_f$ in terms of the expression (72). In particular we need to untangle the definition of the morphisms

$$\Omega_e,\ \Omega_{e'},\ \Omega_{e''},\ \Omega_f\ \in\ \mathrm{End}_{\mathcal{C}}(XX^*X^*X^*X^*XXXL), \tag{92}$$

laid out in (84) and (90):

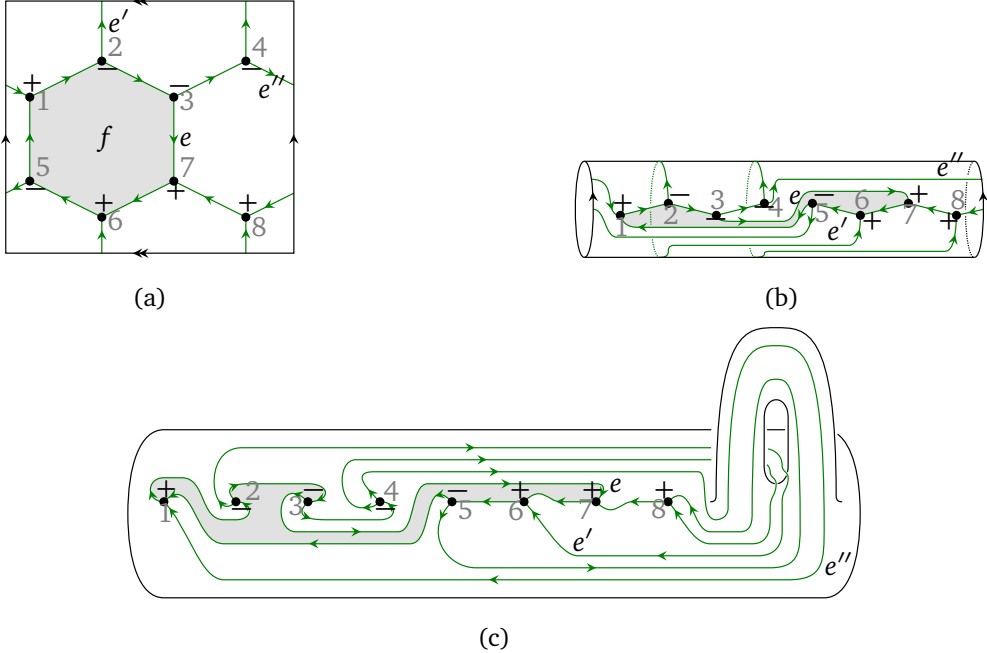

Figure 13: (a) Example of $T^2$ with 1-skeleton. (b) Intermediate step in the mapping to the standard surface $\Sigma_{1,8}$. Next, one embeds this figure as a doughnut in $\mathbb{R}^3$ and further deforms it to arrive at (c). Note that the punctures in (c) are aligned such that each of them has a neighbourhood as in (83).

- As shown in Figure 13c, the edge $e$ starts at $v_3$ and ends at $v_7$ with $s(e) = 1$ and $t(e) = 1$. As $e$ passes neither through nor over the arc of the handle of the standard torus, it is of the 1st type shown in Figure 12. The morphism $a_e(A)$ featuring in (84c) is therefore obtained by twisting the $A$-labelled strand of the idempotent (18) and braiding it with the $X^{\pm}$-labelled strands. The precise expression can be read off to be the one in Figure 14a.

- The edge $e'$ starts at $v_2$ and ends at $v_6$ with $s(e') = 2$ and $t(e') = 2$. It passes through the handle of the standard torus and therefore is of the 2nd type in Figure 12, meaning that $a_{e'}(A)$ will contain the projection/inclusion morphisms (69). The expression for $\Omega_{e'}$ is then the one in Figure 14b.

- The edge $e''$ between $v_4$ and $v_1$, with $s(e'') = 1$ and $t(e'') = 2$, passes over the handle of the standard torus and is of the 3rd type in Figure 12, therefore $a_{e''}(A)$ contains the halfbraiding (74). The expression for $\Omega_{e''}$ is shown in Figure 14c.

- Using the convention below (85), the vertex-edge chain bounding the face $f$ is read from Figure 13 to be

$$v_7 \xrightarrow{e_1} v_6 \xrightarrow{e_2} v_5 \xrightarrow{e_3} v_1 \xrightarrow{e_4} v_2 \xrightarrow{e_5} v_3 \xrightarrow{e_6 = e} v_7, \tag{93}$$

i.e. $f(1) = 7$, $f(2) = 6$, ..., $f(6) = 3$. As the directions of the corresponding edges of $\Gamma$ are the same as in the chain (93), we have $|e_i| = +1$ for all $i = 1, \ldots, 6$, and so (89) yields $\mathrm{coev}_1 = \mathrm{coev}_T \otimes \mathrm{id}_T$ and $\mathrm{ev}_1 = \mathrm{ev}_T \otimes \mathrm{id}_T$. The morphisms $\omega_i$, $i = 1, \ldots, 6$ are picked according to Table 4: the configuration of the face $f$ at $\xrightarrow{e_1} v_6 \xrightarrow{e_2}$ is the one in the first row, at $\xrightarrow{e_2} v_5 \xrightarrow{e_3}$ is the one in the fourth row, ..., at $\xrightarrow{e_6} v_7 \xrightarrow{e_1}$ is the one in the first row (the latter also determines $h_1^{\psi} = \mathrm{id}_T \otimes \psi_1$.). The morphisms $a_{e_i}^{\pm 1}(T^{\pm})$, $i = 1, \ldots, 6$, are determined similarly as when defining $\Omega_e$, $\Omega_{e'}$, $\Omega_{e''}$ above. The final expression for $\Omega_f$ is shown in Figure 15.

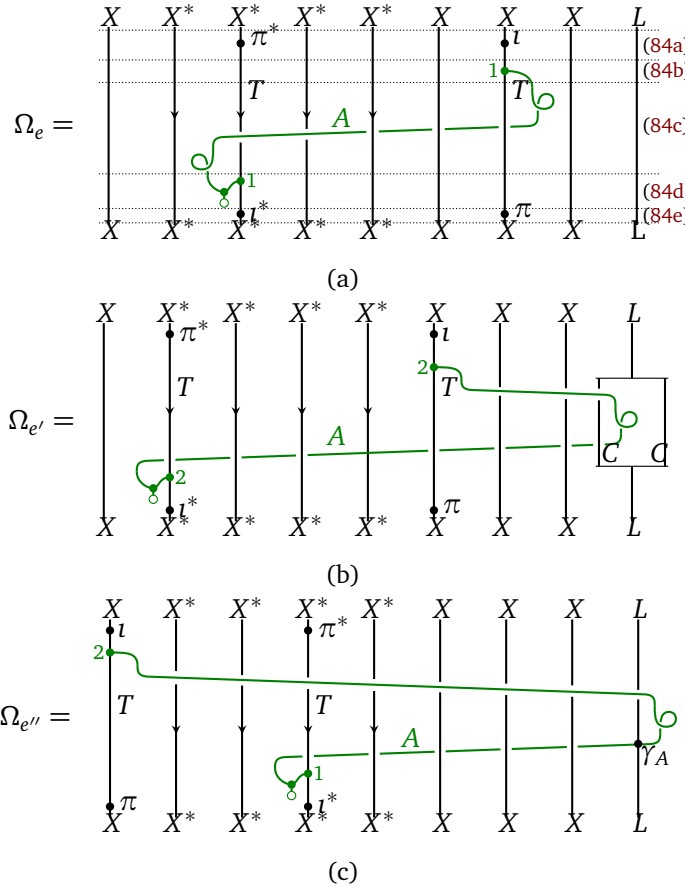

Figure 14: The map $\Omega_e$ in (84) for the edges $e$, $e'$, $e''$ in the $T^2$ example from Figure 13c.

## 5 Examples

In Section 3.2 we reviewed three examples of internal Levin–Wen data: one in an arbitrary MFC $\mathcal{C}$ which is obtained from a condensable algebra in it, and two in the trivial MFC $\mathcal{C} = \mathrm{Vect}_{\Bbbk}$ which are obtained from a spherical fusion category $\mathcal{S}$ and a Hopf algebra $K$ respectively. In this section we explain them in more detail, in particular how the Hamiltonian (42) simplifies in each of these cases. As a reference example, for the latter two instances we specialise the computation from Section 4.2.

### 5.1 Condensations

Let $A$ be a condensable algebra in a MFC $\mathcal{C}$ and consider the internal Levin–Wen datum as in Section 3.2.1. In this case, as one has $X = T = A \cong A^*$ (the latter isomorphism follows from $A$ being symmetric), the state-space of the model in the form (70) reads

$$V = \mathcal{C}(\mathbb{1}, A^{\otimes n} \otimes L^{\otimes g}),\tag{94}$$

where $g$ is the genus of the surface $\Sigma$ and $n$ is the number of vertices of the graph $\Gamma \subseteq \Sigma$.

Since in this particular Levin–Wen datum one has $\pi = \iota = \mathrm{id}_A$, (73) yields $P_v = \mathrm{id}_V$ for all vertices $v \in \Gamma_0$. This also allows us to simplify the expressions (72), (84) for the projector $P_e$ associated to an edge $v_j \xrightarrow{e} v_k$ of $\Gamma$ by removing the lines (84a), (84e). Since $\psi = \mathrm{id}_A$, one can also disregard $\psi^{-2}$ in (84b). For an edge $v_k \xrightarrow{e} v_{k+1}$ which does not braid with other

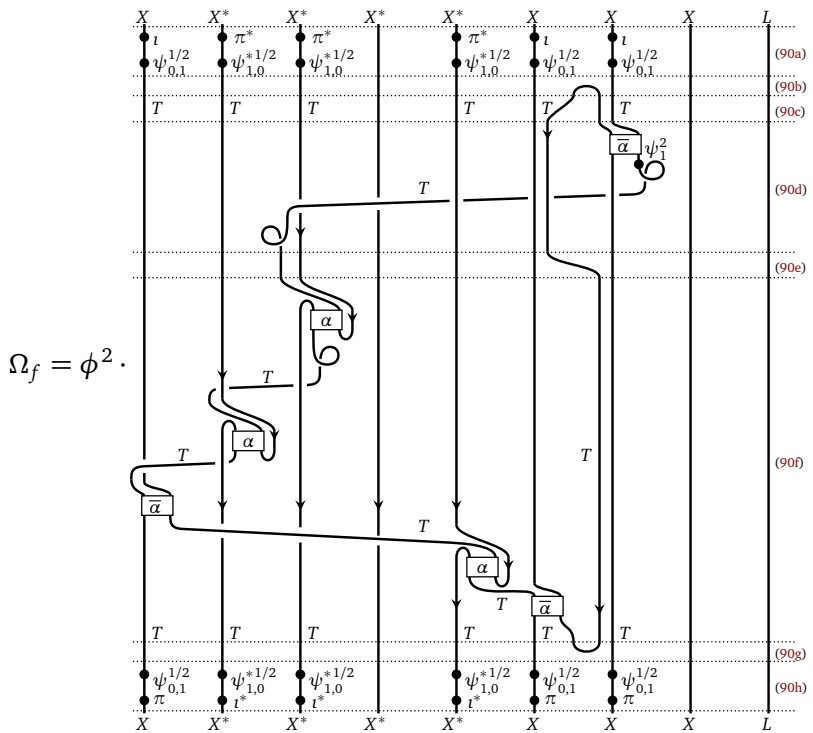

Figure 15: The map $\Omega_f$ in (84) for the face $f$ in Figure 13c.

punctures or wraps around the handles of $\Sigma$, $\Omega_e$ is simply the idempotent (18) projecting the tensor factor $A \otimes A$ at the position $(k, k+1)$ of $A^{\otimes n} \otimes L^{\otimes g}$ onto a single $A$-factor. In other words, the image of $P_e$ is obtained by fusing the two $A$-punctures, so that one has

$$\operatorname{im} P_e \cong \mathcal{C}(\mathbb{1}, A^{\otimes(n-1)} \otimes L^{\otimes g}).\tag{95}$$

One can similarly simplify the expressions (72), (90) for the projector $P_f$ associated to a face bounding the vertex-edge chain $v_{f(1)} \xrightarrow{e_1} v_{f(2)} \cdots \xrightarrow{e_{l-1}} v_{f(l)}$. Using the expressions for the $\alpha/\overline{\alpha}$-morphisms in the orbifold datum (50) and the identities (17), (49) one has for example (cf. (86))

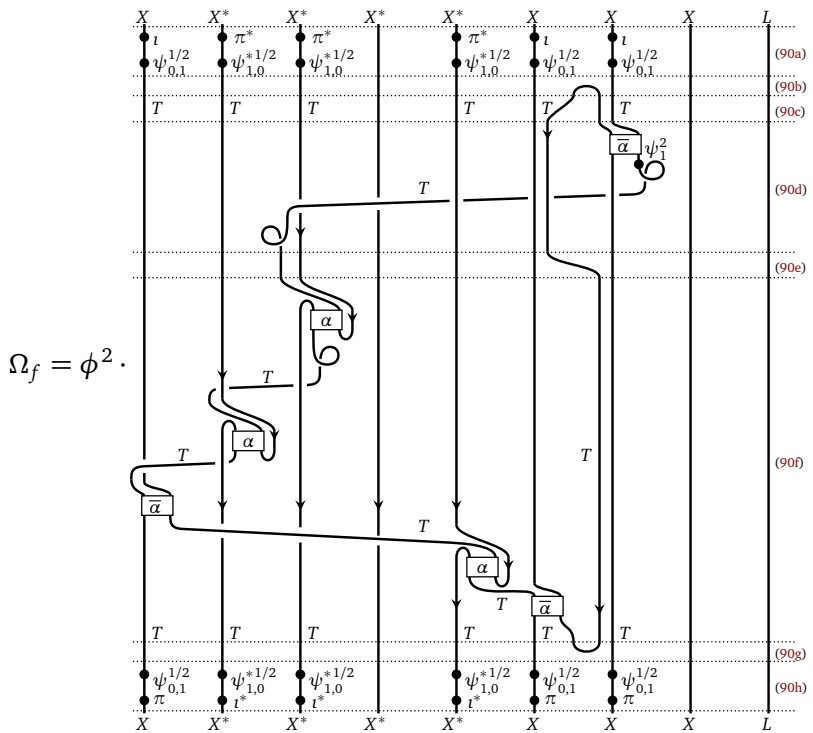

i.e. in this instance of internal Levin–Wen datum the projector of the face $f$ is the composition of the projectors of the bounding edges:

$$P_f = P_{e_{l-1}} \circ \cdots \circ P_{e_1}.\tag{97}$$

Consequently, (44) implies that the ground state space is the common image of all the edge projectors:

$$V_0 = \operatorname{im} \prod_{e \in \Gamma_1} P_e.\tag{98}$$

We now note that the argument leading to (95) can be successively repeated for a collection of edges $S \subseteq \Gamma_1$ forming a spanning tree of $\Gamma$. On the surface $\Sigma$ this has the effect of fusing all

$A$-labelled punctures into a single one, which leaves a (non-trivalent) graph on $\Sigma$ having one vertex and $|\Gamma_1 \setminus S|$ edges. One therefore has

$$V_0 \subseteq \mathrm{im} \prod_{e \in S \subseteq \Gamma_1} P_e \cong \mathcal{C}(\mathbb{1}, A \otimes L^{\otimes g}). \tag{99}$$

Since $A$ is haploid, for $g = 0$, i.e. when $\Sigma = S^2$, this already implies $\dim V_0 = 1$, meaning that the ground state is not degenerate. This is in accord with Theorem 8 and (33), as a TFT of Reshetikhin–Turaev type always assigns a 1-dimensional state space to the 2-sphere. On the 2-torus $\Sigma = T^2$, $V_0$ is in general a proper subspace in (99). To see this, let us explicitly compute the image on the right-hand side of (98): we collapse all vertices into a single $A$-puncture as described above taking the initial admissible skeleton to be the one in Figure 13. This results in a non-trivalent graph on $T^2$ with some parallel edges, which can be subsequently removed, as they result in applying the same idempotent twice when computing the image (98):

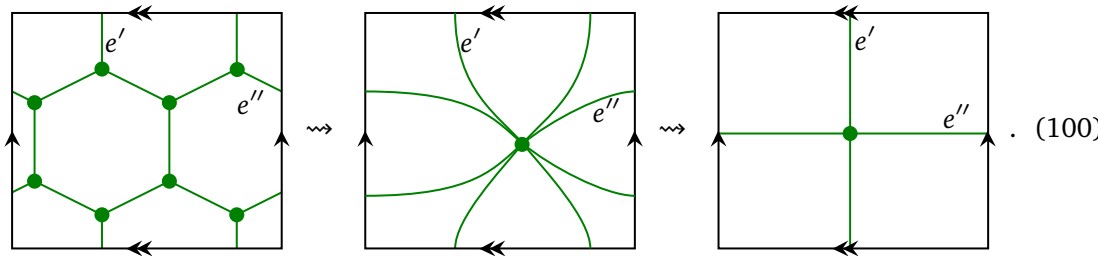

$$\tag{100}$$

This leaves two edges, denoted above by $e'$ and $e''$, each of which has the associated projector on the space $\mathcal{C}(\mathbb{1}, A \otimes L)$, whose forms are $f \mapsto \widetilde{\Omega}_{e'} \circ f$ and $f \mapsto \widetilde{\Omega}_{e''} \circ f$ for endomorphisms $\widetilde{\Omega}_{e'}, \widetilde{\Omega}_{e''} \in \mathrm{End}_{\mathcal{C}}(A \otimes L)$. Choosing an analogous parametrisation by the standard surface as shown in Figure 13c, the endomorphisms $\widetilde{\Omega}_{e'}, \widetilde{\Omega}_{e''}$ can be read off similarly as $\Omega_{e'}, \Omega_{e''}$ in Figure 14, the difference being that most of the $X{=}A$-labelled strands are not present due to the initial fusing step, and $A$-$A$-crossings and $A$-twists are not present due to $A$ being commutative:

$$\tag{101}$$

This yields $V_0$ as the image of the projector $\mathcal{C}(\mathbb{1}, A \otimes L) \ni f \mapsto \widetilde{\Omega}_{e''} \circ \widetilde{\Omega}_{e'} \circ f$. By Theorem 8 and (33), the dimension of $V_0$ is given by the number $|\mathrm{Irr}_{\mathcal{C}_A^{\mathrm{loc}}}|$ of isomorphism classes of simple local $A$-modules. Explicitly, a basis $\{v_\alpha\}$ of $V_0$ is given in terms of simple local $A$-modules $M_\alpha$, $\alpha \in \mathrm{Irr}_{\mathcal{C}_A^{\mathrm{loc}}}$ via

$$v_\alpha = \left[ \mathbb{1} \xrightarrow{(\Delta \circ \eta) \otimes \mathrm{coev}_{M_\alpha}} AAM_\alpha M_\alpha^* \xrightarrow{\mathrm{id} \otimes \rho \otimes \mathrm{id}} AM_\alpha M_\alpha^* \xrightarrow{\mathrm{id} \otimes \iota} AL \right]. \tag{102}$$

Here $\iota$ is obtained by decomposing $M_\alpha$ into simple objects $i$ of $\mathcal{C}$ and embedding into $L$ via $\mathrm{coev}_i$ (recall (13)). Since we do not need this basis below, we skip the details.

## 5.2 Original Levin–Wen models

In this section we show how the model based on the internal Levin–Wen datum from Section 3.2.2 reproduces the original Levin–Wen model from [43].

Figure 16: The vector spaces $V \supseteq V_V \supseteq V_E$.

### 5.2.1 The state space

Let $\Sigma$ be a compact oriented surface and $\Gamma \subseteq \Sigma$ an admissible 1-skeleton. As the internal Levin–Wen datum we use has the underlying modular fusion category (MFC) $\mathrm{Vect}_{\Bbbk}$, there are a few simplifications:

- The object $L$ defined in (13) is simply given by $L = \Bbbk$.

- All braidings and twists are trivial.

- The state space is $V = \mathrm{Hom}_{\Bbbk}\left(\Bbbk, X^{\otimes \Gamma_0}\right) \cong X^{\otimes \Gamma_0}$, where $\Gamma_0$ is the set of vertices of $\Gamma$ (combine (39) with (14)). Here, $X^{\otimes \Gamma_0} := \otimes_{v \in \Gamma_0} X^{|v|}$ is a $\Gamma_0$-fold tensor product of the $X$ or $X^*$ with every copy of $X$ or $X^*$ associated to one vertex according to the orientation of the vertex.

Since $X = \oplus_{i,j,k \in I} x$ (recall (55h)) for a set $I = \mathrm{Irr}_{\mathcal{S}}$ of representatives of irreducible elements in $\mathcal{S}$. $V$ can be written as a direct sum of subspaces associated to colourings of the graph $\Gamma$ as follows (see Figure 16):

- Half-edges are coloured by objects $i, j, k, \cdots \in I$.

- In every such colouring we associate a copy of the vector space $x$ or $x^*$ to every vertex, depending on the orientation of that vertex.

- The state space is the vector space $\oplus x^{\otimes \Gamma_0} = \oplus \otimes_{v \in \Gamma_0} x^{|v|}$, where we sum over every such colouring.

**Vertex and edge projectors and their image**

The vertex projector $P_v$ is a direct sum of maps $x \to x$ over all colourings of (the half-edges attached to) $v$. On one colouring by a triple $(i, j, k)$, $P_v$ acts by applying the idempotent map

$$x \xrightarrow{\pi_{ijk}} \mathcal{S}(k, ij) \xrightarrow{\iota_{ijk}} x, \tag{103}$$

from (57). The map $P_v$ thus projects to a subspace of $x$ which we can identify with $\mathcal{S}(k, ij)$. Note that this subspace is only non-zero if the triple $(i, j, k)$ is admissible, i.e. if the multiplicity of $k$ in $i \otimes j$ is non-zero. Similarly, if the vertex is coloured with $x^*$, we can identify the image of the projector $P_v$ with $\mathcal{S}(ij, k)$.

Let $P_V : V \to V$ be the product of the $P_v$'s and denote the image of $P_V$ by $V_V$. Explicitly, $V_V$ is the direct sum of graphs with half-edges coloured by $I$, but now taking the vector spaces $\mathcal{S}(k, ij)$ or $\mathcal{S}(ij, k)$ instead of $x$ and $x^*$ as in Figure 16

For every edge $e$ in $\Gamma$ connecting vertices $v$ and $w$ we now consider the edge projector $P_e$. We restrict our description to the situation in Figure 16, where the two vertices are distinct,

one is positively oriented and the other is negatively oriented. We give the projector on a colouring of $v$ by $(i,j,k)$ and $w$ by $(l,m,n)$ by a composition of maps:

$$x^* \otimes x \xrightarrow{\iota^*_{lmn} \otimes \pi_{ijk}} \mathcal{S}(lm,n) \otimes \mathcal{S}(k,ij) \xrightarrow{\delta_{kn}} \mathcal{S}(lm,k) \otimes \mathcal{S}(k,ij) \xrightarrow{\pi^*_{lmk} \otimes \iota_{ijk}} x^* \otimes x \,. \tag{104}$$

A priori, the two colourings of the half-edges of $e$ at $v$ and $w$ do not need to coincide. This map projects onto the space of admissible colourings, where both half-edges of $e$ are coloured by the same simple object of $\mathcal{S}$. This holds analogously for any other configuration of the edge $e$ and the two vertices $v$ and $w$.

Write $P_{\mathrm{E}} : V \to V$ for the product of the $P_e$'s and set $V_{\mathrm{E}} = \mathrm{im}(P_{\mathrm{E}})$. The image $V_{\mathrm{E}}$ is a subspace of $V_{\mathrm{V}}$ and can be described as the direct sum over colourings of edges of $\Gamma$ by $I$ (rather than half-edges), see Figure 16.

**Face projectors**

We will now compute the face projector for the face from the torus example in Figure 13. The face projector is given by the string diagram in Figure 15. As we are working with orbifold data in $\mathrm{Vect}_{\Bbbk}$, the braidings and twists are trivial. Disregarding those, the face projector is a composition of the maps $\pi, \iota$, their duals, the (co-)evaluation of $T$, the maps and

$$\alpha' := \left[ T^* \otimes T \xrightarrow{\mathrm{id} \otimes \mathrm{coev}_T} T^* \otimes T \otimes T \otimes T^* \xrightarrow{\mathrm{id} \otimes \alpha \otimes \mathrm{id}} T^* \otimes T \otimes T \otimes T^* \xrightarrow{\mathrm{ev}_T \otimes \mathrm{id}} T \otimes T^* \right]. \tag{105}$$

**Lemma 13.** The map $\alpha'$ is given on $\lambda \in \mathcal{S}(k,ij) \subseteq T$ and $\widehat{\mu} \in \mathcal{S}(lm,n) \subseteq T^*$ by

$$\alpha'(\widehat{\mu} \otimes \lambda) = \sum_{\sigma,\tau} F^{\lambda\widehat{\mu}}_{\sigma\widehat{\tau}} \tau \otimes \widehat{\sigma} \,. \tag{106}$$

*Proof.* We compute the map (105) step-by-step:

$$\widehat{\mu} \otimes \lambda \xmapsto{\mathrm{id} \otimes \mathrm{coev}_T} \sum_{\sigma} \widehat{\mu} \otimes \lambda \otimes \sigma \otimes \widehat{\sigma} \xmapsto{\mathrm{id} \otimes \alpha \otimes \mathrm{id}} \sum_{\sigma,\tau,\rho} F^{\lambda\widehat{\rho}}_{\sigma\widehat{\tau}} \widehat{\mu} \otimes \rho \otimes \tau \otimes \widehat{\sigma}$$

$$\xmapsto{\mathrm{ev}_T \otimes \mathrm{id}} \sum_{\sigma,\tau,\rho} F^{\lambda\widehat{\rho}}_{\sigma\widehat{\tau}} \mathrm{tr}_{\mathcal{S}}(\widehat{\mu} \circ \rho) \, \tau \otimes \widehat{\sigma} = \sum_{\sigma,\tau} F^{\lambda\widehat{\mu}}_{\sigma\widehat{\tau}} \tau \otimes \widehat{\sigma} \,. \tag{107}$$

$\square$

We further simplify the diagram in Figure 15 by taking its 'middle' part without the $\phi^2$-factor only, i.e. omitting the maps $\pi, \pi^*, \iota, \iota^*$ and $\psi^{1/2}_{i,j}, \psi^{*\,1/2}_{i,j}$ at the top and the bottom. The latter introduce some dimension factors which we will include in the final formula by hand. The resulting diagram is depicted in Figure 17. In it we also indicate the labels of the concrete elements $\epsilon, \cdots \in T$ and $\widehat{\delta}, \cdots \in T^*$ which occur in the computation below. Each of them is assumed to be in the subspace of type $\mathcal{S}(k,ij) \subseteq T$ or $\mathcal{S}(ij,k) \subseteq T^*$ for some $i,j,k \in I$. Some of the domains can be partially read off from Figure 18, for example the domain of $\gamma$ is $a \in I$, others can be deduced from identities (25) of $\alpha$ and $\overline{\alpha}$ morphisms, for example we take $\sigma \in \mathcal{S}(f,kf')$ for $k \in I$.

We now have everything set to compute the face projector. In every step of the computation only two tensor factors change while the other tensor factors are left invariant. Thus, to make the computation more legible we leave out any tensor factors not required in a particular step. In Figure 17 the steps (108a)–(108h) in the computation below are separated by dashed lines

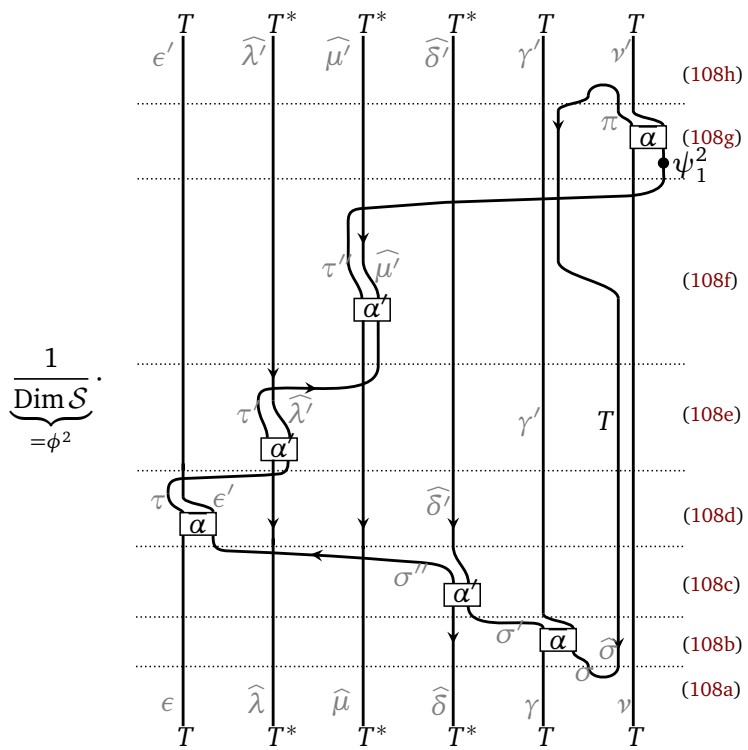

Figure 17: The simplified version of the diagram (90) in $\mathcal{C} = \text{Vect}_\Bbbk$. The steps in computation (108) are separated by dashed lines.

and labelled accordingly. This yields:

$$\epsilon \otimes \widehat{\lambda} \otimes \widehat{\mu} \otimes \widehat{\delta} \otimes \gamma \otimes \nu \xrightarrow{\cdots \otimes \text{coev}_T \otimes \cdots} \sum_{k,\sigma} \left( \cdots \otimes \gamma \otimes \sigma \otimes \widehat{\sigma} \otimes \cdots \right) \tag{108a}$$

$$\xrightarrow{\cdots \overline{a} \cdots} \sum_{k,\gamma',\sigma',\sigma} \overline{F}^{\gamma \widehat{\sigma'}}_{\sigma \widehat{\gamma'}} \left( \cdots \sigma' \otimes \gamma' \otimes \widehat{\sigma} \cdots \right) = \sum_{\cdots} \left( \cdots \right) \left( \cdots \widehat{\delta} \otimes \sigma' \cdots \right) \tag{108b}$$

$$\xrightarrow{\cdots \alpha' \cdots} \sum_{\delta',\sigma'',\ldots} (\cdots) F^{\sigma' \widehat{\delta}}_{\delta' \widehat{\sigma''}} \left( \cdots \sigma'' \otimes \widehat{\delta'} \cdots \right) \mapsto \sum_{\cdots} (\cdots) \left( \epsilon \otimes \sigma'' \cdots \right) \tag{108c}$$

$$\xrightarrow{\cdots \overline{a} \cdots} \sum_{\epsilon',\tau,\ldots} (\cdots) \overline{F}^{\epsilon \widehat{\tau}}_{\sigma'' \widehat{\epsilon'}} \left( \cdots \tau \otimes \epsilon' \cdots \right) \mapsto \sum_{\cdots} (\cdots) \left( \cdots \widehat{\lambda} \otimes \tau \cdots \right) \tag{108d}$$

$$\xrightarrow{\cdots \alpha' \cdots} \sum_{\lambda',\tau',\ldots} (\cdots) F^{\tau \widehat{\lambda}}_{\lambda' \widehat{\tau'}} \left( \cdots \tau' \otimes \widehat{\lambda'} \cdots \right) \mapsto \sum_{\cdots} (\cdots) \left( \cdots \widehat{\mu} \otimes \tau' \cdots \right) \tag{108e}$$

$$\xrightarrow{\cdots \alpha' \cdots} \sum_{\mu',\tau'',\ldots} (\cdots) F^{\tau' \widehat{\mu}}_{\mu' \widehat{\tau''}} \left( \cdots \tau'' \otimes \widehat{\mu'} \cdots \right) \mapsto \sum_{\cdots} (\cdots) \left( \cdots \nu \otimes \tau'' \right) \tag{108f}$$

$$\xrightarrow{\cdots \overline{a} \circ (\text{id}_T \otimes \psi_1^2) \cdots} \sum_{\nu',\pi,\sigma,\ldots} (\cdots) \cdot d_k \cdot \overline{F}^{\nu \widehat{\pi}}_{\tau'' \widehat{\nu'}} \left( \cdots \widehat{\sigma} \otimes \pi \otimes \nu' \right) \tag{108g}$$

$$\xrightarrow{\cdots \frac{1}{\text{Dim} \mathcal{S}} \cdot \text{ev}_T \cdots} \sum_{\nu',\pi,\sigma,\ldots} (\cdots) \cdot d_k \cdot \overline{F}^{\nu \widehat{\pi}}_{\tau'' \widehat{\nu'}} \cdot \frac{1}{\text{Dim} \mathcal{S}} \cdot \text{tr}_{\mathcal{S}} (\widehat{\sigma} \circ \pi) \left( \cdots \otimes \nu' \right)$$

$$= \sum_{\nu',\ldots} (\cdots) \cdot d_k \cdot \overline{F}^{\nu \widehat{\sigma}}_{\tau'' \widehat{\nu'}} \cdot \frac{1}{\text{Dim} \mathcal{S}} \left( \cdots \otimes \nu' \right). \tag{108h}$$

Gathering all the terms as well as the previously omitted factors due to $\psi_{i,j}^{1/2}, \psi_{i,j}^{*1/2}$-insertions, we obtain the following formula for the face operator:

$$
\epsilon \otimes \widehat{\lambda} \otimes \widehat{\mu} \otimes \widehat{\delta} \otimes \gamma \otimes \nu \longmapsto \sum_{k,a',b',c',d',e',f' \in I} \sum_{\substack{\epsilon',\lambda',\mu',\delta',\gamma',\nu' \\ \sigma,\sigma',\sigma'',\tau,\tau',\tau''}} \frac{\left(d_a d_b d_c d_d d_e d_f d_{a'} d_{b'} d_{c'} d_{d'} d_{e'} d_{f'}\right)^{1/2} \cdot d_k}{\operatorname{Dim} \mathcal{S}}
$$

$$
\cdot \overline{F}_{\sigma \widehat{\gamma}'}^{\gamma \widehat{\sigma}'} \cdot F_{\delta' \widehat{\sigma}''}^{\sigma' \widehat{\delta}} \cdot \overline{F}_{\sigma'' \widehat{\epsilon}'}^{\epsilon \widehat{\tau}} \cdot F_{\lambda' \widehat{\tau}''}^{\tau \widehat{\lambda}} \cdot F_{\mu' \widehat{\tau}''}^{\tau' \widehat{\mu}} \cdot \overline{F}_{\tau'' \widehat{\nu}'}^{\nu \widehat{\sigma}}
$$

$$
\cdot \left(\epsilon' \otimes \widehat{\lambda'} \otimes \widehat{\mu'} \otimes \widehat{\delta'} \otimes \gamma' \otimes \nu'\right). \tag{109}
$$

To see how the labels on the face have transformed, see Figure 18.

### 5.2.2 Comparison with the original Levin–Wen model

Let us now compare the original Levin–Wen string net model from [43] with our model. To obtain the 6-index symbols $F_{kln}^{ijm} \in \Bbbk$ from [43], we have to restrict ourselves to spherical fusion categories where every multiplicity $\dim \mathcal{S}(k, ij)$ for simple objects $i, j, k \in I$ is at most one. Then:

- The state space $V$ in our model is slightly bigger than the one in [43], as we colour half-edges with irreducible objects and [43] colours edges. The difference can be bridged by considering the image of the edge projectors instead of $V$. Because of this, there is no analogue for edge projectors in [43].

- The vertex projectors of our model are analogues for the *electric charge operator* from [43, Eq. (11)].

- The $F$-symbols $F_{kln}^{ijm}$ in [43] are dependent only on irreducible objects $i, j, k, l, m, n \in I$, while our $F$-symbols are defined in terms of elements of the Hom-spaces between these objects. By choosing appropriate bases of the Hom-spaces and renormalising the $F$-symbols, we can translate between the two settings. For this it is essential, that the Hom-spaces are at most one-dimensional.

- For a hexagonal lattice, the face projector of the internal Levin–Wen model is given by (109), i.e. a sum over products of six $F$-symbols. After an appropriate translation, we obtain the same formulas as for the *magnetic flux operator* from [43, Eq.'s (12),(13)], we skip the details.

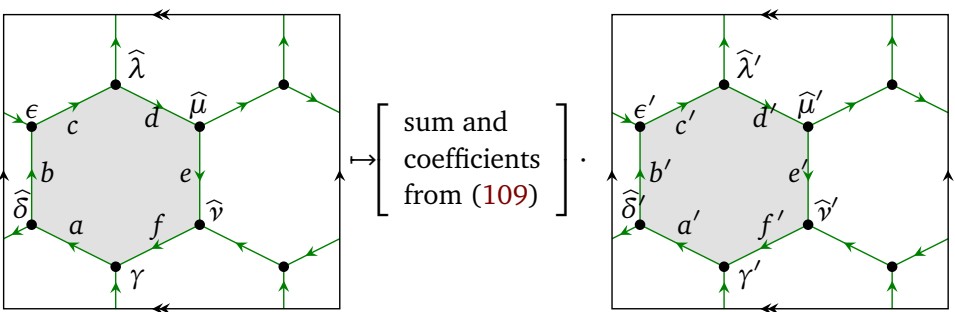

Figure 18: Computation of the face projector (cf. (109)). The input vertices are coloured with $\gamma, \epsilon, \nu \in T$ and $\widehat{\delta}, \widehat{\lambda}, \widehat{\mu} \in T^*$ with the relevant tensor factors in the domains being $a, b, c, d, e, f \in I$. The output vertices are labelled similarly.

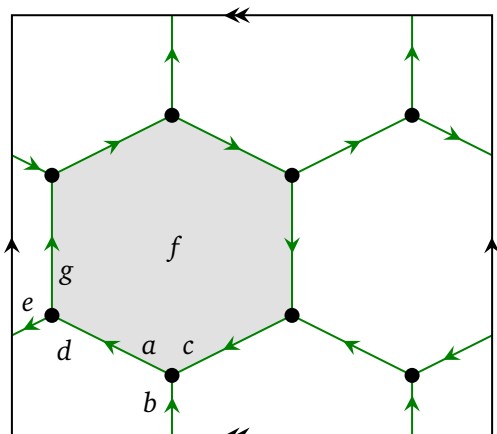

Figure 19: A 1-skeleton for the torus with labels $a, \dots, g \in H$ for the six edge ends at the two bottom left vertices.

Finally, a quick argument shows that the space of ground states $V_0$ coincides with the one from [43]: By Theorem 8 and (33), we have $V_0 \cong Z^{\mathrm{RT}}_{\mathcal{C}_{\mathbb{A}}}(\Sigma)$, where $\mathcal{C} = \mathrm{Vect}_{\Bbbk}$ and $\mathbb{A} = \mathbb{A}_{\mathcal{S}}$ is the orbifold data from the spherical fusion category $\mathcal{S}$. By [45, Thm. 4.1], $\mathcal{C}_{\mathbb{A}_{\mathcal{S}}}$ is equivalent to the Drinfeld centre $\mathcal{Z}(\mathcal{S})$ as ribbon categories. Then

$$V_0 \cong Z^{\mathrm{RT}}_{\mathcal{Z}(\mathcal{S})}(\Sigma) \cong Z^{\mathrm{TV}}_{\mathcal{S}}(\Sigma) \cong H^{\mathrm{LW}}_{\mathcal{S}}(\Sigma), \tag{110}$$

where we denote by $Z^{\mathrm{TV}}_{\mathcal{S}}$ the Turaev–Viro–Barrett–Westbury TFT based on the spherical fusion category $\mathcal{S}$ and by $H^{\mathrm{LW}}_{\mathcal{S}}$ the space of ground states of the Levin–Wen model from [43]. The last equivalence is shown in [6, Thm. 3.1].

## 5.3 Kitaev models

The original Kitaev model [35] was generalised in [7] to an input of a finite-dimensional semisimple Hopf algebra $K$. In this formulation, the ground states and excited states of the Kitaev model were proven in [1, 6] to be isomorphic to those of the Levin–Wen model with spherical fusion category $K$-rep of finite-dimensional $K$-modules. In this section we discuss the internal Levin–Wen model obtained from a finite-dimensional semisimple Hopf algebra $K$ as in Section 3.2.3. We find that our model reproduces an appropriately chosen Kitaev model based on $K$.

**The state space**

For our example we consider the admissible 1-skeleton $\Gamma$ on the torus in Figure 13. It has 8 vertices, hence, since we have $\mathcal{C} = \mathrm{Vect}_{\Bbbk}$ and $L = \Bbbk$, the state space (70) takes the form

$$V = \mathrm{Hom}_{\Bbbk}(\Bbbk, XX^*X^*X^*XXX) \cong K^{\otimes 24}, \tag{111}$$

where we used $X = K^{\otimes 3} \cong X^*$ (see Table 3). This can be seen as an assignment of a copy of $K$ to every half-edge of the graph $\Gamma$, see Figure 19.

In what follows we compute the projectors for a vertex, an edge, and a face of the 1-skeleton $\Gamma$. We compare these projectors to the ones of the Kitaev model associated to a modified graph $\Gamma'$. To obtain $\Gamma'$ one simply adds an additional vertex in the middle of every edge of $\Gamma$, see Figure 20. We do this in order to identify the state space $V = K^{\otimes 24}$ of the internal Levin–Wen model with the extended Hilbert space $\mathcal{H} = K^{\otimes 24}$ of the Kitaev model: the internal Levin–Wen model labels every half-edge of $\Gamma$ with $K$, whereas the Kitaev model labels every edge of $\Gamma'$ with $K$.

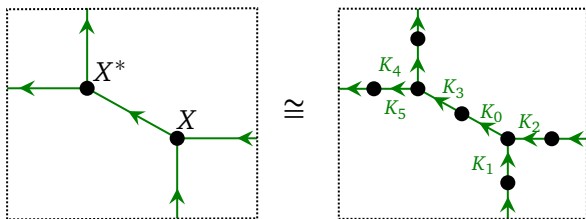

Figure 20: Modifying the graph $\Gamma$ by inserting additional vertices provides an isomorphism to the state space of the Kitaev model. Here we identify $X = K^{\otimes 3} = K_0 \otimes K_1 \otimes K_2$ and $X^* = K^{\otimes 3} = K_3 \otimes K_4 \otimes K_5$.

### Vertex projectors

For a vertex labelled with $a \otimes b \otimes c \in K^{\otimes 3} = X$ the vertex projector is the map $\iota \circ \pi : X \to X$ (see (64)):

$$
\begin{aligned}
\iota \circ \pi (a \otimes b \otimes c) &= \iota \left( a_{(2)} b \otimes a_{(1)} c \right) = S(\lambda_{(1)}) \otimes \lambda_{(3)} a_{(2)} b \otimes \lambda_{(2)} a_{(1)} c \\
&= a S(\lambda_{(1)}) \otimes \lambda_{(3)} b \otimes \lambda_{(2)} c \, .
\end{aligned} \tag{112}
$$

Similarly we obtain for the dual map $\pi^* \circ \iota^* : X^* \to X^*$ (see (67)):

$$
\pi^* \circ \iota^* (d \otimes e \otimes g) = \lambda_{(1)} d \otimes e S(\lambda_{(2)}) \otimes g S(\lambda_{(3)}) \, . \tag{113}
$$

These are precisely the vertex projectors $A_v^\lambda$ of the Kitaev model (cf. [56, Sec. 4]).

### Edge projectors

We now turn to the edge projector taking the edge labelled by $a$ and $d$ in Figure 19 as example. Removing the unnecessary tensor factors in the general formula (84), one arrives at the map

$$
(\pi^* \otimes \iota) \circ \beta \circ (\iota^* \otimes \pi) : X^* \otimes X \to X^* \otimes X \, , \tag{114}
$$

where $\beta$ is the map taking $T^* \otimes T$ to the relative tensor product $T^* \otimes_A T$, i.e. coming from the composition of (84d),(84b) and implementing an idempotent similar to the one in (18), where the coproduct $\Delta_{\mathrm{Fr}}$ in (60) is used. In this configuration, the relative tensor product is taken with respect to the action $\rhd_0$ on $T$ and its dual action $\lhd_0$ on $T^*$. Computing the map yields

$$
\begin{aligned}
&(\pi^* \otimes \iota) \circ \beta \circ (\iota^* \otimes \pi)([d \otimes e \otimes g] \otimes [a \otimes b \otimes c]) \\
&= (\pi^* \otimes \iota) \circ \beta \left( [ed_{(2)} \otimes gd_{(1)}] \otimes [a_{(2)} b \otimes a_{(1)} c] \right) \\
&= (\pi^* \otimes \iota) \left( \left[ ed_{(2)} S(\lambda_{(1)}^1) \otimes gd_{(1)} S\left(\lambda_{(2)}^1\right) \right] \otimes \left[ \lambda_{(4)}^1 a_{(2)} b \otimes \lambda_{(3)}^1 a_{(1)} c \right] \right) \\
&= \left[ \lambda_{(1)}^2 dS\left(\lambda_{(1)}^1\right) \otimes eS(\lambda_{(2)}^2) \otimes gS(\lambda_{(3)}^2) \right] \otimes \left[ \lambda_{(2)}^1 aS(\lambda_{(1)}^3) \otimes \lambda_{(3)}^3 b \otimes \lambda_{(2)}^3 c \right] \, ,
\end{aligned} \tag{115}
$$

where here and below we denote by $\lambda^1, \lambda^2, \lambda^3, \dots$ different copies of the normalised Haar integral $\lambda \in K$ (see (58)).

Again comparing with the Kitaev model we see that this map is a composition of two vertex projectors $A_v^\lambda$ of the original Kitaev model together with an additional vertex projector for a vertex added in the middle of the connecting edge labelled with $a$ and $d$ in Figure 19 as in Figure 20 (cf. [56, Sec. 4]).

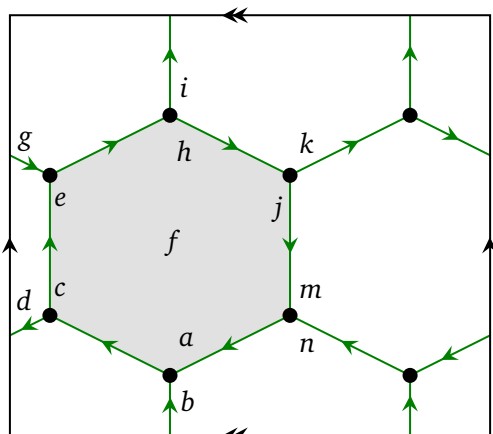

Figure 21: Vertex labels used in the computation of the face projector.

**Face projectors**

The computation for the face projectors is more involved. We will compute the face projector $P_f$ for the face $f$ of the torus example in Figure 13, which, as in Section 5.2, boils down to evaluating the diagram in Figure 15. Since we again work with an orbifold datum in $\mathrm{Vect}_{\Bbbk}$, here too the braiding and twist morphisms are trivial. Furthermore, we again omit the projectors $\pi, \iota^*$ and embeddings $\iota, \pi^*$, i.e. we only compute the projector on $TT^*T^*T^*TT$ obtained from the 'middle' part of the diagram in Figure 15. The outcome is the same as the diagram in Figure 17, but read for the orbifold datum based on the Hopf algebra $K$ instead. In particular, one has $\psi = \mathrm{id}$ and $\phi^2 = \frac{1}{|K|}$, see Table 3. The computation is performed for the input shown in Figure 21, where two elements of $K$ are used to label each of the six vertices, reflecting the fact that we have $T = K^{\otimes 2} = T^*$. The map $\alpha'$ in (105) in this case reads:

$$\alpha' \colon [a \otimes b] \otimes [c \otimes d] \mapsto \left[a_{(2)}c \otimes a_{(1)}d_{(2)}\lambda_{(1)}\right] \otimes \left[S(\lambda_{(2)}) \otimes bd_{(1)}\right]. \tag{116}$$

We provide a detailed computation of this map, as it is similar to other computations for the internal Levin–Wen datum based on $K$, where we provide fewer details for brevity:

$$[a \otimes b] \otimes [c \otimes d] \xmapsto{\cdots \mathrm{coev}_T} [a \otimes b] \otimes [c \otimes d] \otimes \left[\lambda^1_{(1)} \otimes \lambda^2_{(1)}\right] \otimes \left[S(\lambda^1_{(2)}) \otimes S(\lambda^2_{(2)})\right]$$

$$\xmapsto{\cdots \alpha \cdots} [a \otimes b] \otimes \left[S(\lambda^3_{(1)}) \otimes d_{(1)}\lambda^2_{(1)}\right] \otimes \left[\lambda^3_{(3)}c \otimes \lambda^3_{(2)}d_{(2)}\lambda^1_{(1)}\right] \otimes \left[S(\lambda^1_{(2)}) \otimes S(\lambda^2_{(2)})\right]$$

$$\xmapsto{\mathrm{ev}_T \cdots} |K|^2 \int\left(aS(\lambda^3_{(1)})\right) \int\left(bd_{(1)}\lambda^2_{(1)}\right) \cdot \left[\lambda^3_{(3)}c \otimes \lambda^3_{(2)}d_{(2)}\lambda^1_{(1)}\right] \otimes \left[S(\lambda^1_{(2)}) \otimes S(\lambda^2_{(2)})\right]$$

$$\overset{(59c)}{=} |K|^2 \int\left(\lambda^3_{(1)}S(a)\right) \int\left(bd_{(1)}\lambda^2_{(1)}\right) \cdot \left[\lambda^3_{(3)}c \otimes \lambda^3_{(2)}d_{(2)}\lambda^1_{(1)}\right] \otimes \left[S(\lambda^1_{(2)}) \otimes S(\lambda^2_{(2)})\right]$$

$$\overset{(59b)}{=} |K|^2 \int\left(\lambda^3_{(1)}\right) \int\left(\lambda^2_{(1)}\right) \cdot \left[\lambda^3_{(3)}a_{(2)}c \otimes \lambda^3_{(2)}a_{(1)}d_{(2)}\lambda^1_{(1)}\right] \otimes \left[S(\lambda^1_{(2)}) \otimes S(S(bd_{(1)})\lambda^2_{(2)})\right]$$

$$\overset{(58)}{=} |K|^2 \int\left(\lambda^3\right) \int\left(\lambda^2\right) \cdot \left[a_{(2)}c \otimes a_{(1)}d_{(2)}\lambda^1_{(1)}\right] \otimes \left[S(\lambda^1_{(2)}) \otimes S(S(bd_{(1)}))\right]$$

$$\overset{(59c)}{=} \left[a_{(2)}c \otimes a_{(1)}d_{(2)}\lambda^1_{(1)}\right] \otimes \left[S(\lambda^1_{(2)}) \otimes bd_{(1)}\right]. \tag{117}$$

Now we can compute the face projector for the Levin-Wen datum based on $K$. Similarly as in (108), in each step of the computation below only two tensor factors change - so again we attempt to make it more legible by leaving out any tensor factors not required in a particular

step. The steps (118a) to (118h) in the computation below correspond to the steps (108a)–(108h) separated by dashed lines in Figure 17. We then have:

$$[e\otimes g]\otimes[h\otimes i]\otimes[j\otimes k]\otimes[c\otimes d]\otimes[a\otimes b]\otimes[m\otimes n]$$

$$\xrightarrow{\cdots\text{coev}_T\cdots} \cdots[a\otimes b]\otimes\big[\lambda^1_{(1)}\otimes\lambda^2_{(1)}\big]\otimes\big[S(\lambda^1_{(2)})\otimes S(\lambda^2_{(2)})\big]\cdots \tag{118a}$$

$$\xrightarrow{\cdots\bar{\alpha}\cdots}\cdots\big[a_{(2)}\lambda^1_{(1)}\otimes S(\lambda^3_{(1)})\big]\otimes\big[\lambda^3_{(3)}a_{(1)}\lambda^2_{(1)}\otimes\lambda^3_{(2)}b\big]\cdots=\cdots[c\otimes d]\otimes\big[a_{(2)}\lambda^1_{(1)}\otimes S(\lambda^3_{(1)})\big]\cdots \tag{118b}$$

$$\xrightarrow{\cdots\alpha'\cdots}\cdots\big[c_{(2)}a_{(2)}\lambda^1_{(1)}\otimes c_{(1)}S(\lambda^3_{(1)(1)})\lambda^4_{(1)}\big]\otimes\big[S(\lambda^4_{(2)})\otimes dS(\lambda^3_{(1)(2)})\big]\cdots$$
$$\mapsto\ [e\otimes g]\otimes\big[c_{(2)}a_{(2)}\lambda^1_{(1)}\otimes c_{(1)}S(\lambda^3_{(1)(1)})\lambda^4_{(1)}\big]\cdots \tag{118c}$$

$$\xrightarrow{\cdots\bar{\alpha}\cdots}\big[e_{(2)}c_{(2)}a_{(2)}\lambda^1_{(1)}\otimes S(\lambda^5_{(1)})\big]\otimes\big[\lambda^5_{(3)}e_{(1)}c_{(1)}S(\lambda^3_{(1)(1)})\lambda^4_{(1)}\otimes\lambda^5_{(2)}g\big]\cdots$$
$$\mapsto\ \cdots[h\otimes i]\otimes\big[e_{(2)}c_{(2)}a_{(2)}\lambda^1_{(1)}\otimes S(\lambda^5_{(1)})\big]\cdots \tag{118d}$$

$$\xrightarrow{\cdots\alpha'\cdots}\cdots\big[h_{(2)}e_{(2)}c_{(2)}a_{(2)}\lambda^1_{(1)}\otimes h_{(1)}S(\lambda^5_{(1)(1)})\lambda^6_{(1)}\big]\otimes\big[S(\lambda^6_{(2)})\otimes iS(\lambda^5_{(1)(2)})\big]\cdots$$
$$\mapsto\ \cdots\big[S(\lambda^6_{(2)})S(\lambda^5_{(1)(1)})\otimes iS(\lambda^5_{(1)(2)})\big]\otimes[j\otimes k]\otimes\big[h_{(2)}e_{(2)}c_{(2)}a_{(2)}\lambda^1_{(1)}\otimes h_{(1)}\lambda^6_{(1)}\big]\cdots \tag{118e}$$

$$\xrightarrow{\cdots\alpha'\cdots}\cdots\big[j_{(2)}h_{(3)}e_{(2)}c_{(2)}a_{(2)}\lambda^1_{(1)}\otimes j_{(1)}h_{(2)}\lambda^6_{(1)(2)}\lambda^7_{(1)}\big]\otimes\big[S(\lambda^7_{(2)})\otimes kh_{(1)}\lambda^6_{(1)(1)}\big]\cdots$$
$$=\ \cdots\big[j_{(2)}h_{(3)}e_{(2)}c_{(2)}a_{(2)}\lambda^1_{(1)}\otimes j_{(1)}h_{(2)}\lambda^7_{(1)}\big]\otimes\big[S(\lambda^7_{(2)})\lambda^6_{(1)(2)}\otimes kh_{(1)}\lambda^6_{(1)(1)}\big]\cdots$$
$$\mapsto\ \cdots[m\otimes n]\otimes\big[j_{(2)}h_{(3)}e_{(2)}c_{(2)}a_{(2)}\lambda^1_{(1)}\otimes j_{(1)}h_{(2)}\lambda^7_{(1)}\big] \tag{118f}$$

$$\xrightarrow{\cdots\bar{\alpha}\cdots}\cdots\big[S(\lambda^1_{(2)})\otimes S(\lambda^2_{(2)})\big]\otimes\big[m_{(2)}j_{(2)}h_{(3)}e_{(2)}c_{(2)}a_{(2)}\lambda^1_{(1)}\otimes S(\lambda^8_{(1)})\big]\otimes\big[\lambda^8_{(3)}m_{(1)}j_{(1)}h_{(2)}\lambda^7_{(1)}\otimes\lambda^8_{(2)}n\big]\cdots \tag{118g}$$

$$\xrightarrow{\cdots\frac{1}{|K|}\cdot\text{ev}_T\cdots}\frac{1}{|K|}\cdot|K|^2\int\big(m_{(2)}j_{(2)}h_{(3)}e_{(2)}c_{(2)}a_{(2)}\lambda^1_{(1)}S(\lambda^1_{(2)})\big)\int\big(S(\lambda^8_{(1)})S(\lambda^2_{(2)})\big)\cdots$$
$$=\ |K|\int\big(m_{(2)}j_{(2)}h_{(3)}e_{(2)}c_{(2)}a_{(2)}\big)\int\big(S(\lambda^8_{(1)})S(\lambda^2_{(2)})\big)\big[\lambda^5_{(3)}e_{(1)}c_{(1)}S(\lambda^3_{(1)(1)})\lambda^4_{(1)}\otimes\lambda^5_{(2)}g\big]$$
$$\otimes\big[S(\lambda^6_{(2)})S(\lambda^5_{(1)(1)})\otimes iS(\lambda^5_{(1)(2)})\big]\otimes\big[S(\lambda^7_{(2)})\lambda^6_{(1)(2)}\otimes kh_{(1)}\lambda^6_{(1)(1)}\big]$$
$$\otimes\big[S(\lambda^4_{(2)})\otimes dS(\lambda^3_{(1)(2)})\big]\otimes\big[\lambda^3_{(3)}a_{(1)}\lambda^2_{(1)}\otimes\lambda^3_{(2)}b\big]\otimes\big[\lambda^8_{(3)}m_{(1)}j_{(1)}h_{(2)}\lambda^7_{(1)}\otimes\lambda^8_{(2)}n\big]$$
$$=\ |K|\int\big(m_{(2)}j_{(2)}h_{(3)}e_{(2)}c_{(2)}a_{(2)}\big)\int\big(S(\lambda^2)\big)\big[\lambda^5_{(3)}e_{(1)}c_{(1)}S(\lambda^3_{(1)(1)})\lambda^4_{(1)}\otimes\lambda^5_{(2)}g\big]$$
$$\otimes\big[S(\lambda^6_{(2)})S(\lambda^5_{(1)(1)})\otimes iS(\lambda^5_{(1)(2)})\big]\otimes\big[S(\lambda^7_{(2)})\lambda^6_{(1)(2)}\otimes kh_{(1)}\lambda^6_{(1)(1)}\big]$$
$$\otimes\big[S(\lambda^4_{(2)})\otimes dS(\lambda^3_{(1)(2)})\big]\otimes\big[\lambda^3_{(3)}a_{(1)}S(\lambda^8_1)\otimes\lambda^3_{(2)}b\big]\otimes\big[\lambda^8_{(3)}m_{(1)}j_{(1)}h_{(2)}\lambda^7_{(1)}\otimes\lambda^8_{(2)}n\big]$$
$$=\ \int\big(m_{(2)}j_{(2)}h_{(3)}e_{(2)}c_{(2)}a_{(2)}\big)\big[\lambda^5_{(3)}e_{(1)}c_{(1)}S(\lambda^3_{(1)})\lambda^4_{(1)}\otimes\lambda^5_{(2)}g\big]$$
$$\otimes\big[S(\lambda^6_{(2)})S(\lambda^5_{(1)(1)})\otimes iS(\lambda^5_{(1)(2)})\big]\otimes\big[S(\lambda^7_{(2)})\lambda^6_{(1)(2)}\otimes kh_{(1)}\lambda^6_{(1)(1)}\big]$$
$$\otimes\big[S(\lambda^4_{(2)})\otimes dS(\lambda^3_{(2)})\big]\otimes\big[\lambda^3_{(3)}a_{(1)}S(\lambda^8_{(1)})\otimes\lambda^3_{(2)}b\big]\otimes\big[\lambda^8_{(3)}m_{(1)}j_{(1)}h_{(2)}\lambda^7_{(1)}\otimes\lambda^8_{(2)}n\big]. \tag{118h}$$

It follows that our face projector is a composition of the vertex and face projectors of the Kitaev model. In more detail, the face projector $P_f$ is the following sequence of operations in the Kitaev model:

i) Add a vertex in the middle of every edge. Previously unlabelled edges are labelled by $1_K \in K$. This step comes from the fact we used $T$ instead of $X$ in the computation (118).

ii) Apply the vertex projectors $A_v^\lambda$ for all vertices adjacent to the face.

iii) Apply the face projector $B_f^\int$ (cf. [56, Sec. 4]).

Finally, we give a quick overview of the comparison of the internal Levin–Wen model and the Kitaev model:

| Internal Levin–Wen model | Kitaev model |
|---|---|
| admissible, trivalent graph $\Gamma$ | modified graph $\Gamma'$ |
| internal LW-datum $\mathbb{A}_K$ | finite-dimensional, semisimple Hopf algebra $K$ |
| state space $V$ | extended space $\mathcal{T}$ |
| ground states $V_0$ | protected space $\mathcal{M}$ |
| face projectors | face projectors |
| vertex and edge projectors | vertex projectors |

# 6   Conclusion

We have introduced the concept of internal Levin–Wen models, a novel approach to describing and realising topological phases of matter. By employing the notion of an orbifold datum within a given phase, these models provide a powerful framework for condensing anyons and transitioning between different 2-dimensional topological phases. Internal Levin–Wen models are not only mathematically intriguing but also offer a deeper understanding of the relationships between various phases of matter. They bridge the gap between previously studied models, such as Levin–Wen and Kitaev models, as well as the more abstract model for anyon condensation using separable commutative algebra objects in a given phase, providing a unifying perspective on these examples and the ones yielded by the instances of orbifold data beyond those discussed in this work, such as in [46].

The internal Levin–Wen models are characterised by their commuting-projector Hamiltonians, which are constructed out of defects in the initial phase $\mathcal{C}$, arranged into a lattice configuration. The ground state spaces of these Hamiltonians give rise to new topological phases of matter, which, depending on the input, can yield an arbitrary phase which is Witt-equivalent to $\mathcal{C}$. Unless $\mathcal{C}$ is the trivial phase, the model fails to be local, meaning that unlike conventional lattice models, where the state space consists of independent degrees of freedom at each lattice site, internal Levin–Wen models introduce a lattice of anyonic particles, resulting in a collective, non-local state space. The non-locality results in more involved computations, which nevertheless are mostly repetitive in nature and can be carried out for example using computer algebra. To illustrate this point we have performed a number of computations in the context of internal Levin–Wen models in varying degrees of abstraction. We have shown how one gets an explicit formula for the terms of the Hamiltonian based on an arbitrary orbifold datum and specialised them for the instances when it is obtained from a separable commutative algebra in $\mathcal{C}$, a spherical fusion category and a semisimple finite-dimensional Hopf algebra. The outcomes of these computations demonstrate the aforementioned relation between the internal Levin–Wen models and those of anyon condensation, original Levin–Wen and the Kitaev models.

Future directions building on this work may involve a deeper exploration of the properties of the internal Levin–Wen Hamiltonian, such as the analysis of excitations and the development of a string-net description of the ground state space. Additionally, the notions of orbifold data in defect TFTs, on which this work relies, are versatile enough to suggest similar applications to the study of topological phases on manifolds with other tangential structures than orientations and/or beyond the setting of semisimplicity.

## Acknowledgments

The authors would like to thank Kevin Walker and Sukhwinder Singh for helpful discussions.

**Funding information**   IR is partially supported by the Deutsche Forschungsgemeinschaft via the Cluster of Excellence EXC 2121 "Quantum Universe" - 390833306. VM greatly acknowledges the support of Max Planck Institute for Mathematics (MPIM) in Bonn, the Junior Research Fellows' programme at Erwin Schrödinger Institute for Mathematics and Physics (ESI) in Vienna as well as the Research Council of Lithuania under the grant S-PD-22-79 "Lattice systems in topological quantum field theories".

## A    Conditions on orbifold data

Below we list the conditions on an orbifold datum $(A, T, \alpha, \overline{\alpha}, \psi, \phi)$ in terms of string diagrams (Figure 22) and in terms of surface diagrams (Figure 23).

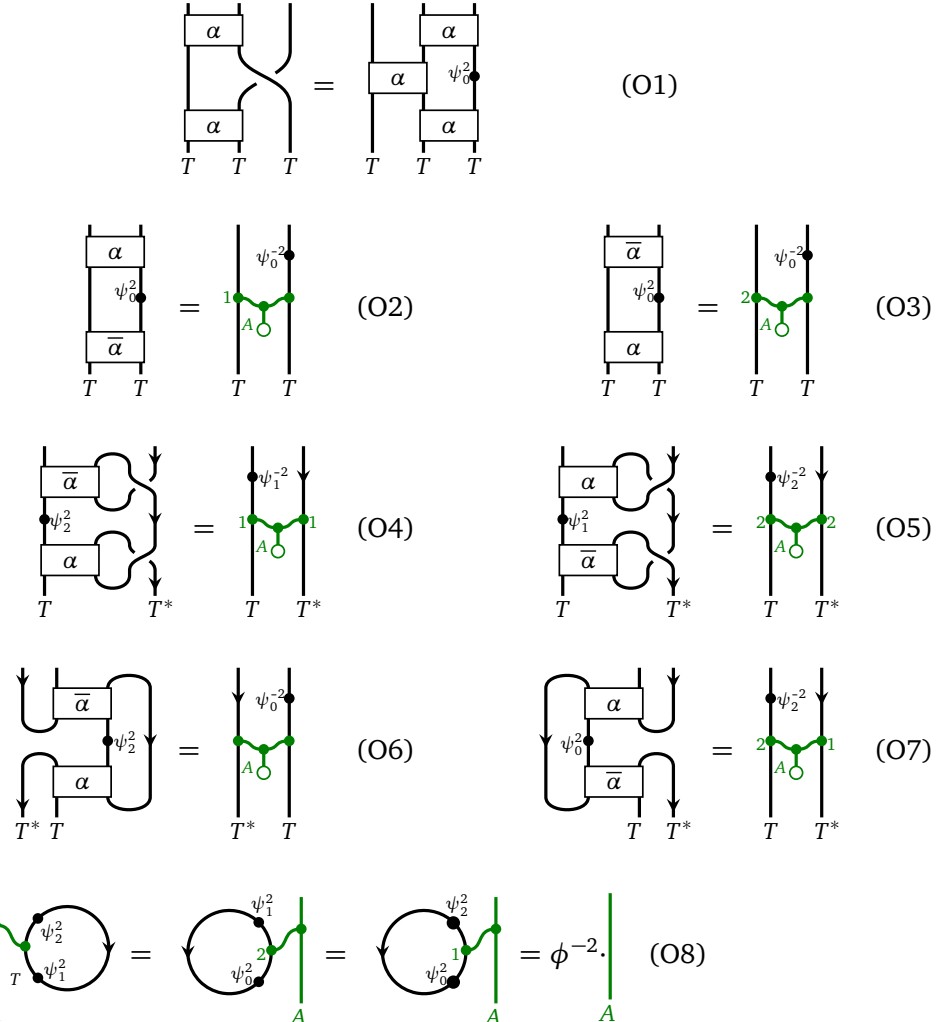

Figure 22: Conditions on a 3-dimensional orbifold datum $(A, T, \alpha, \overline{\alpha}, \psi, \phi)$.

Figure 23: Conditions on a 3-dimensional orbifold datum $(A, T, \alpha, \overline{\alpha}, \psi, \phi)$. Cf. [17, Fig. 18].

# B  Proof of proposition 9

In this section we give a proof that the tuple $(A = K_{\mathrm{Fr}}, T, \alpha, \overline{\alpha}, \psi, \phi)$ in Section 3.2.3 defines an orbifold datum. More precisely, we show that it fulfils the identities (O1), (O3), (O6) and (O8) in Figure 22. The remaining conditions can be shown by analogous computations. We denote elements of $K$ or $\overline{K}$ by letters $a, \ldots, f$, distinguish different copies of the Haar integral $\lambda = \lambda^1 = \lambda^2 = \ldots$ by upper indices and remind the reader that $\psi = \mathrm{id}_K$ is trivial. The computations use the identities (58), (59a), (59b), (59c).

To show identity (O1) we compute both sides of it separately. The left-hand side reads $(\alpha \otimes \mathrm{id}_T) \circ (\mathrm{id}_T \otimes \tau_{T,T}) \circ (\alpha \otimes \mathrm{id}_T)$, where $\tau_{T,T}$ denotes the swap of the tensor factors. We compute this map as an endomorphism of $T^{\otimes 3} = K^{\otimes 6}$ by applying the maps in the composition step-by-step. The two copies of $K$ belonging to one copy of $T$ are indicated by square brackets, e.g. $[a \otimes b] \in K^{\otimes 2} = T$. The left-hand side of (O1) then yields:

$$[a \otimes b] \otimes [c \otimes d] \otimes [e \otimes f] \xrightarrow{\alpha \otimes \mathrm{id}_T} \left[S(\lambda^1_{(1)}) \otimes b_{(1)}d\right] \otimes \left[\lambda^1_{(3)}a \otimes \lambda^1_{(2)}b_{(2)}c\right] \otimes [e \otimes f]$$

$$\xrightarrow{\mathrm{id}_T \otimes \tau_{T,T}} \left[S(\lambda^1_{(1)}) \otimes b_{(1)}d\right] \otimes [e \otimes f] \otimes \left[\lambda^1_{(3)}a \otimes \lambda^1_{(2)}b_{(2)}c\right]$$

$$\xrightarrow{\alpha \otimes \mathrm{id}_T} \left[S(\lambda^2_{(1)}) \otimes b_{(1)}d_{(1)}f\right] \otimes \left[\lambda^2_{(3)}S(\lambda^1_{(1)}) \otimes \lambda^2_{(2)}b_{(2)}d_{(2)}e\right] \otimes \left[\lambda^1_{(3)}a \otimes \lambda^1_{(2)}b_{(3)}c\right]. \quad \text{(B.1)}$$

The right-hand side of (O1) yields:

$$[a \otimes b] \otimes [c \otimes d] \otimes [e \otimes f] \xrightarrow{\mathrm{id}_T \otimes \alpha} [a \otimes b] \otimes \left[S(\lambda^1_{(1)}) \otimes d_{(1)}f\right] \otimes \left[\lambda^1_{(3)}c \otimes \lambda^1_{(2)}d_{(2)}e\right]$$

$$\xrightarrow{\alpha \otimes \mathrm{id}_T} \left[S(\lambda^2_{(1)}) \otimes b_{(1)}d_{(1)}f\right] \otimes \left[S(\lambda^2_{(3)})a \otimes \lambda^2_{(2)}b_{(2)}S(\lambda^1_{(1)})\right] \otimes \left[\lambda^1_{(3)}c \otimes \lambda^1_{(2)}d_{(2)}e\right]$$

$$\xrightarrow{\mathrm{id}_T \otimes \alpha} \left[S(\lambda^2_{(1)}) \otimes b_{(1)}d_{(1)}f\right] \otimes \left[S(\lambda^3_{(1)}) \otimes \lambda^2_{(2)}b_{(2)}S(\lambda^1_{(2)})\lambda^1_{(3)}d_{(2)}e\right]$$

$$\otimes \left[\lambda^3_{(3)}\lambda^2_{(4)}a \otimes \lambda^3_{(2)}\lambda^2_{(3)}b_{(3)}S(\lambda^1_{(1)})\lambda^1_{(4)}c\right]$$

$$= \left[S(\lambda^2_{(1)}) \otimes b_{(1)}d_{(1)}f\right] \otimes \left[\lambda^2_{(3)}S(\lambda^3_{(1)}) \otimes \lambda^2_{(2)}b_{(2)}d_{(2)}e\right] \otimes \left[\lambda^3_{(3)}a \otimes \lambda^3_{(2)}b_{(3)}c\right], \quad \text{(B.2)}$$

where we have used the identities $\varepsilon(\lambda^1) = 1$ and $S(\lambda^3_{(1)}) \otimes \lambda^3_{(2)}k = kS(\lambda^3_{(1)}) \otimes \lambda^3_{(2)}$ for $k = \lambda^2_{(3)}$ in the last step. By comparing with the left side we conclude that (O1) holds.

For (O3) we compute the left-hand side of the identity:

$$[a \otimes b] \otimes [c \otimes d] \xrightarrow{\alpha} \left[S(\lambda^1_{(1)}) \otimes b_{(1)}d\right] \otimes \left[\lambda^1_{(3)}a \otimes \lambda^1_{(2)}b_{(2)}c\right]$$

$$\xrightarrow{\bar{\alpha}} \left[S(\lambda^1_{(1)})\lambda^1_{(4)}a \otimes S(\lambda^2_{(1)})\right] \otimes \left[\lambda^2_{(3)}S(\lambda^1_{(2)})\lambda^1_{(3)}b_{(2)}c \otimes \lambda^2_{(2)}b_{(1)}d\right]$$

$$= \left[a \otimes S(\lambda^2_{(1)})\right] \otimes \left[\lambda^2_{(3)}b_{(2)}c \otimes \lambda^2_{(2)}b_{(1)}d\right] = \left[a \otimes bS(\lambda^2_{(1)})\right] \otimes \left[\lambda^2_{(3)}c \otimes \lambda^2_{(2)}d\right]. \quad \text{(B.3)}$$

The right-hand side of (O3) is given by the formula in (62), which is identical to the left-hand side we just computed.

We now compute the left side of (O6):

$$[a \otimes b] \otimes [c \otimes d] \xrightarrow{\mathrm{id} \otimes \mathrm{coev}_T} [a \otimes b] \otimes [c \otimes d] \otimes \left[\lambda^1_{(1)} \otimes \lambda^2_{(1)}\right] \otimes \left[S(\lambda^1_{(2)}) \otimes S(\lambda^2_{(2)})\right]$$

$$\xrightarrow{\mathrm{id}_{T^*} \otimes \alpha \otimes \mathrm{id}_T} [a \otimes b] \otimes \left[S(\lambda^3_{(1)}) \otimes d_{(1)}\lambda^2_{(1)}\right] \otimes \left[\lambda^3_{(3)}c \otimes \lambda^3_{(2)}d_{(2)}\lambda^1_{(1)}\right] \otimes \left[S(\lambda^1_{(2)}) \otimes S(\lambda^2_{(2)})\right]$$

$$\xrightarrow{\mathrm{ev}_T \otimes \mathrm{id}_{T^* \otimes T}} |K|^2 \int (aS(\lambda^3_{(1)})) \int (bd_{(1)}S(\lambda^2_{(1)})) \left[\lambda^3_{(3)}c \otimes \lambda^3_{(2)}d_{(2)}\lambda^1_{(1)}\right] \otimes \left[S(\lambda^1_{(2)}) \otimes S(\lambda^2_{(2)})\right]$$

$$= \left[a_{(2)}c \otimes a_{(1)}d_{(2)}\lambda^1_{(1)}\right] \otimes \left[S(\lambda^1_{(2)}) \otimes bd_{(1)}\right]$$

$$\xrightarrow{\widetilde{\mathrm{coev}}_T \otimes \mathrm{id}_{T \otimes T^*}} \left[S(\lambda^2_{(1)}) \otimes S(\lambda^3_{(1)})\right] \otimes \left[\lambda^2_{(2)} \otimes \lambda^3_{(2)}\right] \otimes \left[a_{(2)}c \otimes a_{(1)}d_{(2)}\lambda^1_{(1)}\right]$$

$$\otimes \left[S(\lambda^1_{(2)}) \otimes bd_{(1)}\right]$$

$$\xrightarrow{\mathrm{id}_{T^*} \otimes \bar{\alpha} \otimes \mathrm{id}_T} \left[S(\lambda^2_{(1)} \otimes S(\lambda^3_{(1)}))\right] \otimes \left[\lambda^2_{(3)}a_{(2)}c \otimes S(\lambda^4_{(1)})\right]$$

$$\otimes \left[\lambda^4_{(3)}\lambda^2_{(2)}a_{(1)}d_{(2)}\lambda^1_{(1)} \otimes \lambda^4_{(2)}\lambda^3_{(2)}\right] \otimes \left[S(\lambda^1_{(2)}) \otimes bd_{(1)}\right]$$

$$\xrightarrow{\text{id}_{T^*\otimes T}\otimes\widetilde{\text{ev}}_T} |K|^2 \int(\lambda_{(3)}^4\lambda_{(2)}^2 a_{(1)}d_{(2)}\lambda_{(1)}^1 S(\lambda_{(2)}^1))\int(\lambda_{(2)}^4\lambda_{(2)}^3 bd_{(1)})$$

$$\cdot\left[S(\lambda_{(1)}^2)\otimes S(\lambda_{(1)}^3)\right]\otimes\left[\lambda_{(3)}^2 a_{(2)}c\otimes S(\lambda_{(1)}^4)\right]$$

$$=|K|\int(\lambda_{(3)}^4\lambda_{(2)}^2 a_{(1)}d_{(2)})\left[S(\lambda_{(1)}^2)\otimes bd_{(1)}\lambda_{(2)}^4\right]\otimes\left[\lambda_{(3)}^2 a_{(2)}c\otimes S(\lambda_{(1)}^4)\right]$$

$$=\left[\overline{\lambda_{(1)}^2}\otimes bd_{(1)}S(d_{(2)})S(a_{(1)})S(\lambda_{(2)}^2)\right]\otimes\left[\lambda_{(4)}^2 a_{(3)}c\otimes\lambda_{(3)}^2 a_{(2)}d_{(3)}\right]$$

$$=\left[S(\lambda_{(1)}^2)\otimes bS(a_{(1)})S(\lambda_{(2)}^2)\right]\otimes\left[\lambda_{(4)}^2 a_{(2)}c\otimes\lambda_{(3)}^2 a_{(3)}d\right]$$

$$=\left[aS(\lambda_{(1)}^2)\otimes bS(\lambda_{(2)}^2)\right]\otimes\left[\lambda_{(4)}^2 c\otimes\lambda_{(3)}^2 d\right]. \tag{B.4}$$

Here we used the identity $\int(\lambda_{(1)}k)S(\lambda_{(2)})=\frac{1}{|K|}k$ for $k\in K$. The right-hand side of (O6) yields:

$$\left([a\otimes b]\triangleleft_0 S(\lambda_{(1)})\right)\otimes\left(\lambda_{(2)}\triangleright_0[c\otimes d]\right)=\left[aS(\lambda_{(1)})\otimes bS(\lambda_{(2)})\right]\otimes\left[\lambda_{(4)}c\otimes\lambda_{(3)}d\right], \tag{B.5}$$

which is identical to the left-hand side as expected.

Finally, we show one of the identities in (O8), starting with the left-hand side:

$$a\xrightarrow{\text{id}\otimes\text{coev}_T} a\otimes\left[\lambda_{(1)}^1\otimes\lambda_{(1)}^2\right]\otimes\left[S(\lambda_{(2)}^1)\otimes S(\lambda_{(2)}^2)\right]$$

$$\xrightarrow{(\mu\otimes\triangleright_0)\circ\Delta(1_K)\otimes\text{id}_{T^*}} aS(\lambda_{(1)}^3)\otimes\left[\lambda_{(3)}^3\lambda_{(1)}^1\otimes\lambda_{(2)}^3\lambda_{(1)}^2\right]\otimes\left[S(\lambda_{(2)}^1)\otimes S(\lambda_{(2)}^2)\right]$$

$$\xrightarrow{\widetilde{\text{coev}}_T} aS(\lambda_{(1)}^3)\cdot|K|^2\int(\lambda_{(3)}^3\lambda_{(1)}^1 S(\lambda_{(2)}^1))\int(\lambda_{(2)}^3\lambda_{(1)}^2 S(\lambda_{(2)}^2))$$

$$=aS(\lambda_{(1)}^3)\cdot|K|^2\int(\lambda_{(3)}^3)\int(\lambda_{(2)}^3)=|K|\cdot a=\frac{1}{\phi^2}\cdot a. \tag{B.6}$$

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
