# Peer review of "Internal Levin-Wen models"

_SciPost Physics, doi:SciPost Phys. 17, 088 (2024)_

## Round 1 · Referee Report · Anonymous (Referee 1) · 2023-12-22

Report

\documentclass[12pt]{article}
\textheight 24cm
\textwidth 16.5cm
\topmargin -1cm
\oddsidemargin -0.5cm
\usepackage{amsfonts}
\usepackage{amsmath}

\begin{document}

\begin{center}
{\large Report on the paper {\em "Internal Levin-Wen models"}, by V. Mulevi\v{c}ius, I. Runkel, and T. Voss}
\end{center}

\vspace{3ex}
I believe that this is a very good paper, that bridges the gap between mathematical constructions in topological field
theory and the more concrete world of quantum lattice models. It is carefully written, probably as clearly as possible,
given the assigned goal of presenting all the required steps, from the category theory-based construction of TFT's with
defects, including generalized orbifolds, to the detailed derivation of Levin-Wen models.
The prerequisite section 2 covers a lot of material, most of it based on previous works from two authors and their collaborators,
but it manages to focus on the essential ideas without too many technical details. The core of the paper is the
construction of internal Levin-Wen models, given in section 3, which also provides the algebraic data needed to cover the three
concrete applications presented later, namely the case of condensable algebras $A$ in a modular fusion category $\mathcal{C}$,
the original Levin-Wen model obtained with $\mathcal{C}=\mathrm{Vec}$ and orbifold data taken from a spherical fusion category,
and a generalized Kitaev model arising from $\mathcal{C}=\mathrm{Vec}$ and orbifold data defined by a semisimple Hopf algebra.
Section 4 presents the key steps to implement this general mathematical setting with specific quantum mechanical lattice models.
An interesting feature of these models is that their Hilbert space is not in general a tensor product of local spaces attached to
vertices and links of the 2D lattice, but it is realized as a space of homomorphisms in the incipient category $\mathcal{C}$
from the identity object to a tensor product of an object $X$ (and its dual) that can be viewed physically as a configuration space
for a collection of $X$-type anyons located on the lattice sites. This lattice can be put on a surface of arbitrary genus, which complicates
the construction of the local projectors, attached to sites, edges and faces of the lattice, but everything is worked out in detail, with
a useful example on a piece of hexagonal lattice on a torus. The last section 5 presents the explicit forms of lattice models thus obtained
for the three settings mentioned above. Clearly, this paper is not easy to read for someone coming from the condensed matter
community. But I feel that, by its detailed and careful exposition, it could stimulate further studies of this larger class
of string net models within this community. Therefore, it is clear to me that this paper should be published as it stands, given that I don't see
exactly how its presentation could be improved, while keeping its length fixed.

\end{document}

---

## Round 1 · Referee Report · Anonymous (Referee 2) · 2024-2-20

Strengths

1. The paper is well-referenced, and does a good job of discussing related constructions and situating this work in relation to those.
2. There are many nice figures, to help illustrate the technical results. These figures give an intuitive depiction of how, for example, the TQFT is used to construct the Hamiltonian, which is very helpful.
3. The paper is self-contained, and makes an effort to present the necessary background material in a relatively accessible way. This, in itself, is potentially of value, particularly for physicists who are entering this particular community.
4. The method used to make this construction is new and original.

Weaknesses

1. The paper does not do a great job of motivating why another construction of commuting projector models is interesting. In particular, there are closely related constructions, such as those of 56 and 34, which if I understand correctly, can realize with commuting projector models the same set of phases. The authors could make a stronger case in the introduction for one or more of the following:
-- What do we learn about topological phases from this construction?
-- What kinds of computations (numerical or analytical) might be more practical with this construction than with existing ones?
-- Does this construction give a more general template that could be used to build new types of models that might realize novel topological or SPT phases?

2. There is a lot of data involved in the construction. While this data is thoroughly explained at an abstract level, some concrete examples earlier in the paper would (at least for me) make it easier to understand what these are, and why they play the roles that they do. For example, the authors could choose a relatively familiar MTC (maybe the Ising CFT?) and illustrate all of the choices of the remaining data, and what kinds of models the construction yields.

Report

This work presents an alternative construction of commuting projector models for topological phases in 2+1 dimensions, which realizes a more general class of phases than the original construction by Levin and Wen. This is achieved essentially by incorporating additional data into the construction, which can be viewed as the data for an ordinary topological field theory with defects, together with what the authors refer to as an orbifold datum, which essentially specifies what is often referred to as a condensation.

Overall the paper is of high quality, and the construction is new and sufficiently original to merit publication in Scipost physics. That said, I think that the authors could do more to make the work accessible and of interest to a wider audience. The details of the construction are highly technical, and many readers -- especially those who are not so familiar with the formalism of anyon models and topological quantum field theories-- may be left wondering what the payoff of understanding the details of this construction would be. I suggest that the authors address this by adding one (or more) relatively simple, concrete examples near the beginning of the paper that illustrate the main advantages of their construction relative to others in the literature.

Requested changes

A. Existing constructions of commuting projector models for topological phases fall into two categories:
1. If the TQFT is a Drinfeld center (or, equivalently, it admits a gapped boundary to the vacuum), then it can be realized by a 2D commuting projector model -- essentially, a generalized string net.
2. If the TQFT is not a Drinfeld center, it can be realized at the surface of a 3D commuting projector model of the Walker-Wang type. (As discussed in Ref. 34, the Walker-Wang bulk needs only to have the same anomaly).

In particular, zero-correlation length 2D bulk models all admit gapped boundaries, and a commuting projector model for a TQFT with e.g. a non-vanishing central charge is possible in these existing constructions only at the boundary of a 3-dimensional system. I am confused about the status of your construction in this regard. The paper uses space-time diagrams to depict the Hamiltonian, so that as I understand it here the spatial manifold is always assumed to have no boundary (being itself the boundary of a space-time). But many of the examples are Hamiltonians that are actually realized as commuting projectors on 2D lattice models, for which a gapped spatial boundary is possible. I anticipate that in cases where the topological theory is not a Drinfeld center, the 3-dimensional nature of the model is more fundamental, but it was not clear to me from the current text how this plays out in the actual models. A detailed discussion of some very simple examples (e.g. the Toric code, versus a simple abelian Chern-Simons theory such as U(1)_2) would probably help clarify this distinction. It would also be helpful if the authors can comment on this point in the introduction.

B. As noted above, I personally think that having a thorough discussion of all models that can be constructed from some simple choice of C early on in the paper would make it easier to understand the general idea.

---

## Round 2 · Referee Report · Anonymous (Referee 3) · 2024-8-19

Strengths

1- Novel lattice construction of topological orders beyond those accessible with standard Levin-Wen construction 2- Clear exposition with many high-quality diagrams that provide intuition for the abstract underlying categorical concepts

Report

Dear authors and editors,

my sincere apologies for the delayed response.

In short, I agree with the previous referees that this is a very valuable addition to the literature and I am happy to recommend it for publication. As far as I can tell, the authors have addressed the comments made by the previous with the improvements in their resubmission.

I have only a few minor comments and a request for clarification that the authors might find useful:

  • It is mentioned that the lattice model is non-local, because the state space on which the Hamiltonian is not of tensor product form. As far as I can tell however, the individual terms in the Hamiltonian still only act on a subset of the degrees of freedom that surround a vertex/edge/fase, and does not introduce long-range interactions. Is this correct? If so, the use of "non-locality" is a bit confusing, since this is most often taken to mean a Hamiltonian which includes long-range interactions between degrees of freedom that are far away from each other. As an example, lattice gauge theories (quantum double models in which one manifestly enforces the vertex term) also do not have a tensor product state space, but we still consider these models to be local in that there is a finite (Lieb-Robinson) velocity at which information can propagate due to the locality of the interactions.

  • From a practical point of view, realising standard Levin-Wen models is difficult due to the number of degrees of freedom on which the Hamiltonian acts (in the case of no multiplicities, this is a 12-body term). This is a very relevant problem however, since the explicit realisation of e.g. the Fibonacci topological order provides a means to do universal topological quantum computation via the braiding of the topological quasiparticles. The standard Levin-Wen construction can only produce doubled Fibonacci; does the internal Levin-Wen construction of this paper for the Fibonacci topological phase admit an equally complicated Hamiltonian since it lives on top of a conventional Levin-Wen model?

  • Given that this construction is able to produce chiral topological order, is there any way to use these lattice models to explore the necessarily gapless boundary theories for these models? Is it easy to see that there cannot exist a gapped boundary theory?

Recommendation

Publish (easily meets expectations and criteria for this Journal; among top 50%)

---

## Round 2 · Referee Report · Anonymous (Referee 4) · 2024-8-28

Strengths

I think that this paper is very interesting, the main result is an important contribution to the field. Moreover, the paper is very well written with many beautiful illustrating pictures.

Weaknesses

I did not find any obvious weakness in this paper.

Report

In recent years, one of the authors Runkel and his collaborators have systematically developed the theory of orbiford constructions in 2+1D TQFT. It is a state-sum construction that largely generalizes the so-called anyon condensation construction, which is reversible. The orbiford construction is reversible and closer to physical reality. As far as I can tell, this work is a Hamiltonian version of this state-sum construction. I think that it is a very interesting and important result, which was long anticipated in this field.

Requested changes

In the orbiford data, an algebra A decorates a surface (1-codimensional). I suspect that this work is the first physical realization of the condensation theory of topological defects of codimension 1 (see arXiv:2403.07813). It is helpful if the authors can add some comments of about it.

Recommendation

Publish (surpasses expectations and criteria for this Journal; among top 10%)

---

## Round 2 · Author Response

We are grateful to the referees for reading our preprint, and for several helpful suggestions and remarks. We addressed the changes A and B as inquired by one of the referees in the following way:

• For personal convenience we start with change B, which requested a discussion on what models can be constructed given the initial phase C, perhaps for a concrete simple choice of C. In the previous version this was briefly addressed in the introduction: the phases, realized as ground states spaces of our models, are exactly the phases D which are Witt-equivalent to C, i.e. the product CxD’ has to be a Drinfeld centre. Admittedly this was too brief, to remedy this we added a new section “Universality of Internal Levin—Wen models” to the introduction. In it we explain how our models are capable of performing both anyon condensation as well as the opposite procedure: de-condensation. We also sketch a particular example of this: su(2)_10 phase can be condensed into the Ising phase, so there exist an input for our model, which would combine Ising anyons in a way such that su(2)_10 would emerge (see newly added diagram (1.2)). We do think however that a detailed discussion of this and other similar examples would significantly expand the volume of this already lengthy paper, so, with referees’ permission, we would like to postpone this for a future work.

• Change A requested to explain more the relation between the models introduced in the paper and the models living at a boundary of a Walker--Wang (WW) lattice models. To address this, we expanded the subsection in the introduction where WW models are discussed – we now emphasise that WW provide one with a way to realise topological phases which possibly are not Drinfeld centres, whereas our models show how to entangle and fuse/braid anyons in an existing system so that a new phase emerges independently on what these systems are. For example, the system supporting the initial anyons can come from a boundary WW model, but they can also be realised in other ways (e.g. Ising anyons as Majorana fermions).

Besides these changes, we also corrected a number of typographic errors.

---

## Editorial Decision

published